# Dissecting Sample Hardness: A Fine-Grained Analysis of Hardness Characterization Methods for Data-Centric AI

**Nabeel Seedat**
University of Cambridge
ns741@cam.ac.uk

**Fergus Imrie**
UCLA
imrie@ucla.edu

**Mihaela van der Schaar**
University of Cambridge
mv472@cam.ac.uk

## Abstract

Characterizing samples that are difficult to learn from is crucial to developing highly performant ML models. This has led to numerous Hardness Characterization Methods (HCMs) that aim to identify "hard" samples. However, there is a lack of consensus regarding the definition and evaluation of "hardness". Unfortunately, current HCMs have only been evaluated on specific types of hardness and often only qualitatively or with respect to downstream performance, overlooking the fundamental quantitative identification task. We address this gap by presenting a fine-grained taxonomy of hardness types. Additionally, we propose the Hardness Characterization Analysis Toolkit (H-CAT), which supports comprehensive and quantitative benchmarking of HCMs across the hardness taxonomy and can easily be extended to new HCMs, hardness types, and datasets. We use H-CAT to evaluate 13 different HCMs across 8 hardness types. This comprehensive evaluation encompassing over *14K setups* uncovers strengths and weaknesses of different HCMs, leading to practical tips to guide HCM selection and future development. Our findings highlight the need for more comprehensive HCM evaluation, while we hope our hardness taxonomy and toolkit will advance the principled evaluation and uptake of data-centric AI methods.

## 1 Introduction

**Data quality, an important ML problem.** Data quality is crucial to the performance and robustness of machine learning (ML) models (Jain et al., 2020; Gupta et al., 2021; Renggli et al., 2021; Sambasivan et al., 2021; Li et al., 2021). Unfortunately, challenges arise in real-world data that make samples "hard" for ML models to learn from effectively, including but not limited to mislabeling, outliers, and insufficient coverage (Chen et al., 2021; Li et al., 2021). These "hard" samples or data points can significantly hamper the performance of ML models, creating a barrier to ML adoption in practical applications (Bedi et al., 2019; West, 2020). For instance, a model trained on mislabeled samples can lead to inaccurate predictions (Krishnan et al., 2016; Gupta et al., 2021). Outliers can bias the model to learn suboptimal decision boundaries (Liu et al., 2022; Eduardo et al., 2022; Krishnan et al., 2016), harming model performance. Long tails of samples can result in poor model performance for these cases (Feldman, 2020; Hooker et al., 2019; Hooker, 2021; Agarwal et al., 2022). Consequently, "hard" samples can pose serious challenges for training and the performance of ML models, making it crucial to identify these samples. This is especially important in where manually identifying "hard" samples is expensive, impractical, or time-consuming given the scale.

**Characterizing hardness, a growing area.** Recent interest in data-centric AI, which aims to ensure and improve data quality, has led to the development of systematic methods to characterize the data used to train ML models (Liang et al., 2022; Seedat et al., 2023b; 2022b). Data characterization typically assigns scores to each sample based on its learnability and utility for an ML task, thereby facilitating the identification of "hard" samples. We collectively refer to methods that perform the characterization as Hardness Characterization Methods (HCMs).

After samples are characterized, how and for what purpose they are used can differ. For example: (1) curating datasets via sample selection to improve model performance (Maini et al., 2022; Swayamdipta et al., 2020; Seedat et al., 2022a; Northcutt et al., 2021a; Pleiss et al., 2020; Agarwal

et al., 2022; Seedat et al., 2023a), (2) sculpting the dataset to reduce computational requirements while maintaining performance (Toneva et al., 2019; Paul et al., 2021; Sorscher et al., 2022; Mindermann et al., 2022), (3) guiding acquisition of additional samples (Zhang et al., 2022), or (4) understanding learning behavior from a theoretical perspective (Baldock et al., 2021; Shwartz et al., 2022; Jiang et al., 2021).

**Challenges in definition and evaluation.** A fundamental and *overlooked* aspect is that while different HCMs tackle the issue of "hardness", it remains a vague and ill-defined term in the literature. The lack of a clear definition of hardness types has led HCMs, seemingly tackling the same problem, to unintentionally target and assess different aspects of hardness (Table 1). The lack of clarity is further exacerbated by: (1) *qualitative evaluation*: a significant focus on post hoc *qualitative* assessment and downstream improvement, instead of the fundamental hardness identification task, and (2) *narrow and unrepresentative scope*: even when quantitative evaluation has been performed, it has typically focused on a single hardness type, neglecting different manifestations of hardness. The lack of comprehensive and quantitative evaluation means we do not know how different HCMs perform on different hardness types and whether they indeed identify the correct samples of interest.

> *Can we define sample hardness manifestations and then comprehensively and systematically evaluate the capabilities of different HCMs to correctly detect the hard samples?*

**Unified taxonomy and benchmarking framework.** To answer this question, we begin by defining a taxonomy of hardness types across three broad categories: (a) Mislabeling, (b) OoD/Outlier, (c) Atypical. We then introduce the **H**ardness-**C**haracterization **A**nalysis **T**oolkit (**H-CAT**), which, to the best of our knowledge, is the first unified data characterization benchmarking framework focused on hardness. Using H-CAT, we comprehensively and quantitatively benchmark 13 state-of-the-art HCMs across various hardness types. In doing so, we address recent calls for more rigorous benchmarking (Guyon, 2022) and understanding of existing ML methods (Lipton & Steinhardt, 2019; Snoek et al., 2018). We make the following contributions:

**Contributions:** ① *Hardness taxonomy*: we formalize a systematic taxonomy of sample-level hardness types, addressing the current literature's ad hoc and narrow scope. By defining the different dimensions of hardness, our taxonomy paves the way for a more rigorous evaluation of HCMs. ② *Benchmarking framework*: we propose H-CAT, which is both *(i) a benchmarking standard* to evaluate the strengths of different HCMs across the hardness taxonomy and *(ii) a unified software tool* integrating 13 different HCMs. With extensibility in mind, H-CAT can easily incorporate new HCMs, hardness types, and datasets, thus enhancing its utility for both researchers and practitioners. ③ *Systematic & Quantitative HCM evaluation*: we use H-CAT to comprehensively benchmark and evaluate 13 different HCMs across 8 different hardness types, comprising *over 14K* experimental setups. ④ *Insights*: our benchmark provides novel insights into the capabilities of different HCMs when dealing with different hardness types and offers practical usage tips for researchers and practitioners. The variability in HCM performance across hardness types underscores the importance of multi-dimensional evaluation, exposing gaps and opportunities in current HCMs. We hope H-CAT will promote rigorous HCM evaluations and inspire new advances in data-centric AI.

## 2 HARDNESS CHARACTERIZATION AND TAXONOMY

We now outline the hardness characterization problem and formalize a hardness taxonomy for HCMs.

**Learning problem.** Consider the typical supervised learning setting, with $\mathcal{X}$ and $\mathcal{Y}$ input and output spaces, respectively. We assume a k-class classification problem, i.e. $\mathcal{Y} = [k]$, where $[k] = \{1, \ldots, k\}$, with a training dataset $\mathcal{D} = \{(x_i, y_i) \mid i \in [N]\}$ with $N \in \mathbb{N}^+$ samples, where $x_i \in \mathcal{X}$ and $y_i \in \mathcal{Y}$. The goal is to learn a model $f_\theta : \mathcal{X} \to \mathcal{Y}$, parameterized by $\theta \in \Theta$.

**Hardness problem.** To understand the intricacies of data hardness, we start by providing a broad definition of the hardness problem common to all HCMs. As a starting point, let us frame the hardness problem in general terms: specifically, some samples or data points are easier for a model to learn from, whilst others may either be harder to learn from or harm the model's performance.

Formally, this assumes that the training dataset can be decomposed as $\mathcal{D} = \mathcal{D}_e \cup \mathcal{D}_h$, where $\mathcal{D}_e$ are easy samples and $\mathcal{D}_h$ are hard samples. We denote the corresponding joint distributions as $\mathcal{P}_{XY}$, $\mathcal{P}_{XY}^e$ and $\mathcal{P}_{XY}^h$. Going beyond this general definition to different manifestations of hardness requires a more rigorous characterization. However, what constitutes a "hard" sample has not been rigorously

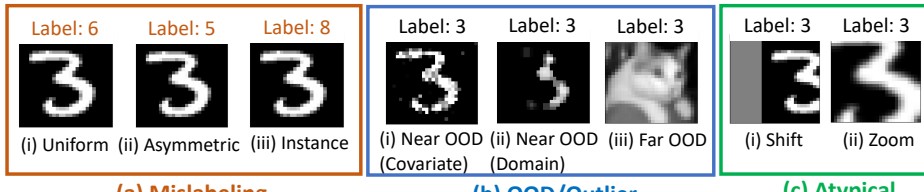

Figure 1: Examples of the hardness types included within our taxonomy and supported by H-CAT. HCMs need to be comprehensively assessed across different dimensions to quantify their ability to handle different hardness types we might expect in practice. See Sec. 2.1 for precise definitions of hardness types.

defined in the literature. To address this gap, we first formalize a taxonomy of hardness in Sec. 2.1, providing a systematic and formal definition of different types of sample-level hardness.

**Data characterization.** The goal of data characterization is to assign a scalar "hardness" score to each sample in $\mathcal{D}$, thereby allowing us to order samples in $\mathcal{D}$ according to their scores. Typically, a selector function applies a threshold $\tau$ and then assigns a hardness label $g \in \mathcal{G}$, where $\mathcal{G} = \{Easy, Hard\}$, to each sample $(x_i, y_i)$, i.e. assign samples in $\mathcal{D}$ to either $\mathcal{D}_e$ or $\mathcal{D}_h$. We refer to these methods as **Hardness Characterization Methods (HCM)**, having the following input-output paradigm, where methods differ based on their scoring mechanism:

■ **Inputs:** (i) Dataset $\mathcal{D} = \{(x_i, y_i)\}$ drawn from both $\mathcal{P}_{XY}$. (ii) Learning algorithm $f_\theta$.
■ **Outputs:** Assign a score $s_i$ to sample $(x_i, y_i)$. Apply threshold $\tau$ to assign a hardness label $g \in \mathcal{G}$, where $\mathcal{G} = \{Easy, Hard\}$, which is then used to partition $\mathcal{D} = \mathcal{D}_e \cup \mathcal{D}_h$.

We group the HCMs into broad classes (see Table 1) determined based on the core metric or approach each method uses to characterize example hardness, namely (1) Learning dynamics-based: relying on metrics computed during the training process itself to characterize example hardness; (2) Distance-based: using the distance or similarity of examples in an embedding space; (3) Statistical-based: using statistical metrics computed over the data to characterize example hardness.

## 2.1 TAXONOMY OF HARDNESS

**Hardness taxonomy formalism.** The term "hardness" is broad and can manifest in various ways, as demonstrated by the different types of hardness previously examined (see Table 1). Therefore, we must first formalize a taxonomy representative of the different types of hardness we might expect in practice. Each type of hardness is characterized by a latent or unseen hardness perturbation function $h$ that creates hard samples $\mathcal{D}_h$ from samples in $\mathcal{D}$. We denote hardness perturbations on $X$ as $X \rightarrow X^*$ and $Y$ as $Y \rightarrow Y^*$. The effect of each hardness perturbation is explained in terms of the relationship between the joint probability distributions $P^e_{XY}(x, y)$ and $P^h_{XY}(x, y)$.

The taxonomy deals with three broad types (and corresponding subtypes) of hardness: (1) Mislabeling, (2) OoD/Outlier, and (3) Atypical, as illustrated in Figure 1. We anchor the different hardness manifestations with respect to relevant literature for each subtype. We define the various types next.

**Mislabeling:** Samples where the true label is replaced with an incorrect label, such that sample $(x, y) \rightarrow (x, y^*)$. The main distinction between easy and hard samples lies in the label space, leading to different conditional probability distributions: $P^e_{Y|X}(y|x) \neq P^h_{Y|X}(y|x)$. Note, the marginal probability distributions are the same $P^e_X(x) = P^h_X(x)$.

We consider three subtypes of mislabeling: (i) Uniform, (ii) Asymmetric, and (iii) Instance. The HCM literature primarily focuses on evaluation with a *uniform* noise model (Paul et al., 2021; Swayamdipta et al., 2020; Pleiss et al., 2020; Toneva et al., 2019; Maini et al., 2022; Jiang et al., 2021; Mindermann et al., 2022; Baldock et al., 2021), with equal probability of mislabeling across classes. However, in reality, mislabeling is often *asymmetric*, where mislabeling is label-dependent (Northcutt et al., 2021a; Sukhbaatar et al., 2015) or *instance-specific*, where certain mislabeling is more likely given the sample (Jia et al., 2022; Han et al., 2020; Hendrycks et al., 2018; Song et al., 2022). For example, mislabeling an image of a car as a truck is more likely than mislabeling a car as a dog.

Formally, the subtypes differ in their noise models to perturb the true labels, defined by the probabilities $P(Y^* = j | Y = i)$, where $i$ and $j$ are the true and perturbed labels, respectively.

- *Uniform:*    $P(Y^* = j | Y = i) = 1/k-1$      $\forall i, j \in [k], i \neq j$ $\left.\right\}$ Instance Independent
- *Asymmetric:*  $P(Y^* = j | Y = i) = p_{ij}$        $\forall i, j \in [k], i \neq j$
- *Instance:*     $P(Y^* = j | Y = i, X = x) = p_{ij}(x)$  $\forall i, j \in [k], i \neq j$ $\left.\right\}$ Instance Dependent

Table 1: HCMs, even in similar classes, are often (1) *not* quantitatively evaluated and (2) assess different hardness types. The tick and cross denote if quantitative and in brackets qualitative evaluation was performed, e.g. ✗(✔) - *not* quantitative (qualitative). For HCM descriptions, see Appendix A.

| HCM Class | HCM Name | Mislabeling | | | OoD/Outlier | | Atypical |
| --- | --- | --- | --- | --- | --- | --- | --- |
| | | Uniform | Asymmetric | Instance | Near OoD | Far OoD | Long-tail |
| Learning-based (Margin) | AUM (Pleiss et al., 2020) | ✔(✔) | ✔(✔) | ✗(✗) | ✗(✔) | ✗(✗) | ✗(✗) |
| Learning-based (Uncertainty) | Data Maps (Swayamdipta et al., 2020) | ✗(✔) | ✗(✗) | ✗(✗) | ✗(✗) | ✗(✗) | ✗(✗) |
| | Data-IQ (Seedat et al., 2022a) | ✗(✔) | ✗(✗) | ✗(✗) | ✗(✗) | ✗(✗) | ✗(✔) |
| Learning-based (Loss) | Small-Loss (Xia et al., 2021) | ✗(✔) | ✗(✔) | ✗(✔) | ✗(✗) | ✗(✗) | ✗(✔) |
| | Action scores (Arriaga et al., 2023) | ✔(✔) | ✗(✗) | ✗(✗) | ✗(✗) | ✗(✗) | ✗(✔) |
| | RHO-Loss (Mindermann et al., 2022) | ✗(✔) | ✗(✗) | ✗(✗) | ✗(✗) | ✗(✗) | ✗(✔) |
| Learning-based (Gradient) | GraNd (Paul et al., 2021) | ✗(✔) | ✗(✗) | ✗(✗) | ✗(✔) | ✗(✗) | ✗(✔) |
| | VoG (Agarwal et al., 2022) | ✗(✗) | ✗(✗) | ✗(✗) | ✔(✔) | ✗(✗) | ✗(✔) |
| Learning-based (Forgetting) | Forgetting Scores (Toneva et al., 2019) | ✗(✔) | ✗(✗) | ✗(✗) | ✗(✔) | ✗(✗) | ✗(✔) |
| | SSFT (Maini et al., 2022) | ✔(✔) | ✗(✗) | ✗(✗) | ✗(✗) | ✗(✗) | ✗(✔) |
| Learning-based (Statistics) | Detector (Jia et al., 2022) | ✔(✔) | ✗(✗) | ✗(✗) | ✗(✗) | ✗(✗) | ✗(✗) |
| | EL2N (Paul et al., 2021) | ✗(✔) | ✗(✗) | ✗(✗) | ✗(✔) | ✗(✗) | ✗(✔) |
| Distance-based | Prototypicality (Sorscher et al., 2022) | ✗(✗) | ✗(✗) | ✗(✗) | ✗(✔) | ✗(✗) | ✗(✗) |
| Information theory | PVI (Ethayarajh et al., 2022) | ✗(✔) | ✗(✗) | ✗(✗) | ✗(✗) | ✗(✗) | ✗(✗) |
| Statistical-based | Cleanlab (Northcutt et al., 2021a) | ✗(✗) | ✔(✔) | ✔(✔) | ✗(✗) | ✗(✗) | ✗(✗) |
| | ALLSH (Zhang et al., 2022) | ✗(✗) | ✗(✗) | ✗(✗) | ✗(✗) | ✗(✗) | ✗(✗) |
| | Agreement (Carlini et al., 2019) | ✗(✗) | ✗(✗) | ✗(✗) | ✗(✔) | ✗(✗) | ✗(✔) |
| | Data Shapley (Ghorbani & Zou, 2019) | ✔(✔) | ✗(✗) | ✗(✗) | ✔(✔) | ✗(✗) | ✗(✗) |

**OoD/Outlier:** Samples where the covariates undergo a transformation/shift, such that sample $(x, y) \rightarrow (x^*, y)$. The distinction between easy and hard samples lies in the feature space, leading to different marginal probability distributions $P_X^e(x) \neq P_X^h(x)$. Further, for any subset $S$ within the support of $P_X$, $P_X^h(S) = 0$. Note, conditional probability distributions remain consistent where $P_{Y|X}^e(y|x) = P_{Y|X}^h(y|x)$. We consider two subtypes differing in degree of shift, for clarity denoted by an arbitrary distance measure between distributions $dist(\cdot, \cdot)$.

- *Near OoD* (Anirudh & Thiagarajan, 2023; Mu & Gilmer, 2019; Hendrycks & Dietterich, 2018; Sun et al., 2023b; Yang et al., 2023; Tian et al., 2021): samples which have their features transformed or shifted such that they remain proximal to the original samples in $\mathcal{D}$, e.g, introducing noise, pixelating an image, or adding subtle texture changes. In this case, $dist(P_X^h, P_X)$ is positive but relatively small, indicating the nearness of the perturbed distribution to the original. We can represent this bounded distance such that $0 < dist(P_X^h, P_X) \leq \epsilon$, with $\epsilon > 0$.

- *Far OoD* (Mukhoti et al., 2022; Winkens et al., 2020; Graham et al., 2022; Yang et al., 2023): samples which are distinct and likely unrelated to samples in $\mathcal{D}$, often not belonging to the same data-generating process. This could be by sampling from a different dataset or a perturbation s.t. $dist(P_X^h, P_X)$ is significantly large, i.e. $dist(P_X^h, P_X) \gg \epsilon$. For example, for a dataset of digits, images of dogs or cats are distinctly different and unrelated. They are not just rare occurrences but images of dogs or cats represent a different data generation process compared to the digits.

**Atypical:** Samples that, although rare, are still valid instances deviating from common patterns (Yuksekgonul et al., 2023). Atypical samples are inherently part of the primary data distribution, but located in its long tail or less frequent regions (Feldman, 2020; Hooker et al., 2019; Hooker, 2021; Agarwal et al., 2022). The distinction between easy and hard samples leads to different marginal probability distributions $P_X^e(x) \neq P_X^h(x)$, where $P_X^h(x)$ is very small, highlighting their rarity or infrequency. Here, for any subset $S$ within the support of $P_X$, $P_X^h(S) > 0$. This signifies that the long-tail samples, though rarer in occurrence, are still within the bounds of the primary data-generating process. For example, these could be images with atypical variations or vantage points compared to the standard pattern (Agarwal et al., 2022).

**Contrasting OoD/Outlier and Atypical.** Both have different marginal probability distributions $P_X^e(x) \neq P_X^h(x)$. The difference is that OoD/Outliers come from a shifted or completely different distribution than the original data, falling outside the support of $P_X$. In contrast, atypical samples are rare samples from the tails and could naturally arise, falling *within* the support of $P_X$.

From a practitioner's standpoint, these distinctions can dictate different courses of action. OoD/outliers represent likely anomalies or errors that we should detect for potential sculpting or filtering from the dataset. In contrast, atypical samples are rare cases deviating from the "norm", with no or limited similar examples. They are still valid points and should not necessarily be discarded; in fact, there might be a need to gather more of such samples. The goal of surfacing atypical samples is both for dataset auditing and understanding edge cases. We provide additional example images to provide further intuition of the difference between OoD and Atypical in Appendix A, Fig. 3.

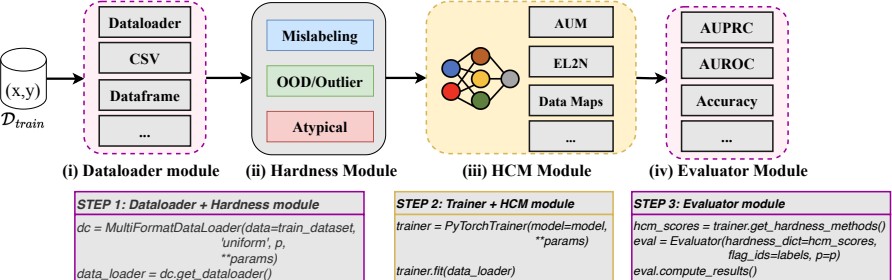

Figure 2: H-CAT facilitates comprehensive benchmarking of HCMs for multiple hardness types. Examples of the single-chain API workflow is shown below — with example usage in Appendix C. The modules described in Sec. 3 are easily extended to new HCMs, datasets, or evaluation metrics

## 2.2 GAPS AND LIMITATIONS OF CURRENT HCMS WITHIN OUR TAXONOMY

To provide a more comprehensive perspective, we critically analyze various HCMs within our proposed taxonomy. Alarmingly, we find inconsistent and diverse definitions of "hardness", even within the same class of methods (refer to Table 1). Such discrepancies, as evidenced by their experimental evaluations, point to a challenge: HCMs, despite appearing to measure similar constructs, in fact, evaluate different "hardness" dimensions. Table 1 highlights two critical deficiencies discussed below, underscoring the urgent need for a systematic evaluation framework that accurately captures the scope and applicability of each HCM across different hardness types.

**Issue 1: Qualitative or indirect measures.** Many HCMs limit their evaluation to (i) *qualitative* analyses: merely showcasing flagged samples, or (ii) *indirect measures*: for example, showing downstream performance improved when removing flagged samples without directly quantifying what the HCM captures. This overlooks the necessity for an objective, quantitative evaluation.

**Issue 2: Narrow and unrepresentative scope.** HCMs that conduct quantitative evaluation focus on a single hardness type and often target the simplest manifestation. An example of this shortfall is seen in the handling of mislabeling. Many HCMs only focus on uniform mislabeling, thereby failing to account for the more realistic and complex scenarios of asymmetric or instance-wise mislabeling. Beyond this, many HCMs only test on mislabeling, overlooking other types of hardness.

These limitations emphasize the value of our fine-grained taxonomy in categorizing and evaluating various hardness types. By laying a solid foundation for the systematic evaluation and comparison of HCMs, our taxonomy enables the design and selection of HCMs that are tailored to specific hardness challenges, thereby promoting the development of more robust ML systems.

## 3 H-CAT: A BENCHMARKING FRAMEWORK FOR HCMS

To address the aforementioned limitations and facilitate benchmarking of HCMs, we propose the Hardness Characterization Analysis Toolkit (H-CAT). H-CAT serves two purposes: *(1) empirical benchmarking standard*: supporting comprehensive and quantitative benchmarking of HCMs on different hardness types within the taxonomy across multiple data modalities and *(2) software toolkit*: H-CAT unifies multiple HCMs under a single unified interface for easy usage by practitioners.

**H-CAT Design.** H-CAT has four core modules as described below, which are called sequentially (Figure 2). The framework follows widely adopted objected-oriented paradigms with fit-predict interfaces (here, update-score). The workflow is simple with single-chain API calls. The stepwise composability aims to facilitate easy benchmarking and allows H-CAT to be used outside of benchmarking as a data characterization tool by practitioners.

- *Dataloader module*: loads a variety of data types for ease of use, including Torch datasets, NumPy arrays, and Pandas DataFrames, allowing users to easily use H-CAT with their chosen datasets.
- *Hardness module*: generates controllable hardness for different hardness types in the taxonomy.
- *HCM module*: provides a unified HCM interface with 13 HCMs implemented. It wraps the trainer module which is a conventional PyTorch training loop.
- *Evaluator module*: computes the HCM evaluation metrics to correctly identify the ground-truth hard samples using the scores provided by each HCM. User-specified metrics can easily be included by operating on the raw HCM scores.

**Extensibility.** H-CAT is easily extendable to include new HCMs, hardness types, or datasets by defining a simple wrapper class. For details and step-by-step code examples, refer to Appendix C.

## 4 BENCHMARKING FRAMEWORK SETUP

We now describe three key aspects of the benchmarking setup — the implementation of the Hardness Module, HCM Module and Evaluator Module.

### 4.1 HARDNESS MODULE

We describe the implementation of our hardness perturbations $h$. We focus on image data here, as most HCMs (10/13) have been developed for this modality. However, we discuss the corresponding hardness perturbations for tabular data in the Appendix.

**Mislabeling:** (i) *Uniform:* random mislabeling, drawn uniformly from all possible class labels. (ii) *Asymmetric:* random mislabeling, drawn asymmetrically via a Dirichlet distribution (Zhang et al., 2021; Bae et al., 2022; Zhu et al., 2022). We denote a special case of asymmetric applicable to ordinal labels, namely *adjacent*. Here, the mislabeling is to the nearest numerical class, e.g. an MNIST digit 3 mislabeled as 2 or 4. (iii) *Instance:* mislabeling probability is conditioned on an instance (reflecting human mislabeling). This can often be determined by domain/user knowledge, e.g. for MNIST, 1 is likely mislabeled as 7; for CIFAR-10, an automobile could be mislabeled as a truck.

**OoD/Outlier:** (i) *Near OoD:* pertubed data is different but related. *Covariate Shift* via Gaussian noise as performed in MNIST-C (Mu & Gilmer, 2019) or *Domain Shift:* image texture from the original photographic image is changed via edge detection and smoothing (Median filter). (ii) *Far OoD:* perturbed data is distinctly different and unrelated. We replace a subset of data with unrelated data, e.g. for MNIST replace with CIFAR-10 images and vice versa.

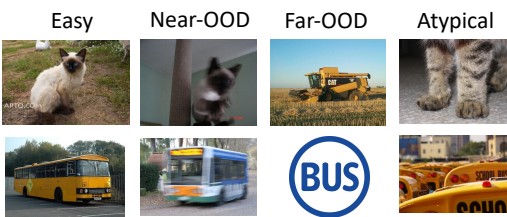

Figure 3: Examples providing intuition on the difference between OoD and Atypical.

**Atypical:** (i) *Shift*: translate and shift the image, causing portions to be cut off in an atypical manner. (ii) *Zoom*: create an atypical perspective by magnifying (X2) features usually seen at a smaller scale.

### 4.2 HCM MODULE

We include 13 widely used HCMs applicable to supervised classification under a unified interface. The HCMs span the range of HCM classes, at least one per class from Table 1: ■ **Learning-based**[1] **(Uncertainty):** Data Maps (Swayamdipta et al., 2020) and Data-IQ (Seedat et al., 2022a); ■ **Learning-based (Loss):** Sample-Loss (Xia et al., 2021; Arriaga et al., 2023) ; ■ **Learning-based (Margin):** Area-under-the-margin (AUM) (Pleiss et al., 2020); ■ **Learning-based (Gradient):** GraNd (Paul et al., 2021) and VoG (Agarwal et al., 2022); ■ **Learning-based (Statistics):** EL2N (Paul et al., 2021) and Noise detector (Jia et al., 2022), ■ **Learning-based (Forgetting):** Forgetting scores (Toneva et al., 2019); ■ **Statistical measures:** Cleanlab (Northcutt et al., 2021a), ALLSH (Zhang et al., 2022), Agreement (Carlini et al., 2019); ■ **Distance-based:** Prototypicality (Sorscher et al., 2022).

Specifically, we focus on HCMs that plug into the training loop and *do not*: (i) alter training, (ii) require repeated training, (iii) need additional datasets beyond the training set, or (iv) require training additional models. Consequently, we exclude the following: RHO-Loss (Mindermann et al., 2022) requires an additional irreducible loss model, SSFT (Maini et al., 2022) requires fine-tuning on a validation dataset, PVI (Ethayarajh et al., 2022) requires training a Null model. We also do not consider Data Shapley (Ghorbani & Zou, 2019) and variants (e.g. Beta Shapley (Kwon & Zou, 2022)), which have been shown to be computationally infeasible with numerical instabilities for higher dimensional data such as MNIST and CIFAR-10 with > 1000 samples (Wang & Jia, 2023).

### 4.3 EVALUATOR MODULE

We directly assess the HCM's capability to detect the hard samples. Recall that HCMs assign a score $s$ to each sample $(x, y)$ and then apply a threshold $\tau$ to assign samples a group $g \in \mathcal{G}$, where $\mathcal{G} = \{Easy, Hard\}$. Many HCMs do not explicitly state how to define $\tau$; hence to account for this we compute two widely used metrics: *AUPRC* (Area Under Precision-Recall Curve) and *AUROC* (Area Under Receiver Operating Curve) — for hard sample detection performance, which we denote D-AUPRC and D-AUROC [2]. User-specified metrics are easily computed on raw HCM scores.

---

[1] Learning-based generally refers to learning/training dynamics based HCMs

[2] to distinguish them from the typical downstream performance metrics.

## 5 COMPREHENSIVE EVALUATION OF HCMs USING H-CAT

We evaluate 13 different HCMs (spanning a range of techniques) across 8 distinct hardness types. To the best of our knowledge, this represents the first comprehensive HCM evaluation, encompassing over **14K** experimental setups (specific combination of HCM, hardness type, perturbation proportion, dataset, model, and seed).

We primarily focus on image datasets, as this is the modality for which the majority of the HCMs (10/13) have been developed. We use the MNIST (LeCun et al., 2010) and CIFAR-10 (Krizhevsky et al.) datasets, as their use is well-established in the HCM literature (Paul et al., 2021; Swayamdipta et al., 2020; Pleiss et al., 2020; Toneva et al., 2019; Maini et al., 2022; Jiang et al., 2021; Mindermann et al., 2022; Baldock et al., 2021). Importantly, they are realistic images yet contain almost no/little mislabeling (<0.5%) (Northcutt et al., 2021b). This contrasts other common image datasets like ImageNet that contain significant mislabeling (over 5%), hence we cannot perform controlled experiments. Furthermore, to show generalizability across modalities, we also evaluate on tabular datasets "Covertype" and "Diabetes130US" benchmark datasets (Grinsztajn et al., 2022) from OpenML (Vanschoren et al., 2014) (see Appendix D.6).

To assess HCM sensitivity to the backbone model, we use two different models for our image experiments with different degrees of parameterization, LeNet and ResNet-18. All experiments are repeated with 3 random seeds and for varying proportions $p$ of hard samples.

We present aggregated results in Figs. 4-7, with more granular results in Appendix D — along with additional experiments. The main paper shows results for 6 out of 8 hardness types. We include results for other sub-types not covered in the main paper in Appendix D including: Domain shift (a type of Near OoD), Zoom shift (a type of Atypical), and Adjacent (a special case of Asymmetric mislabeling), offering similar conclusions. We investigate three aspects of HCMs (**A-C**), distilling the results into **benchmarking takeaways** and **practical tips**.

**A. Hardness detection performance.** Directly evaluate HCM capabilities to detect the hard samples for different hardness types, for varying perturbation proportions $p$ – see Fig. 4.

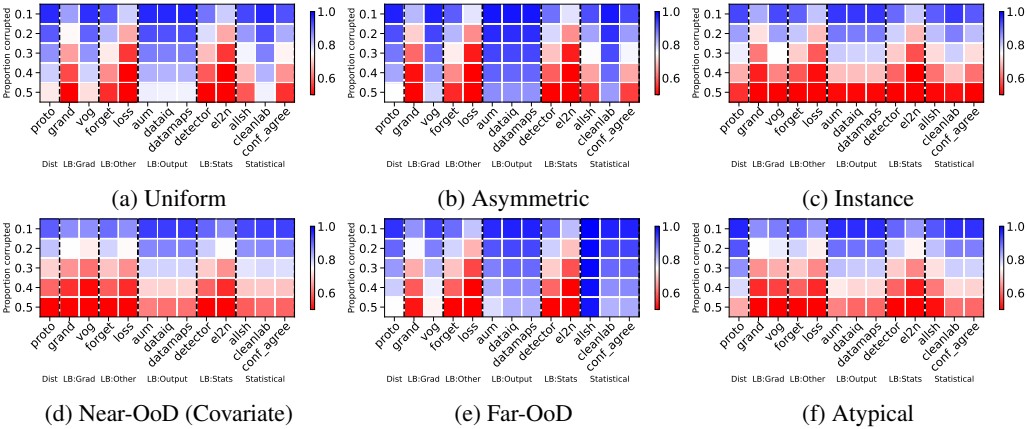

Figure 4: D-AUPRC for different HCMs for different hardness types aggregated across setups. We vary the proportion perturbed, i.e. the proportion of hard examples. Blue is better, red worse. We see variability of HCM capabilities across hardness types and proportions.

**Takeaway A1: Comprehensive testing is vital.** Many HCMs are assessed on a single hardness type. The reality is hardness manifests in many ways. We show HCM performance varies across hardness types and across proportions of hardness, with some more challenging than others, e.g. instance is harder than uniform. This result shows the critical need for comprehensive HCM evaluation.

**Takeaway A2: Hardness types vary in difficulty.** We find that different types of hardness are easier or harder to characterize. For instance, uniform mislabeling or Far-OoD are much easier than data-specific hardness like instance and Atypical. Given the performance differences of HCMs on different hardness types, it becomes important to understand the hardness type expected in practice.

**Takeaway A3:** **Learning dynamics-based methods with respect to output confidence are effective general-purpose HCMs.** In selecting a general-purpose HCM, we find that HCMs that characterize samples using learning dynamics on the confidence — uncertainty-based methods, which use probabilities (DataMaps, Data-IQ) or logits (AUM), are the best performing in terms of AUPRC across the board.

**Takeaway A4:** **HCMs typically used for computational efficiency are surprisingly uncompetitive.** We find that HCMs that leverage gradient changes (e.g. GraNd), typically used for selection to improve computational efficiency, fare well at low $p$. However, at higher $p$, they become notably less competitive compared to simpler and computationally cheaper methods.

**Practical Tip A1:** **HCMs should only be used when hardness proportions are low.** In general, different HCMs have significantly reduced performance at higher proportions of hardness. This is expected as we get closer to 0.5 since it's harder to identify a clear difference between samples.

**B. Rankings and Statistical Significance.** Compare the ranking of methods, as well as assess statistical significance of performance differences using critical difference diagrams (CD diagram) (Demšar, 2006) based on the Siegel-Friedman method (p ≤ 0.05) — see Figs. 5 and 6.

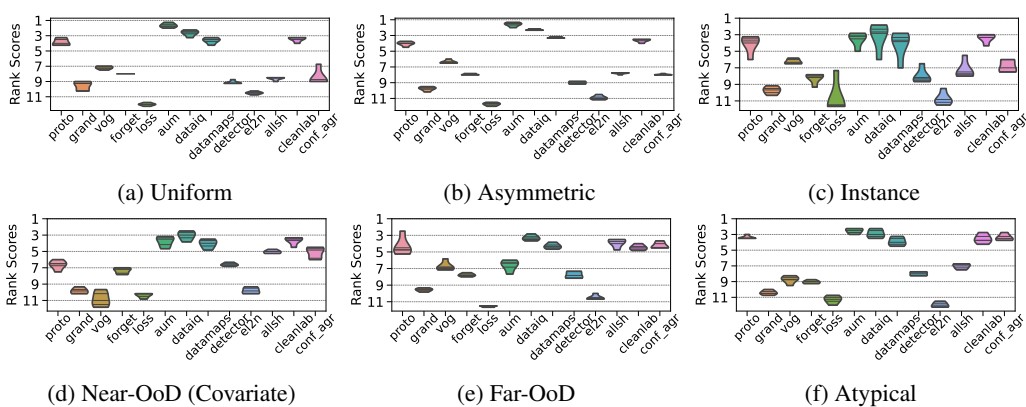

Figure 5: Performance rankings of HCMs vary depending on the hardness type.

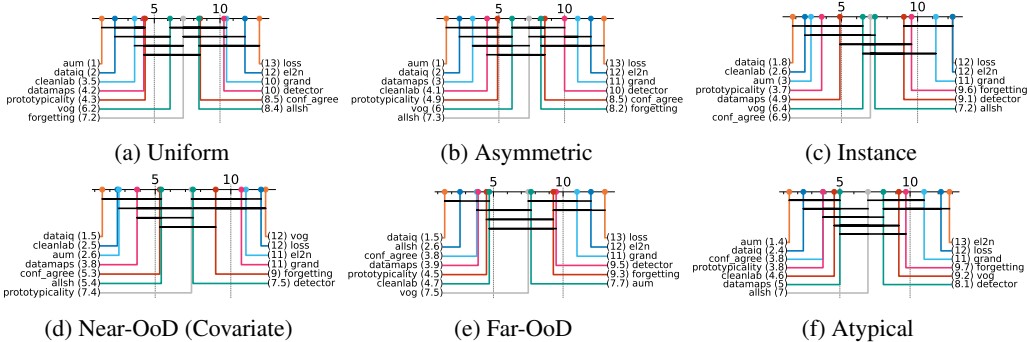

Figure 6: Critical difference diagrams highlight that similar categories/classes of HCMs *do not* have a statistically significant difference in their performance, indicated by the horizontal black lines linking HCMs which are not statistically different. The numbers in brackets denote mean rank.

**Takeaway B1:** **Individual HCMs within a broad "class" of methods are NOT statistically different.** We find from the critical difference diagrams that methods falling into the same class of characterization are not statistically significant from one another (based on the black connected lines), despite the minor performance differences between them. Hence, practitioners should select an HCM within the broad HCM class most suitable for the application.

**Practical Tip B1:** **Selecting an HCM based on the hardness is useful.** We find that confidence is a good general-purpose tool if one does not know the type of hardness. However, if one knows the hardness, one can better select the HCM. For example, Prototypicality, as expected, is very strong on instance hardness as we are able to match samples via similarity of classes in embedding space.

**Practical Tip B2:** **Divergence and distance-based methods are suitable primarily for distributional changes.** Divergence and distance-based methods such as ALLSH and Prototypicality should primarily be used if the hardness is with respect to a shift of the data itself rather than mislabeling.

**C. Stability/Consistency.** The rank ordering of samples is important in data characterization (Maini et al., 2022; Wang & Jia, 2023; Seedat et al., 2022a). Hence, we desire HCM scores to be stable to ensure consistent insights. As is standard (Maini et al., 2022; Seedat et al., 2022a), we compute the Spearman rank correlation across multiple runs — see Fig. 7. We also assess HCM stability/consistency across backbone models and parameterizations in Appendix D.

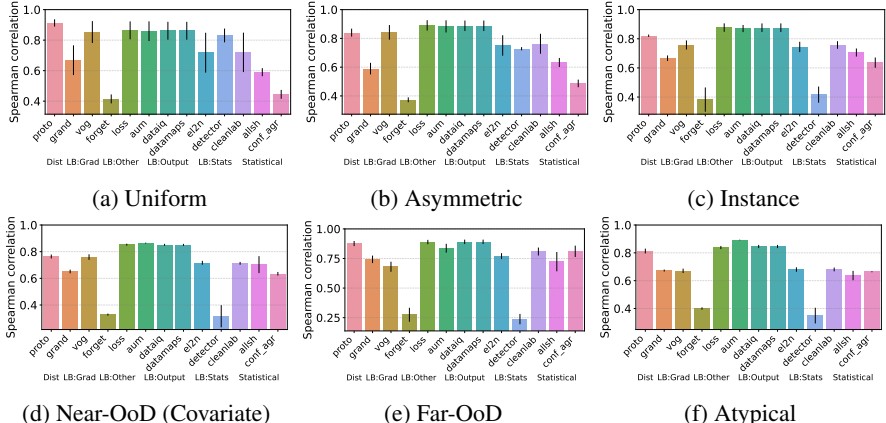

Figure 7: Certain classes of HCMs are more stable and consistent than others (retaining rank order), with higher Spearman rank correlations for multiple runs of the HCM on the same data

**Takeaway C1:** **Learning dynamics-based HCMs using output metrics or distance-based HCMs are most stable and consistent.** We find across all hardness types that the Spearman rank correlation is highest for HCMs that use learning dynamics on the outputs or use distance measures — specifically Uncertainty (Probabilities), Margins (Logits), Loss, or Prototypicality. The low correlation for other HCMs highlights sensitivity to the run itself when characterizing data, which would lead to inconsistent ordering.

**Takeaway C2:** **Insights consistent across backbone models & parameterizations.** As shown in Appendix D, we find similar results (as above) for the Spearman rank correlation of HCM scores across different backbone models and parameterizations. As the scores are consistent, this indicates that the findings and insights computed on those scores will also be consistent, i.e. those HCMs which were the most stable and consistent remain the most stable and consistent.

**Practical Tip C1:** **Select a stable HCM.** Certain HCMs are more sensitive to randomness. Hence, it is advised to select stable and consistent HCMs (higher Spearman correlation) to avoid such effects.

## 6 DISCUSSION

We introduce H-CAT, a comprehensive benchmarking framework for hardness characterization in data-centric AI. We analyzed 13 HCMs spanning a range of techniques for a variety of setups - over **14K**. We hope our framework and insights addressing calls for rigorous benchmarking (Guyon, 2022) and understanding of existing ML methods (Lipton & Steinhardt, 2019; Snoek et al., 2018) will spur advancements in data-centric AI and that H-CAT helps practitioners better evaluate and use HCMs.

**Limitations & Future Work.** No benchmark can exhaustively test all the possible hardness manifestations and this work is no different. However, we cover multiple instances from the three fundamental types of hardness, which is significantly broader than any prior work. Building on this work, future research could investigate cases where multiple types of "hardness" manifest simultaneously (e.g. Near-OoD and mislabeling together) or where hardness is continuous rather than binary. To spur this, we provide an example of simultaneous hardness in Appendix D. From a usage perspective, future work could also look into the best way hardness scores could be used to guide better model training (e.g. data curriculum). Finally, we highlight that HCMs, and by extension H-CAT, cannot tell you which hardness type exists in a dataset; rather, H-CAT serves to benchmark the capabilities of HCMs or as a unified HCM interface.

## ACKNOWLEDGEMENTS

The authors would like to thank Nicolas Huynh, Alicia Curth and the anonymous reviewers for their helpful comments and discussions on earlier drafts of this paper and Robert Davis for discussions on the code. NS and FI gratefully acknowledge funding from the Cystic Fibrosis Trust and NSF grant (1722516), respectively. This work was supported by Azure sponsorship credits granted by Microsoft's AI for Good Research Lab.

## ETHICS AND REPRODUCIBILITY STATEMENTS

**Ethics Statement.** HCMs that accurately identify hard samples can help make models more robust and reliable. This paper highlights the importance of rigorously evaluating HCMs to better understand their capabilities and guide better usage. This paper aims to enable the community to conduct a more systematic hardness characterization through the proposed taxonomy and benchmarking framework.

**Reproducibility Statement.** We include implementation details about our benchmark in Sec. 3 and 4, as well as in Appendix B. The code for the H-CAT framework can be found at `https://github.com/seedatnabeel/H-CAT` or `https://github.com/vanderschaarlab/H-CAT`. Step-by-step code examples can be found in the repository and in Appendix C along with a guide on how to extend H-CAT to new HCMs, hardness types, and datasets.

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

# Appendix: Dissecting Sample Hardness: A Fine-Grained Analysis of Hardness Characterization Methods for Data-Centric AI

## Table of Contents

# A  ADDITIONAL BACKGROUND AND DETAILS

This appendix provides additional details and background on the different HCMs.

## A.1  DESCRIPTION OF HCMS

We now provide a detailed description of the HCMs supported by H-CAT grouped by their class. We briefly describe how they characterize data and the modalities to which the HCMs have been previously applied.

We group the HCMs into broad classes determined based on the core metric or approach each method uses to characterize example hardness, namely

- Learning dynamics-based: These HCMs rely on metrics computed during the training process itself to characterize example hardness. As we describe in the paper, we further divide these into sub-categories based on the specific training dynamic metric used: (i) Margin, (ii) Uncertainty, (iii) Loss, (iv) Gradient, (v) Forgetting.
- Distance-based: These HCMs characterize hardness based on the distance or similarity of examples in an embedding space.
- Statistical-based: These methods use statistical metrics computed over the data to characterize example hardness.

Recall HCMs generally follow the following input-output paradigm:

■ **Inputs:** (i) Dataset $\mathcal{D} = \{(x_i, y_i)\}$ drawn from both $\mathcal{P}_{XY}$. (ii) Learning algorithm $f_\theta$.
■ **Outputs:** Assign a score $s_i$ to sample $(x_i, y_i)$. Apply threshold $\tau$ to assign a hardness label $g \in \mathcal{G}$, where $\mathcal{G} = \{Easy, Hard\}$, which is then used to partition $\mathcal{D} = \mathcal{D}_e \cup \mathcal{D}_h$.

Consequently, we define the scoring function $S$ as the general notation applicable to each HCM. Note that whether a high score or low score indicates a "hard" sample is method-dependent. We highlight HCMs included in H-CAT with a * and formalize those specifically with notation.

We provide a Table below to clarify whether high or low scores mean "hard" for specific HCMs.

Table 2: Overview of HCMs and the meaning of their scoring functions.

| Method | Description | Score |
|---|---|---|
| Area Under the Margin (AUM) | Characterizes data examples based on the margin of a classifier – i.e., the difference between the logit values of the correct class and the next class. | Hard - low scores. |
| Confident Learning | Confident Learning estimates the joint distribution of noisy and true labels — characterizing data as easy and hard for mislabeling. | Hard - low scores |
| Conf Agree | Agreement measures the agreement of predictions on the same example. | Hard - low scores |
| Data IQ | Data-IQ computes the aleatoric uncertainty and confidence to characterize the data into easy, ambiguous, and hard examples. | Hard - low confidence scores. Low Aleatoric Uncertainty scores |
| Data Maps | Data Maps focuses on measuring variability (epistemic uncertainty) and confidence to characterize the data into easy, ambiguous, and hard examples. | Hard - low confidence scores. Low Epistemic Uncertainty scores |
| Gradient Norm (GraNd) | GraNd measures the gradient norm to characterize data. | Hard - high scores |
| Error L2-Norm (EL2N) | EL2N calculates the L2 norm of error over training in order to characterize data for computational purposes. | Hard - high scores |
| Forgetting | Forgetting scores analyze example transitions through training. i.e., the time a sample correctly learned at one epoch is then forgotten. | Hard - high scores |
| Large Loss | Large Loss characterizes data based on sample-level loss magnitudes. | Hard - high scores |
| Prototypicalilty | Prototypicality calculates the latent space clustering distance of the sample to the class centroid as the metric to characterize data. | Hard - high scores |
| Variance of Gradients (VOG) | VoG (Variance of gradients) estimates the variance of gradients for each sample over training | Hard - high scores |
| ALLSH | ALLSH computes the KL divergence of softmax outputs between original and augmented samples to characterize data. | Hard - high scores |

We now outline different HCMs. Specifically, we formalize the HCMs included in H-CAT in a unified notation of $S(x, y)$.

### A.1.1 LEARNING-BASED (MARGIN)

**AUM (Pleiss et al., 2020)**[*]  AUM (Area under the margin) characterizes data examples based on the margin of a classifier – i.e. the difference between the logit values of the correct class and the next class. It has been applied to mislabeling for computer vision (CV).

$$M^{(t)}(x, y) = \overbrace{z_y^{(t)}(x)}^{\text{assigned logit}} - \overbrace{\max_{i \neq y} z_i^{(t)}(x)}^{\text{largest other logit}}. \tag{1}$$

Over all epochs the AUM is then computed as the average of these margin calculations. We define it in terms of $S$.

$$S(x, y) = AUM(x, y) = \frac{1}{T} \sum_{t=1}^{T} M^{(t)}(x, y), \tag{2}$$

$$S(x, y) = \frac{1}{T} \sum_{t=1}^{T} ( \overbrace{z_y^{(t)}(x)}^{\text{assigned logit}} - \overbrace{\max_{i \neq y} z_i^{(t)}(x)}^{\text{largest other logit}}) \tag{3}$$

### A.1.2 LEARNING-BASED (UNCERTAINTY)

**Data-IQ (Seedat et al., 2022a)**[*]  Data-IQ computes the aleatoric uncertainty and confidence to characterize the data into easy, ambiguous, and hard examples and has been studied for the tabular domain.

The aleatoric uncertainty is computed as $v_{\text{al}}(x) = \frac{1}{T} \sum_{t=1}^{T} \mathcal{P}(x, \theta_t)(1 - \mathcal{P}(x, \theta_t))$ and confidence over epochs as $\mathcal{P}(x, \theta_t)$.

For hard samples, the score for an input (x,y) is computed primarily on the confidence with uncertainty used to delineate ambiguous:

$$S(x, y) = \frac{1}{T} \sum_{t=1}^{T} \mathcal{P}(x, \theta_t) \tag{4}$$

$$C(x, y) = \frac{1}{T} \sum_{t=1}^{T} \mathcal{P}(x, \theta_t) \tag{5}$$

**Data Maps (Swayamdipta et al., 2020)**[*]  Data Maps focuses on measuring variability (epistemic uncertainty) and confidence to characterize the data into easy, ambiguous, and hard examples and has been studied for natural language processing (NLP) tasks.

The epistemic uncertainty is computed as $v_{\text{ep}}(x) = \frac{1}{T} \sum_{t=1}^{T} \left[ \mathcal{P}(x, \theta_t) - \bar{\mathcal{P}}(x) \right]^2$ and confidence over epochs as $\mathcal{P}(x, \theta_t)$.

For hard samples, the score for an input (x,y) is computed primarily on the confidence with uncertainty used to delineate ambiguous:

$$S(x, y) = \mathcal{P}(x, \theta_e) \tag{6}$$

### A.1.3 LEARNING-BASED (LOSS)

**Small Loss (Xia et al., 2021)**[*]  Small Loss characterized data based on sample-level loss magnitudes and has been studied for computer vision (CV) tasks.

The small loss for an input $x, y$ is computed as:

$$S(x, y) = \frac{1}{t} \sum_{i=1}^{t} l_i, \tag{7}$$

where   $l_i = \ell(f(w; x), y)$.

**RHO-Loss (Mindermann et al., 2022)**   RHO-Loss estimates the sample loss on a holdout set and has been applied in CV tasks. We do not include it in H-CAT as it requires training an irreducible loss model.

### A.1.4 LEARNING-BASED (GRADIENT)

**GraNd (Paul et al., 2021)**[*]   GraNd measures the gradient norm to characterize data, particularly for computational reasons in computer vision (CV) tasks.

The gradient norm at epoch $t$ for an input $x, y$ is computed as :

$$S(x,y) = \mathbb{E} \left\| \sum_{k=1}^{K} \boldsymbol{\nabla}_{f^{(k)}} \ell\left(f_t(x), y\right)^T \psi_t^{(k)}(x) \right\|_2 \ where \psi_t^{(k)}(x) = \boldsymbol{\nabla}_{\mathbf{w}_t} f_t^{(k)}(x). \tag{8}$$

**VoG (Agarwal et al., 2022)**[*]   VoG (Variance of gradients) estimates the variance of gradients for each sample over training, particularly in computer vision (CV) tasks.

The VoG at epoch $t$ for an input $x, y$ is computed as :

$$S(x,y) = \frac{1}{N} \sum_{t=1}^{N} (VoG_p(x,y)), \tag{9}$$

$$VoG_p(x,y) = \sqrt{\frac{1}{K} \sum_{t=1}^{K} (M_t(x,y) - \mu)^2}. \tag{3}$$

where $\mu = \frac{1}{K} \sum_{t=1}^{K} M_t$

Note $M_t$ is the gradient matrix.

### A.1.5 LEARNING-BASED (FORGETTING)

**Forgetting scores (Toneva et al., 2019)**[*]   Forgetting scores analyze example transitions through training. i.e., the time a sample correctly learned at one epoch is then forgotten. It is commonly used in CV tasks.

The Forgetting score an input $x, y$ is computed as :
$$S(x,y) \leftarrow S(x,y) + \mathbf{1}(\text{prev\_acc}_i > \text{acc}(x,y)) \tag{10}$$

**SSFT (Maini et al., 2022)**   SSFT (Second Split Forgetting Time) measures the time of forgetting when fine-tuning models on a second split. It has been applied in CV tasks. We do not include it in H-CAT as it requires fine-tuning on a validation dataset,

### A.1.6 LEARNING-BASED (STATISTICS)

**EL2N (Paul et al., 2021)**[*]   EL2N (L2 norm of error over training) calculates the L2 norm of error over training in order to characterize data for computational purposes. It is predominantly applied in CV tasks.

The EL2N at epoch $t$ for an input $x, y$ is computed as :
$$S(x,y) = \mathbb{E}\|p(w_t, x) - y\|_2. \tag{11}$$

**Noise detector (Jia et al., 2022)**[*]   Noise detector utilizes a trained detector model to predict noisy samples. The detector is applied to training dynamics and has been used in CV tasks.

Noise detector uses a trained detector $g$ to score x,y as:

$$S(x_i, y_i) = g(L_{y_i}(i)) \tag{12}$$

where $L(i,t) = f^{(t)}(x_i)$[training dynamic]

### A.1.7 DISTANCE-BASED

**Prototypicality (Sorscher et al., 2022)[*]**   Prototypicality calculates the latent space clustering distance to the centroid as the metric to characterize data. It has been applied to CV tasks.

Prototypicality score for x,y is computed as:

$$S(x,y) = \|\phi(x) - \mu_y\|_2 \tag{13}$$

Here, $\phi(x)$ represents the mapping of $x$ to the latent space, and $\mu_y$ is the centroid of the class $y$ in the latent space. The norm $\| \cdot \|_2$ denotes the Euclidean distance.

### A.1.8 STATISTICAL-BASED

**Cleanlab (Northcutt et al., 2021a)[*]**   Cleanlab estimates the joint distribution of noisy and true labels — characterizing data as easy and hard for mislabeling. It can be applied to any modality.

The Cleanlab score for an input $x, y$ is computed as :

$$S(x,y) = \{x \in X \mid y = y^* \neq j\} \text{ for } x \text{ from the off-diagonals of } C_{y_j, y^*}. \tag{14}$$

**ALLSH (Zhang et al., 2022)[*]**   ALLSH computes the KL divergence of softmax outputs between original and augmented samples to characterize data.

The Allsh score for an input $x, y$ is computed as :

$$S(x,y) = D(p_\theta(\cdot|x), p_\theta(\cdot|x')) \tag{15}$$

where $x'$ is a augmented $x$ and $D$ a distance measure (e.g KL-Divergence).

**Agreement (Carlini et al., 2019)[*]**   Agreement measures the agreement of predictions on the same example. It is commonly used in CV tasks.

The Agreement for an input $x, y$ is computed as :

$$S(x,y) = \frac{1}{N} \sum_{i=1}^{N} \max f_i(x) \tag{16}$$

**Data Shapley (Ghorbani & Zou, 2019)**   Data Shapley computes the Shapley value to characterize the value of including a data point. It is commonly used in CV tasks. We also do not consider Data Shapley in H-CAT and variants (e.g. Beta Shapley (Kwon & Zou, 2022)), which have been shown to be computationally unfeasible with numerical instabilities for higher dimensional data such as MNIST and CIFAR-10 with > 1000 samples (Wang & Jia, 2023).

### A.2 PRACTICAL APPLICATIONS OF HCMs.

To illustrate the practical value and need for HCMs in creating high-quality datasets, we highlight the following application domains:

- The Cleanlab HCM was used to audit ML benchmark datasets for mislabeling (Northcutt et al., 2021b).
- The Data Maps HCM was used to audit and create high-quality RNA data in biology (Nabi et al., 2023).
- The Data-IQ HCM was used to audit and curate high-quality sleep staging data (Heremans et al., 2024).
- The Data-IQ and Cleanlab HCMs have been used to guide synthetic data generation (Hansen et al., 2023).
- The Data Maps and Data-IQ HCMs where re-purposed to assess samples for data-centric off-policy evaluation (Sun et al., 2023a).

A.3 Hardness taxonomy for tabular data.

In the main paper, we primarily discussed our manifestation of the hardness taxonomy with respect to images. We describe below how this isn't confined to images and how these hardness types could manifest, for instance, in tabular data.

**Mislabeling:** Mislabeling in tabular data can occur when the label attributed to a particular row (representing a sample) is incorrect. For instance, in a medical dataset, this could occur when a patient with a certain condition is wrongly labeled as not having the condition (or vice versa). Note mislabeling is no different from the image domain as the perturbation is applied on $y$.

Hence, the various sub-types of mislabeling can still apply. Uniform mislabeling means any class of disease can be mislabeled as any other with equal probability. Asymmetric mislabeling would mean certain classes of disease are more likely to be confused with others, for instance, certain diseases might be more commonly misdiagnosed as each other. Consequently, in implementation uniform and asymmetric do not change from images. Adjacent mislabeling (special case of asymmetric) could refer to situations where labels are ordinal and a higher value might be mislabeled as a lower one (or vice versa). Instance-specific mislabeling refers to instances where certain individual observations are more likely to be mislabeled based on their specific attributes. We could still use "human knowledge" of similar classes for tabular data. We demonstrate the use of PCA to embed the data and then find the nearest class for a specific label in embedding space as the instance-wise mislabeling.

**Near-OoD:** Near-OoD in tabular data could refer to rows of data where the feature values are still within the same domain but differ in some way from typical observations. For example, in a financial dataset, a Near OoD sample might be an account with an unusually high number of transactions, which, although not completely unusual, stands apart from typical transaction behavior. This could represent a Covariate Shift where the distribution of the predictors changes, or Domain Shift where the rules of data generation change slightly.

Alternatively, in a dataset of regular health screenings, a Near OoD instance might be a health record of a patient who shows high values in their lab results, which although still plausible, stand apart from typical health metrics. For instance, a patient with very high cholesterol levels would be a Near OoD example in this context, indicating a Covariate Shift where the distribution of predictors changes. In terms of implementation, the continuous variables of our tabular data might be noisy or covariate shifted data. For example, Gaussian noise to continuous features.

**Far-OoD:** Far Out-of-Distribution in tabular data refers to rows that are dramatically different from typical observations in a dataset, or might even represent entirely new categories of data. Using the financial dataset example, a Far OoD sample might be transactions in a new foreign currency not previously seen in the dataset. This type of data is distinctly different and unrelated to the rest of the dataset.

In a medical dataset, Far-OoD might refer to patient records that are substantially different from typical observations or that fall into entirely new categories. For instance, imagine a dataset mainly consisting of adult patients, and suddenly you get data from pediatric patients. Children have different medical characteristics and range for many medical metrics (like heart rate, blood pressure, etc.), so pediatric patient data would be Far OoD in this context. Another example could be the inclusion of a new disease class not previously part of the dataset.

Practically, for a given tabular dataset, this could be data from a different distribution (e.g. shuffled data) or feature relationships that don't match — e.g. older people having a specific feature linked to a disease, but these relationships are now broken. For example, within the framework this could be implemented as follows: for binary covariates, we can swap the features (i.e. flip male to female etc.) — to break the feature relationships such that they are drawn from a completely different distribution. For categorical variables, we can flip the category of a subset of variables in order to break the feature relationships.

**Atypical:** Atypical data in a tabular dataset can refer to observations that, while not incorrect or out of distribution, deviate from common patterns in the data, i.e. they are rare but valid rows of the table (e.g. outliers, extremes — samples from tails of marginal). For example, in a dataset of real estate properties, an atypical sample could be a property with an unusually large number of rooms or an

unusually low price for its size. While such a property could exist (so it's not necessarily mislabeled), it deviates from the common pattern in the data and might be found in the long tail of the distribution.

In a medical dataset, it might refer to patient records that, while not incorrect or out of distribution, deviate from common patterns in the data. For instance, consider a dataset of prostate cancer patients, where naturally, most patients are older. An atypical example might be a young adult with prostate cancer, which would represent an unusual instance, as they fall outside the common pattern of older patients with prostate cancer. From a framework implementation perspective, we could sample and replace with values from the tails of the marginals for the most predictive long-tailed feature (via feature-label correlation).

# B    EXPERIMENTAL DETAILS

This appendix describes our experimental details, computational resources, HCMs, and datasets below.

## B.1    COMPUTATIONAL RESOURCES

Experiments are parallelized to run different setups using three NVIDIA RTX A4000s, with an 18-Core Intel Core i9 and 16 gb of RAM. Each experimental setup can be run on a single GPU.

## B.2    DATASETS

We run our experiments on MNIST and CIFAR-10 datasets. However, these experiments can be run on alternative datasets, as shown in Appendix C.

**MNIST (LeCun et al., 2010)**    : consists of 70,000 small square 28x28 pixel grayscale images of digits from 0 to 9. Each image is a handwritten digit, with 60,000 of the images dedicated for training data and the remaining 10,000 for testing data.

**CIFAR-10 (Krizhevsky et al.)**    : It consists of 60,000 32x32 color images distributed among 10 different classes such as airplanes, dogs, cats, and trucks. The dataset is divided into 50,000 training images and 10,000 testing images. Each class has an equal representation in the dataset with 6,000 images per class.

**Note on selection of MNIST & CIFAR-10.**    We chose the MNIST and CIFAR-10 datasets as benchmarks because they are well-established datasets in the literature. Notably, many HCMs employ MNIST and CIFAR-10 within the context of their limited quantitative evaluation. Further, the noisy label literature frequently uses them for controlled experiments. This prevalence underscores their appropriateness for controlled benchmarking.

Let's compare potential alternatives.

- Completely synthetic data: This would be 100% clean, but too simple and toy-like.

- Large and highly complicated datasets like ImageNet: These contain significant levels of mislabeling (over 5%) (Northcutt et al., 2021b), hence we cannot do controlled experiments.

- MNIST and CIFAR-10: These are real image datasets. But they have been shown to contain almost no/little mislabeling (under 0.5%) (Northcutt et al., 2021b),.

**Tabular data**    : We conducted analogous experiments to the image experiments on tabular datasets, specifically, the OpenML (Vanschoren et al., 2014) datasets 'cover' and 'diabetes130US'. Note these datasets are included in a recent NeurIPS benchmark for tabular models (Grinsztajn et al., 2022). Additionally, this allows us to flexibly use the H-CAT framework with *any* tabular dataset as long as it conforms to the OpenML structure.

## B.3    BACKBONE MODELS

We use the following backbone models: LeNet (7-layer convolutional neural network) and ResNet-18 (18-layer deep convolutional neural network with "skip" or "residual" connections). We train the backbones with a cross-entropy loss using the Adam optimizer. Our learning rate of 1e-3 and batch size of 32 are kept fixed as they showed good convergence. As discussed, we explore different training epochs of 10 and 20 and show consistency irrespective, in Appendix D. This is reasonable as has been shown by (Seedat et al., 2022a) that the learning captured by HCMs (especially in training dynamics) happens in the early epochs. Note, we fix the backbone models in line with principles of Data-Centric AI of systematically assessing the data for a fixed ML model (Liang et al., 2022; Seedat et al., 2023b; Whang et al., 2023; Ng et al., 2021; Jarrahi et al., 2022).

### B.4 HCMs

We outline the implementation of the 13 HCMs supported by H-CAT below. We have described the HCMs in Appendix A.

**1. Learning-based (Margin)**

**AUM (Pleiss et al., 2020)** : Our implementation is based on [3].

**2. Learning-based (Uncertainty)**

**Data-IQ (Seedat et al., 2022a)** : Our implementation is based on [4].

**Data Maps (Swayamdipta et al., 2020)** : Our implementation is based on [5].

**3. Learning-based (Loss)**

**Large Loss (Xia et al., 2021)** : Our implementation computes the loss per sample individually.

**4. Learning-based (Gradient)**

**GraNd (Paul et al., 2021)** : Our implementation is based on the Pytorch implementation from [6] which is based on the original JAX [7].

**VoG (Agarwal et al., 2022)** : Our implementation is based on [8].

**5.Learning-based (Forgetting)**

**Forgetting scores (Toneva et al., 2019)** : Our implementation based on the original paper tracks the learning and forgetting times per sample in order to compute the forgetting score.

**6. Learning-based (Statistics)**

**EL2N (Paul et al., 2021)** : Our implementation is based on the Pytorch implementation from [9] which is based on the original JAX [10].

**Noise detector (Jia et al., 2022)** : Our implementation is based on [11]. We use the noise detector training scripts in the repo.

**7. Distance-based**

**Prototypicality (Sorscher et al., 2022)** : Our implementation follows the original paper and computes prototypicality as follows. We embed the image as a low-dimensional embedding using the backbone network. We then compute the Cosine distances to the mean embedding of the specific label for that sample. The cosine distances being smaller are more prototypical.

**8. Statistical-based**

---

[3]https://github.com/asappresearch/aum
[4]https://github.com/seedatnabeel/Data-IQ
[5]https://github.com/seedatnabeel/Data-IQ
[6]https://github.com/BlackHC/pytorch_datadiet
[7]https://github.com/mansheej/data_diet
[8]https://github.com/chirag126/VOG
[9]https://github.com/BlackHC/pytorch_datadiet
[10]https://github.com/mansheej/data_diet
[11]https://github.com/Christophe-Jia/mislabel-detection

**Cleanlab (Northcutt et al., 2021a)** : Our implementation is based on [12]

**ALLSH (Zhang et al., 2022)** : Our implementation follows the original paper. We compute the KL divergence in the model predictive probabilities for the pairs containing: (1) original input and (2) augmentation of the input. We perform the augmentation using AugLY [13]. The augmentations are selected from Horizontal flip, Random brightness, Random Noise and Random Pixelization.

**Agreement (Carlini et al., 2019)** : Our implementation of confidence agreement produces multiple predictions using MC-Dropout to create 10 pseudo-ensembles. We then compute the mean of the confidences as a measure of agreement, as per the original paper.

### B.5 IMPLEMENTATION DETAILS

We outline the experimental implementation details next. Details on the H-CAT and its usage can be found in Appendix C.

**Hardness detection & Ranking.** In this experiment, we set a different random seed for each of the three runs. For each run, we resample the dataset and change the samples that we perturb (i.e. which samples are hard). We also repeat the experiments for multiple proportions $p$.

We execute all experiments with 'p' values of [0.1, 0.2, 0.3, 0.4, 0.5], except for Far-OoD and Atypical experiments, where 'p' values are restricted to [0.05, 0.1, 0.15, 0.2, 0.25]. The rationale behind this distinction is that, in reality, Far-OoD and Atypical conditions do not typically occur in high proportions. For instance, a sample can't be considered Atypical if it constitutes 50% of occurrences. Therefore, for these conditions, we limit 'p' to a maximum of 0.25.

**Stability/Consistency** In this experiment, we maintain a consistent random data split and the set of perturbed samples, which represent the difficult examples, is also kept constant across different runs. The element of randomness is brought in solely through the training of the backbone model and by extension, the characterization of the HCM. i.e. we have a different seed for the model. This approach is taken because our aim is to evaluate the Stability/Consistency of the HCM across multiple runs when presented with the same data.

---

[12]https://github.com/cleanlab/cleanlab
[13]https://github.com/facebookresearch/AugLy

## C    USING H-CAT

This appendix discusses how to use the H-CAT framework. We link to the codebase, provide a tutorial on using H-CAT (see Sec. C.2), and also show how H-CAT can be extended to new HCMs and datasets (see Sec. C.3).

### C.1    H-CAT BENCHMARKING FRAMEWORK REPOSITORY

To foster ease of use and engagement, we release the H-CAT Benchmarking Framework's codebase with Apache 2.0 License. It can be found at:
https://github.com/seedatnabeel/H-CAT

### C.2    TUTORIAL ON USING H-CAT

Below, we illustrate a tutorial demonstrating how to use H-CAT, highlighting the composable nature and the object-oriented interface making it easy to use. We also provide this in the form of a Jupyter notebook called *tutorial.ipynb*.

```python
from src.trainer import PyTorchTrainer
from src.dataloader import MultiFormatDataLoader
from src.models import *
from src.evaluator import Evaluator
```

Listing 1: Step 1: Necessary imports

```python
hardness = "uniform"
p=0.1
dataset = "mnist"
model_name = "lenet"
epochs = 20
seed = 0

# Defined by prior or domain knowledge
if hardness =="instance":
    if dataset == "mnist":
        rule_matrix = {
                    1: [7],
                    2: [7],
                    3: [8],
                    4: [4],
                    5: [6],
                    6: [5],
                    7: [1, 2],
                    8: [3],
                    9: [7],
                    0: [0]
                }
    if dataset == "cifar":

        rule_matrix = {
                    0: [2],    # airplane (unchanged)
                    1: [9],    # automobile -> truck
                    2: [9],    # bird (unchanged)
                    3: [5],    # cat -> automobile
                    4: [5,7],   # deer (unchanged)
                    5: [3, 4],   # dog -> cat
                    6: [6],    # frog (unchanged)
                    7: [5],    # horse -> dog
                    8: [7],    # ship (unchanged)
                    9: [9],    # truck -> horse
                }

else:
```

```
39      rule_matrix = None
```
Listing 2: Step 2: Define experimental parameters

```
1  characterization_methods =  [
2               "aum",
3               "data_uncert", # for both Data-IQ and Data-Maps
4               "el2n",
5               "grand",
6               "cleanlab",
7               "forgetting",
8               "vog",
9               "prototypicality",
10              "allsh",
11              "loss",
12              "conf_agree",
13              "detector"
14          ],
```
Listing 3: Step 3: Define HCMs to evaluate — if excluded we evaluate all

```
1  import torch
2  import torch.nn as nn
3  import torch.optim as optim
4  from torchvision import datasets, transforms
5
6  if dataset == 'cifar':
7      # Define transforms for the dataset
8      transform = transforms.Compose([
9          transforms.ToTensor(),
10         transforms.Normalize((0.5, 0.5, 0.5), (0.5, 0.5, 0.5))
11     ])
12
13     # Load the CIFAR-10 dataset
14     train_dataset = datasets.CIFAR10(root="./data", train=True, download=
       True, transform=transform)
15     test_dataset = datasets.CIFAR10(root="./data", train=False, download=
       True, transform=transform)
16
17 elif dataset =='mnist':
18     # Define transforms for the dataset
19     transform = transforms.Compose(
20         [transforms.ToTensor(),
21         transforms.Normalize((0.5,), (0.5,))])
22
23
24     # Load the MNIST dataset
25     train_dataset = datasets.MNIST(root="./data", train=True, download=
       True, transform=transform)
26     test_dataset = datasets.MNIST(root="./data", train=False, download=
       True, transform=transform)
27
28
29 total_samples = len(train_dataset)
30 num_classes = 10
31
32 # Set device to use
33 device = torch.device("cuda" if torch.cuda.is_available() else "cpu")
```
Listing 4: Step 4: Load datasets

```
1  # Allows importing data in multiple formats
2
3  dataloader_class = MultiFormatDataLoader(data=train_dataset,
4                                           target_column=None,
5                                           data_type='torch_dataset',
6                                           batch_size=64,
7                                           shuffle=True,
8                                           num_workers=0,
9                                           transform=None,
10                                          image_transform=None,
11                                          perturbation_method=hardness,
12                                          p=p,
13                                          rule_matrix=rule_matrix
14          )
15
16
17 dataloader, dataloader_unshuffled = dataloader_class.get_dataloader()
18 flag_ids = dataloader_class.get_flag_ids()
```

Listing 5: Step 5: Call H-CAT Dataloader Module (with Hardness module parameterized)

```
1  # Instantiate the neural network and optimizer
2  if dataset == 'cifar':
3      if model_name == 'LeNet':
4          model = LeNet(num_classes=10).to(device)
5      if model_name == 'ResNet':
6          model = ResNet18().to(device)
7  elif dataset == 'mnist':
8      if model_name == 'LeNet':
9          model = LeNetMNIST(num_classes=10).to(device)
10     if model_name == 'ResNet':
11         model = ResNet18MNIST().to(device)
12
13 criterion = nn.CrossEntropyLoss()
14 optimizer = optim.Adam(model.parameters(), lr=0.001)
15
16
17 # Instantiate the PyTorchTrainer class
18 trainer = PyTorchTrainer(model=model,
19                          criterion=criterion,
20                          optimizer=optimizer,
21                          lr=0.001,
22                          epochs=epochs,
23                          total_samples=total_samples,
24                          num_classes=num_classes,
25                          characterization_methods=
     characterization_methods,
26                          device=device)
27
28 # Train the model
29 trainer.fit(dataloader, dataloader_unshuffled)
30
31 hardness_dict = trainer.get_hardness_methods()
```

Listing 6: Step 6: Call H-CAT Trainer Module (with HCM module parameterized)

```
1  eval = Evaluator(hardness_dict=hardness_dict, flag_ids=flag_ids, p=p)
2
3  eval_dict, raw_scores_dict = eval.compute_results()
4
5  print(eval_dict)
```

Listing 7: Step 7: Call H-CAT Evaluator Module

### C.3 EXTENDING H-CAT

#### C.3.1 ADDING A NEW HCM

New HCMs can easily be added to H-CAT. We describe the process below.

We take an object-oriented approach and hence new HCMs can be included by inheriting from our base class Hardness_Base defined below.

```python
# Base class for HCMs
class Hardness_Base:
    def __init__(self, name):
        self.name = name
        self._scores = None

    def updates(self):
        pass

    def compute_scores(self):
        pass

    @property
    def scores(self):
        if self._scores is None:
            self.compute_scores()
        return self._scores
```

Listing 8: HCM Base class

To create a new HCM, you would create a new class that inherits from Hardness_Base. In this new subclass, override the updates and compute_scores methods with the specific logic for this new hardness calculation method. We show an example below.

```python
class MyNewHCM(Hardness_Base):
    def __init__(self, name):
        super().__init__(name)

    def updates(self):
        # logic to perform updates goes here
        pass

    def compute_scores(self):
        # logic to compute scores goes here
        self._scores = "some computed value"  # Replace this with your
    computation
```

Listing 9: Example New HCM

The HCM then needs to be added to *trainer.py* in the correct part of the training loop and can be used as below

```python
hcm = MyNewHCM('some name')
hcm.updates()
print(hcm.scores)
```

Listing 10: Usage of the new defined HCM

Note, one needs to think for the HCM if it gets updated every epoch or at the end of training to define placement in the training loop.

### C.3.2 USING A DIFFERENT DATASET

We can use different datasets with H-CAT quite simply.

Before delving further, we show below where the new dataset needs to be passed to the *Dataloader module*. i.e. We need to pass *train_dataset*.

```
# Allows importing data in multiple formats

dataloader_class = MultiFormatDataLoader(data=train_dataset,
                                         target_column=None,
                                         data_type='torch_dataset',
                                         batch_size=64,
                                         shuffle=True,
                                         num_workers=0,
                                         transform=None,
                                         image_transform=None,
                                         perturbation_method=hardness,
                                         p=p,
                                         rule_matrix=rule_matrix
        )
```

Listing 11: Example of passing data to the Dataloader module

The easiest way is to define your dataset as a standard torch dataset outside of H-CAT and simply to pass that into the *MultiFormatDataLoader* and specify a *target_column=None*.

If one has the data as *NumPy arrays* or *torch tensors*, one would need to pass a tuple as *data = (X,y)* and specify a *target_column*.

Other formats are also possible with H-CAT. For instance, if you have images in folders. One could pass data as a CSV file, JSON, or Python dictionary — wherein there is a column with paths to each sample and a column with the target. One can then pass the path to the CSV file or JSON file or pass the Python dict itself. In addition, one should specify the *target_column*. The rest is handled internally within H-CAT to convert the data to an appropriate Torch dataset for training.

That said, ideally if the dataset can be converted to a Torch dataset before passing it to H-CAT that is preferred.

# D    ADDITIONAL EXPERIMENTS

This appendix presents additional experiments and full results across all setups using H-CAT to evaluate different HCMs.

## D.1    MULTIPLE HARDNESS TYPES SIMULTANEOUSLY

**Overview.** As discussed in the main paper, we primarily focussed on single-source hardness. We might also expect cases where multiple hardness types manifest simultaneously. For instance, if an image is Near-OoD (e.g. blurry or different texture), we might be more likely also to mislabel it. As a demonstration of the usage of H-CAT for this purpose, we hence assess the case where both Near-OoD (Covariate and Domain) are paired with Instance-specific mislabeling.

We want to understand the effect of this on HCM capabilities.

**Results.** We show results for both Near-OoD (Covariate) + Instance and Near-OoD (Domain) + Instance below in Figure 8.

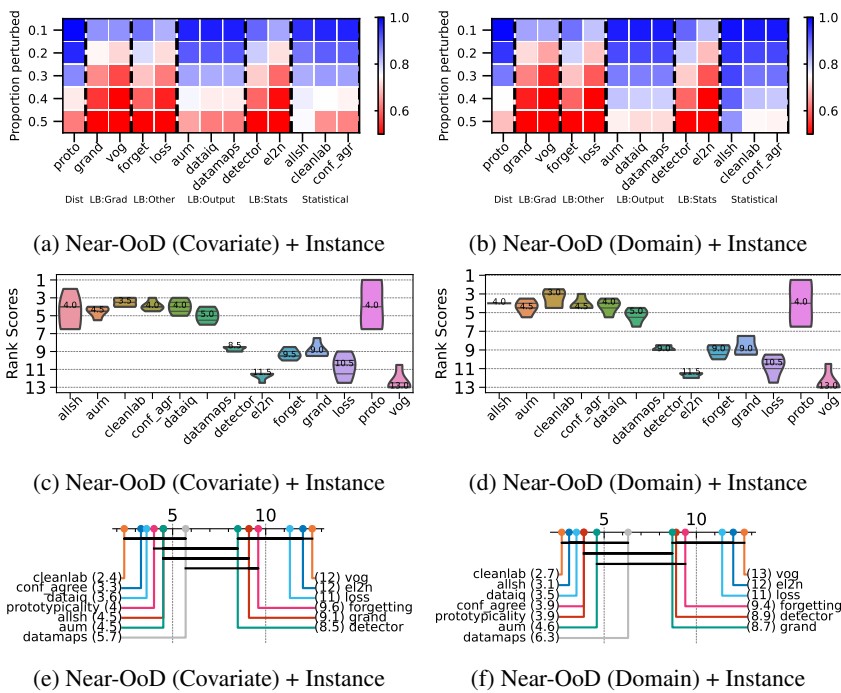

(a) Near-OoD (Covariate) + Instance          (b) Near-OoD (Domain) + Instance

(c) Near-OoD (Covariate) + Instance          (d) Near-OoD (Domain) + Instance

(e) Near-OoD (Covariate) + Instance          (f) Near-OoD (Domain) + Instance

Figure 8: Results using H-CAT for multiple hardness types manifesting simultaneously.

**Takeaway.** Our results indicate that H-CAT can be used to assess multiple hardness types, which manifest simultaneously. We find that HCMs which perform well individually on both types of hardness — still perform well in the simultaneous setting.

## D.2   STABILITY/CONSISTENCY: BACKBONE MODEL TYPE

**Overview.** We assessed HCMs using two different backbone models - LeNet and ResNet-18. We wish to assess the stability/consistency of the backbone model. As discussed in the main paper, we can quantify this based on the Spearman rank correlation — i.e., are the HCMs ordering the samples consistently in a stable manner?

In the main paper, we assessed, within model stability/consistency, the randomness of a run. However, here we assess the rank correlation of models head-to-head for the same hardness type.

**Results.** Figure 9 shows the results across multiple hardness types.

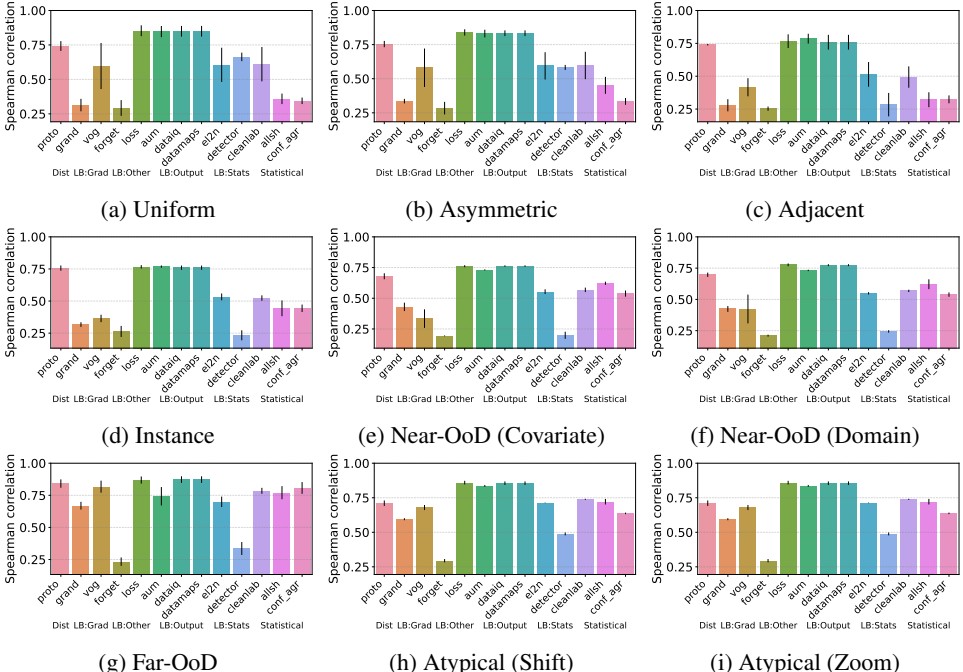

Figure 9: Stability/consistency to model types: higher Spearman rank correlations indicate greater consistency and stability.

**Takeaway.** Of course, the LeNet and ResNet-18 are significantly different in model type and size. However, we find that compared to the results in the main paper, we see minimal degradation in the Spearman rank correlation. This indicates that the backbone model to do the hardness characterization is minimally impacted by the backbone model. i.e. those HCMs which were the most stable and consistent remain the most stable and consistent.

### D.3 STABILITY/CONSISTENCY: PARAMETERIZATIONS

**Overview.** We can train the backbone models differently, with different parameterizations. For instance, with different numbers of epochs, 10 or 20. We wish to assess stability/consistency with respect to different training parameterizations when performing the HCM evaluation. As discussed in the main paper, we can quantify this based on the Spearman rank correlation — i.e. are the HCMs ordering the samples consistently in a stable manner?

We make a head-to-head comparison of scores obtained from the HCMs applied to backbone models trained for 10 and 20 epochs. Ideally, we still would want a high Spearman correlation indicating sample order is maintained.

**Results.** Figure 10 shows the results across multiple hardness types.

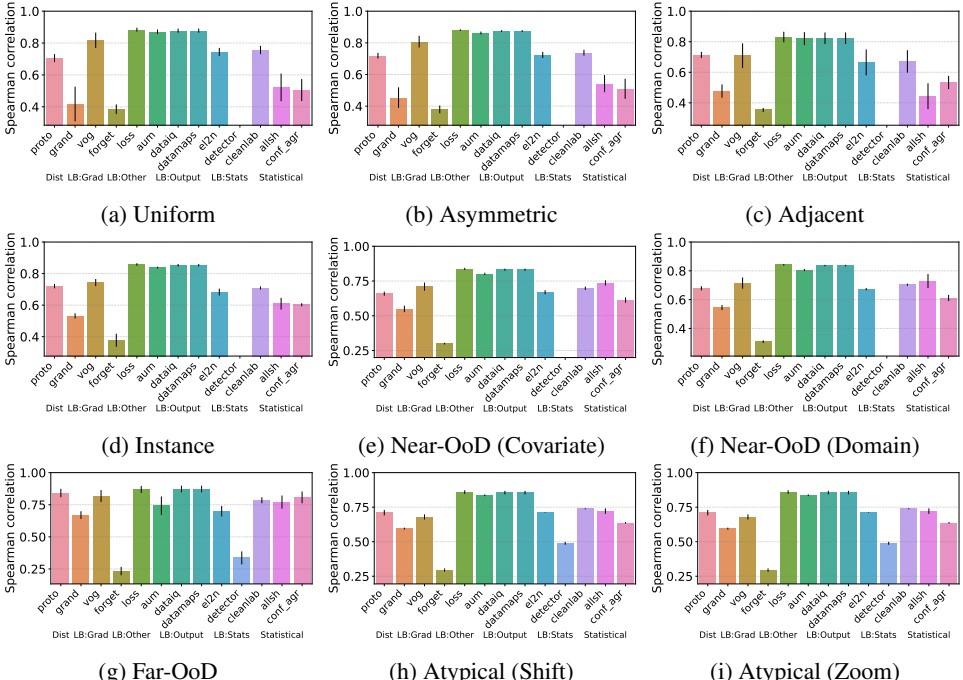

Figure 10: Stability/consistency to parameterizations: higher Spearman rank correlations indicate greater consistency and stability.

**Takeaway.** We investigated the stability/consistency of the model parameterization in terms of the number of epochs used to train the backbone models. We find the Spearman correlations are not as affected by the epoch number as potentially expected. We see the magnitudes of the Spearman correlation for the different HCMs are similar to that of the main paper. Moreover, the HCMs, which were the most stable and consistent, remain the most stable and consistent.

This result is understandable, as it was shown in (Seedat et al., 2022a) that the majority of variation comes from the early epochs (before 10 epochs), hence after the training dynamics on which most of the metrics are computed stabilize. This means that it won't affect our HCM characterization whether we continue training the backbone till 20 epochs. This result simply mirrors this result from the literature and highlights for benchmarking purposes that we don't need to train the backbone model for too many epochs in order to have a good assessment of the HCMs' capabilities.

### D.4 ADDITIONAL AGGREGATED RESULTS

**Overview.** In the main paper, we presented aggregate results for 6 out of 9 hardness manifestations or types. We present the results for the remaining three Adjacent mislabeling, Near-OOD (Domain) and Atypical (Zoom), below in Figures 11-14.

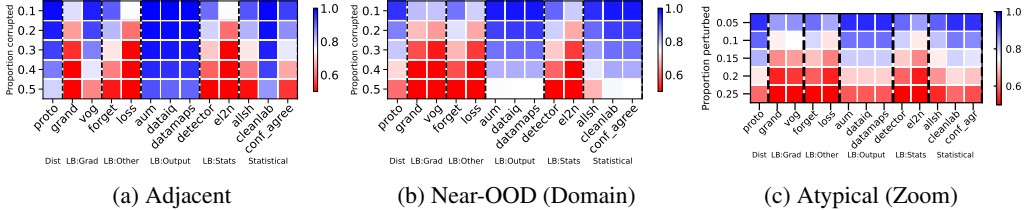

(a) Adjacent        (b) Near-OOD (Domain)        (c) Atypical (Zoom)

Figure 11: AUPRC for different HCMs for different hardness types aggregated across setups. We vary the proportion perturbed. i.e. proportion of hard examples. Blue is better, red worse. We see the variability of HCM capabilities across hardness types and proportions.

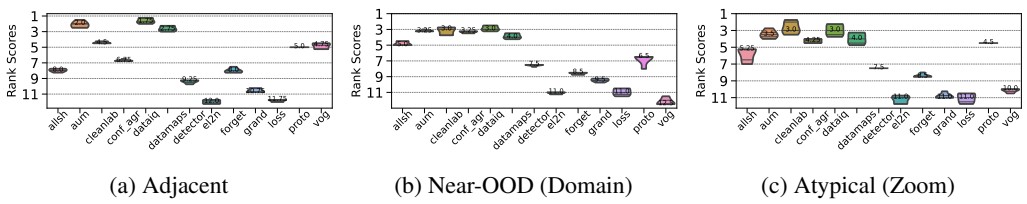

(a) Adjacent        (b) Near-OOD (Domain)        (c) Atypical (Zoom)

Figure 12: Performance rankings of HCMs vary depending on the hardness type.

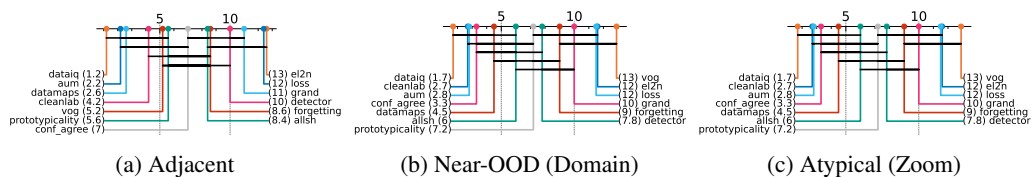

(a) Adjacent        (b) Near-OOD (Domain)        (c) Atypical (Zoom)

Figure 13: Critical difference diagrams highlight that similar categories/classes of HCMs *do not* have a statistically significant difference in their performance, indicated by the horizontal black lines linking HCMs which are not statistically different. The numbers in brackets denote mean rank.

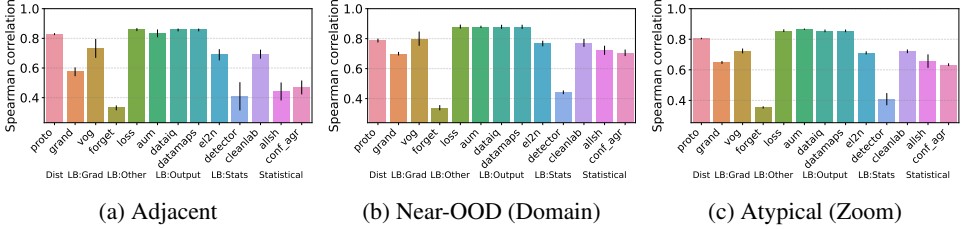

(a) Adjacent        (b) Near-OOD (Domain)        (c) Atypical (Zoom)

Figure 14: Certain classes of HCMs are more stable and consistent than others (retaining rank order), with higher Spearman rank correlations for multiple runs of the HCM on the same data

## D.5 INDIVIDUAL RESULTS - IMAGES

### D.5.1 MISLABELING: ADJACENT

**Heatmaps.**

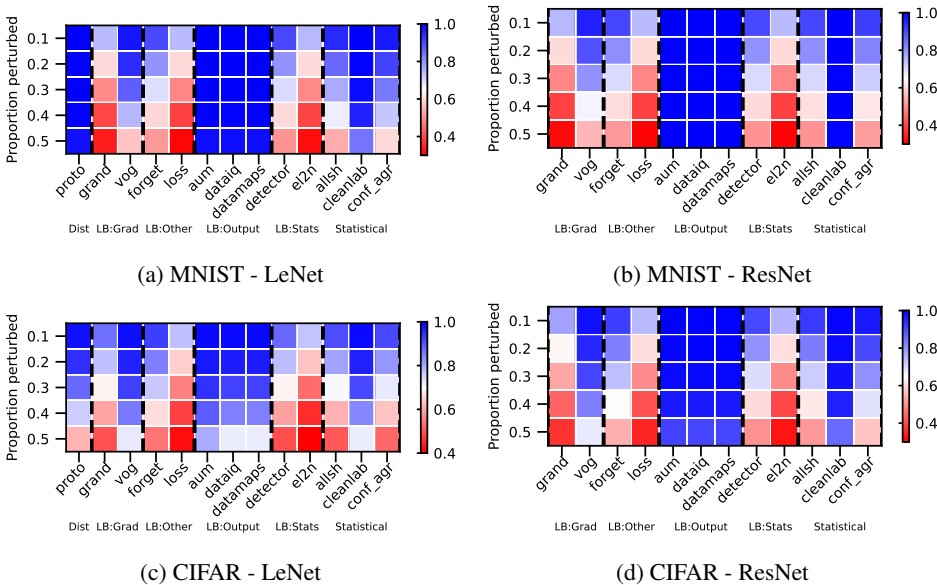

(a) MNIST - LeNet

(b) MNIST - ResNet

(c) CIFAR - LeNet

(d) CIFAR - ResNet

Figure 15: Adjacent mislabeling – AUPRC. We vary the proportion perturbed. i.e. proportion of hard examples. Blue is better, red worse.

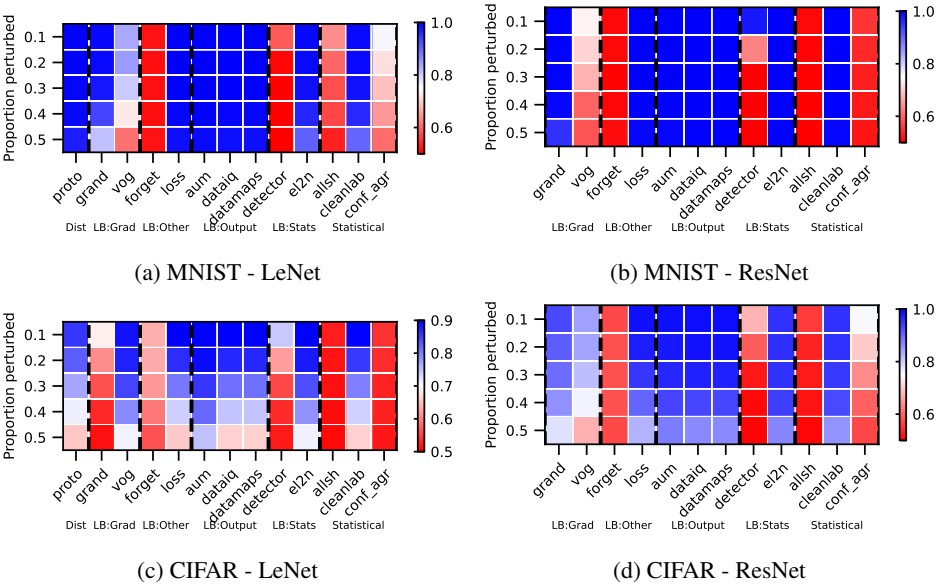

(a) MNIST - LeNet

(b) MNIST - ResNet

(c) CIFAR - LeNet

(d) CIFAR - ResNet

Figure 16: Adjacent mislabeling – AUROC. We vary the proportion perturbed. i.e. proportion of hard examples. Blue is better, red worse.

**Rankings.**

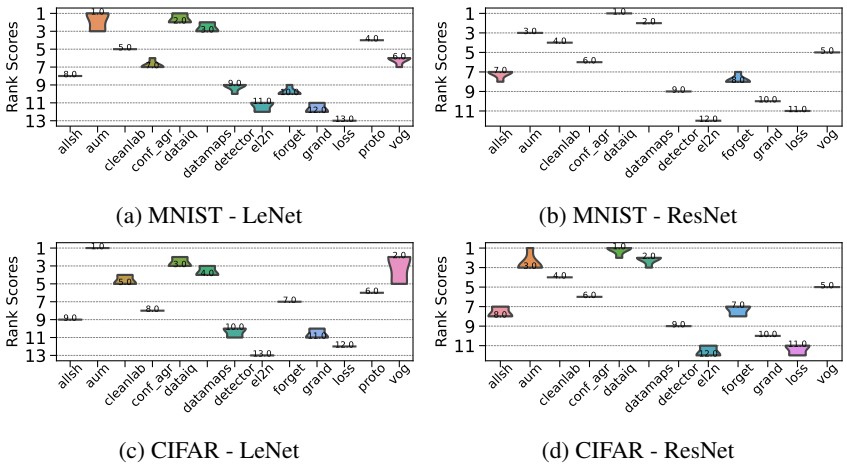

Figure 17: Adjacent mislabeling – AUPRC. Ranking of HCMs.

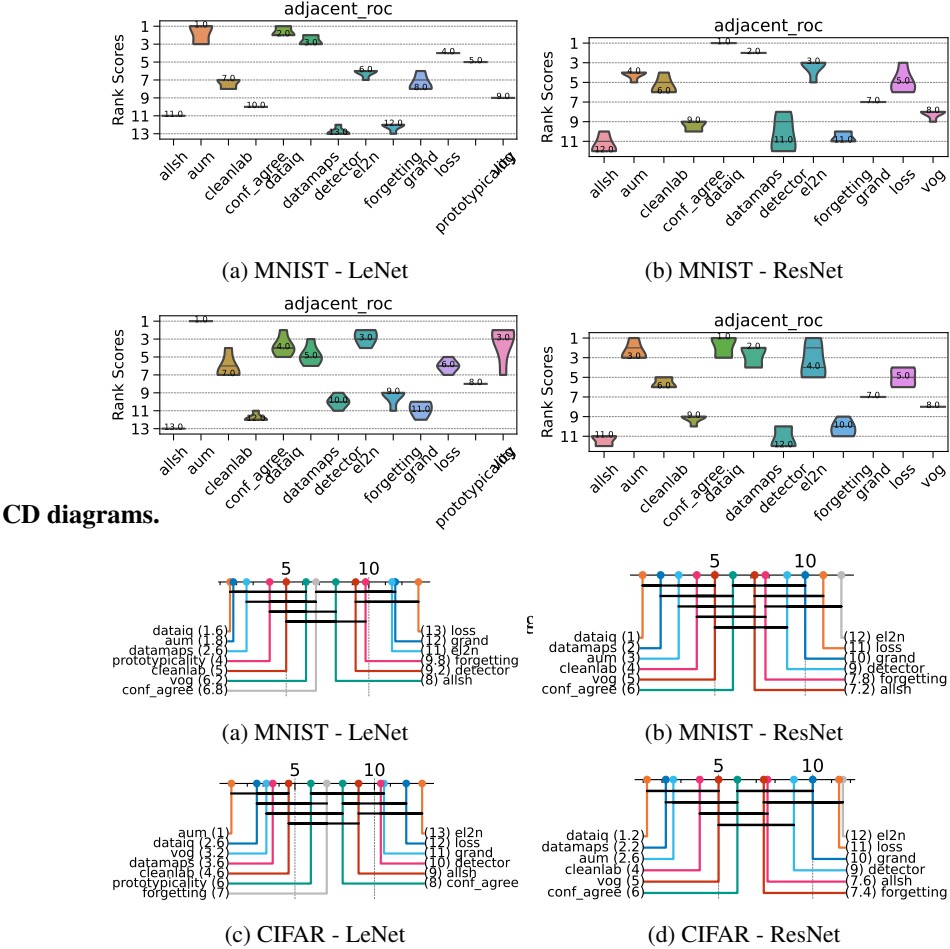

**CD diagrams.**

Figure 19: Adjacent mislabeling - AUPRC. Critical difference diagrams. Black lines connect methods that are not statistically significantly different.

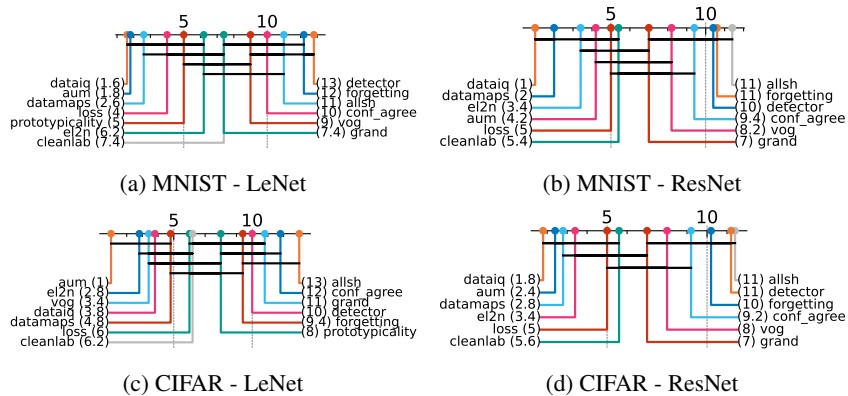

(a) MNIST - LeNet

(b) MNIST - ResNet

(c) CIFAR - LeNet

(d) CIFAR - ResNet

Figure 20: Adjacent mislabeling - AUROC. Critical difference diagrams. Black lines connect methods that are not statistically significantly different.

## Matrix compare

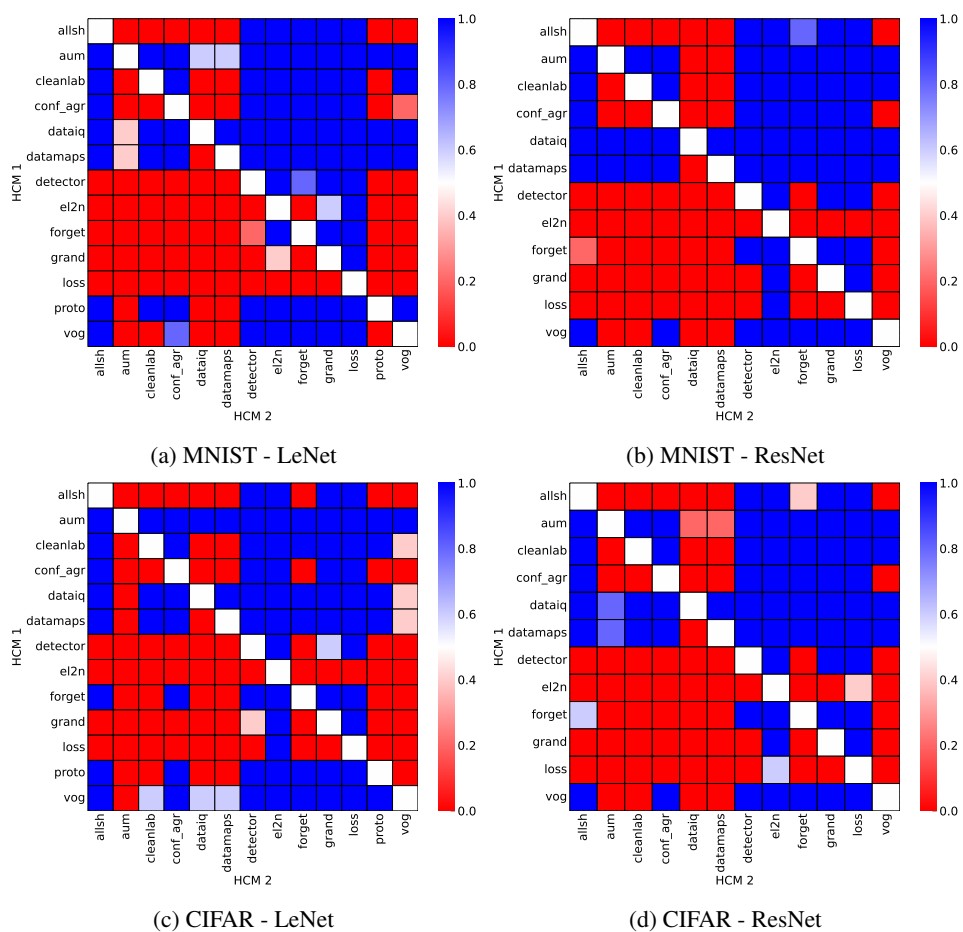

(a) MNIST - LeNet

(b) MNIST - ResNet

(c) CIFAR - LeNet

(d) CIFAR - ResNet

Figure 21: Adjacent mislabeling. Head-to-head comparison matrix assessing for runs when a certain HCM outperforms another. i.e. when y-axis (HCM 1) beats x-axis competitor (HCM 2).

### D.5.2 MISLABELING: ASYMMETRIC

**Heatmaps.**

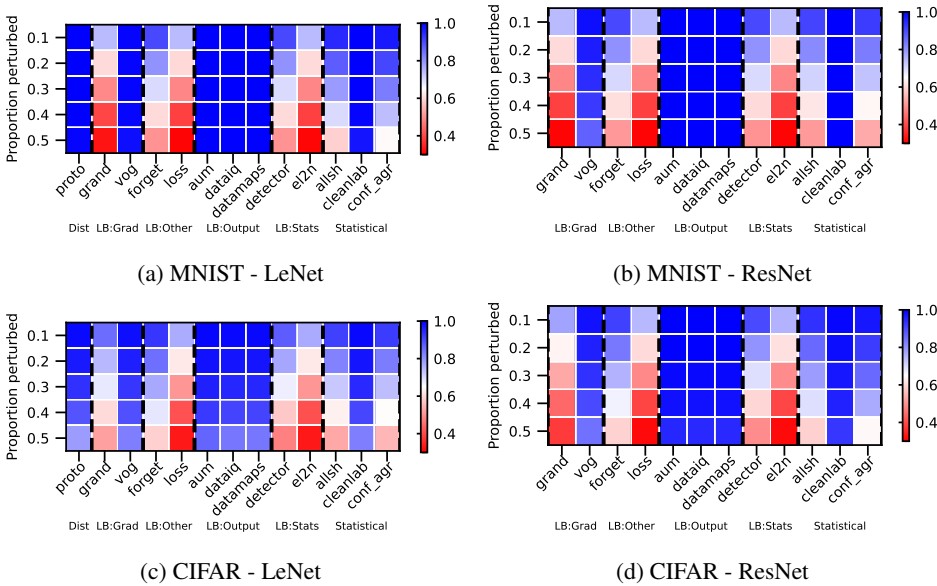

Figure 22: Asymmetric mislabeling - AUPRC. We vary the proportion perturbed. i.e. proportion of hard examples. Blue is better, red worse.

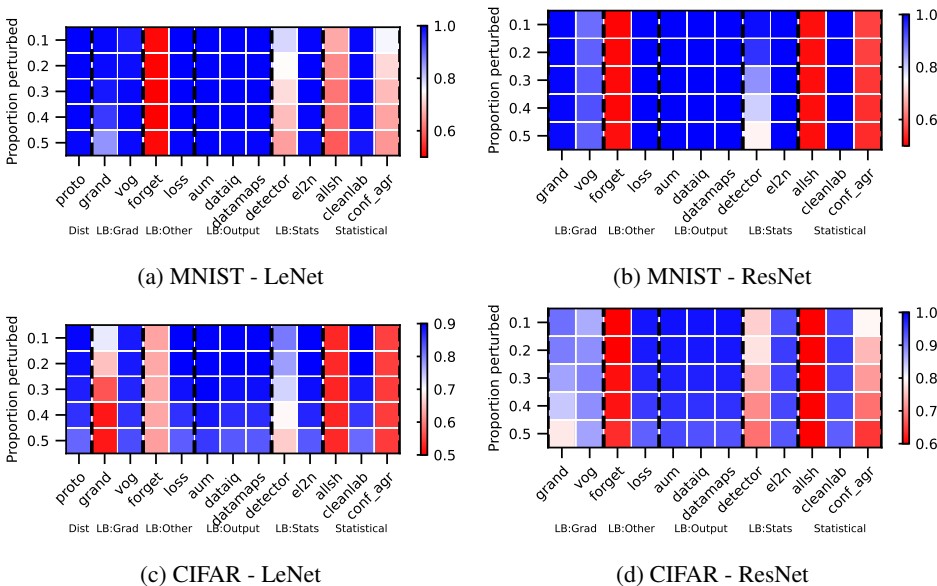

Figure 23: Asymmetric mislabeling - AUROC. We vary the proportion perturbed. i.e. proportion of hard examples. Blue is better, red worse.

**Rankings.**

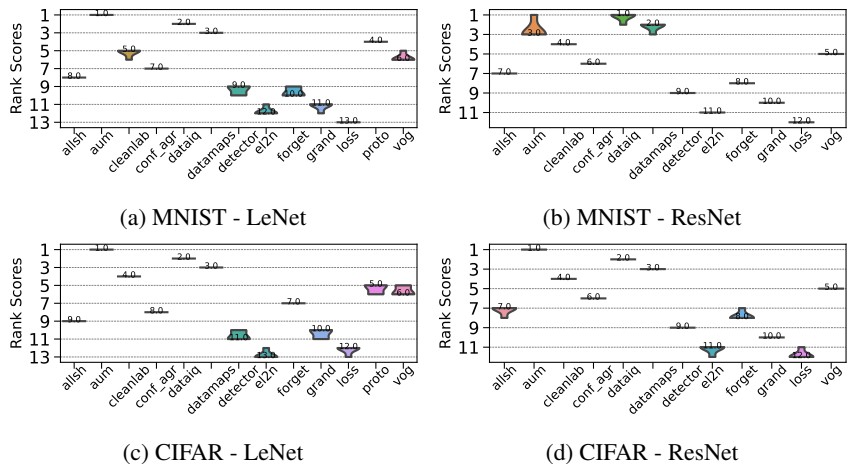

Figure 24: Asymmetric mislabeling - AUPRC. Ranking of HCMs.

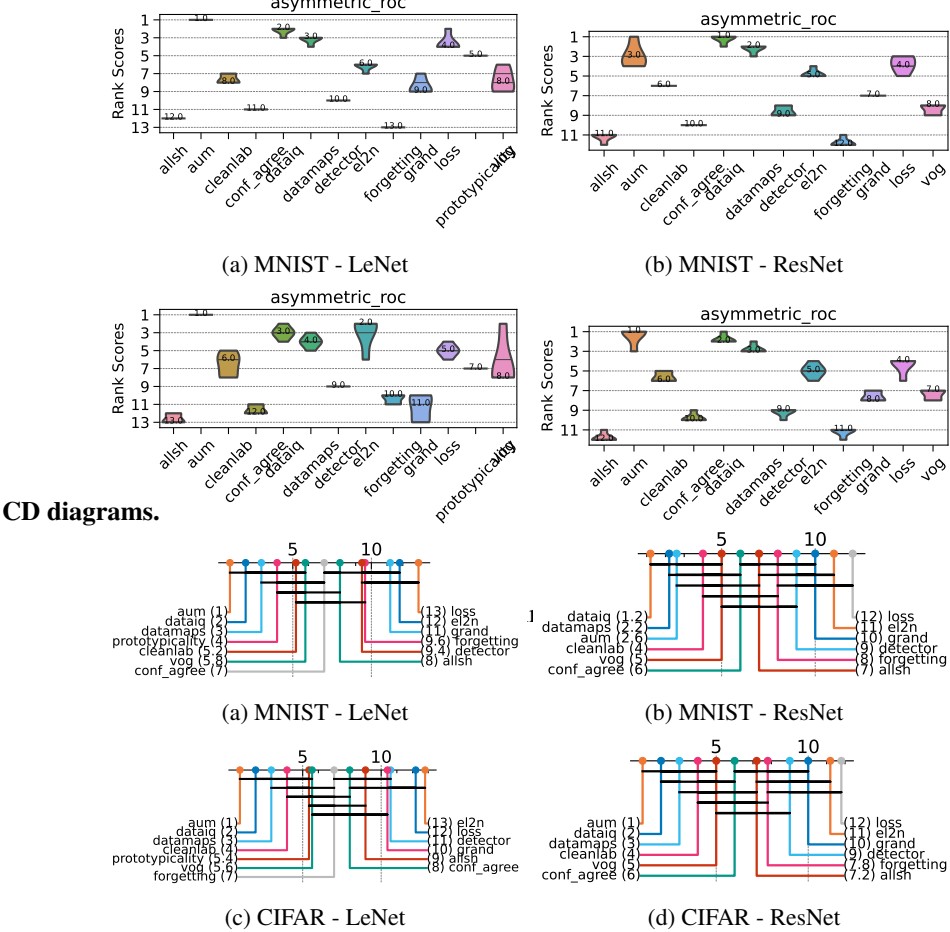

**CD diagrams.**

Figure 26: Asymmetric mislabeling - AUPRC. Critical difference diagrams. Black lines connect methods that are not statistically significantly different.

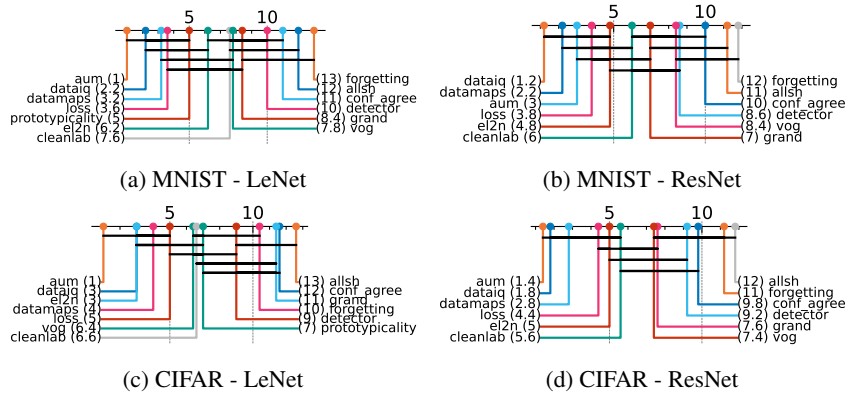

(a) MNIST - LeNet  (b) MNIST - ResNet

(c) CIFAR - LeNet  (d) CIFAR - ResNet

Figure 27: Asymmetric mislabeling - AUROC. Critical difference diagrams. Black lines connect methods that are not statistically significantly different.

**Matrix compare**

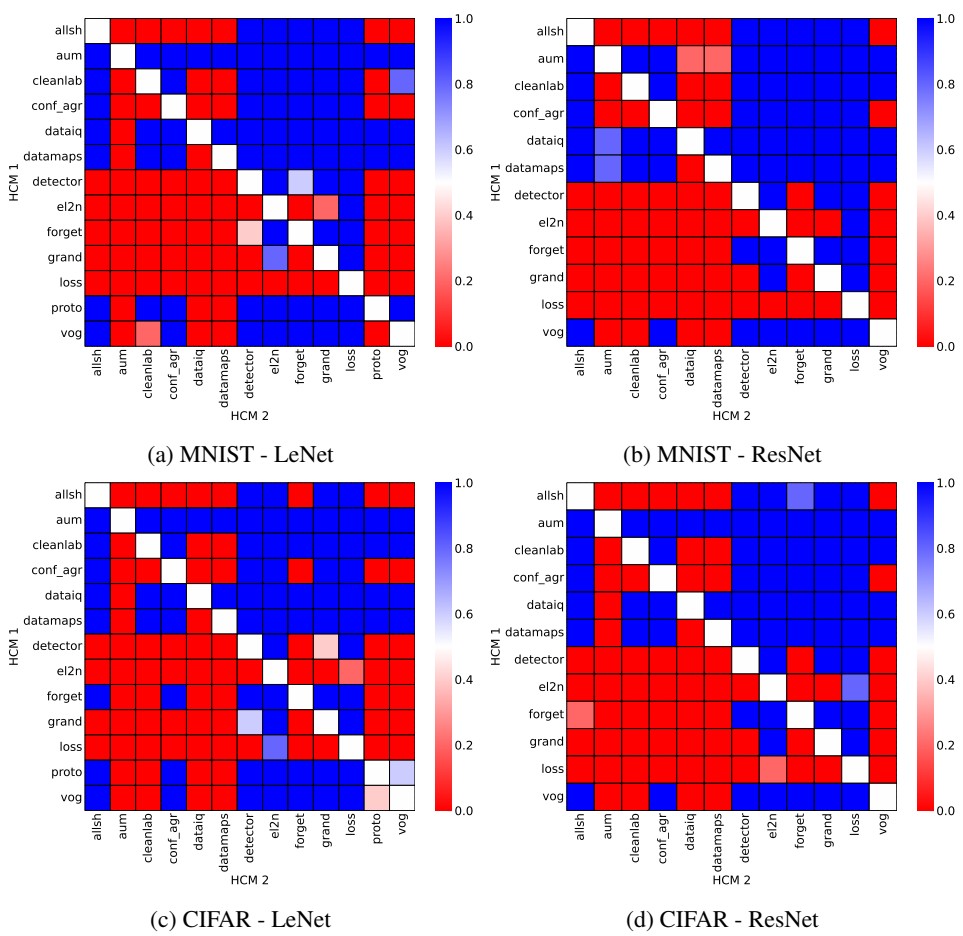

(a) MNIST - LeNet  (b) MNIST - ResNet

(c) CIFAR - LeNet  (d) CIFAR - ResNet

Figure 28: Asymmetric mislabeling. Head-to-head comparison matrix assessing for runs when a certain HCM outperforms another. i.e. when y-axis (HCM 1) beats x-axis competitor (HCM 2).

### D.5.3 MISLABELING: INSTANCE

**Heatmaps.**

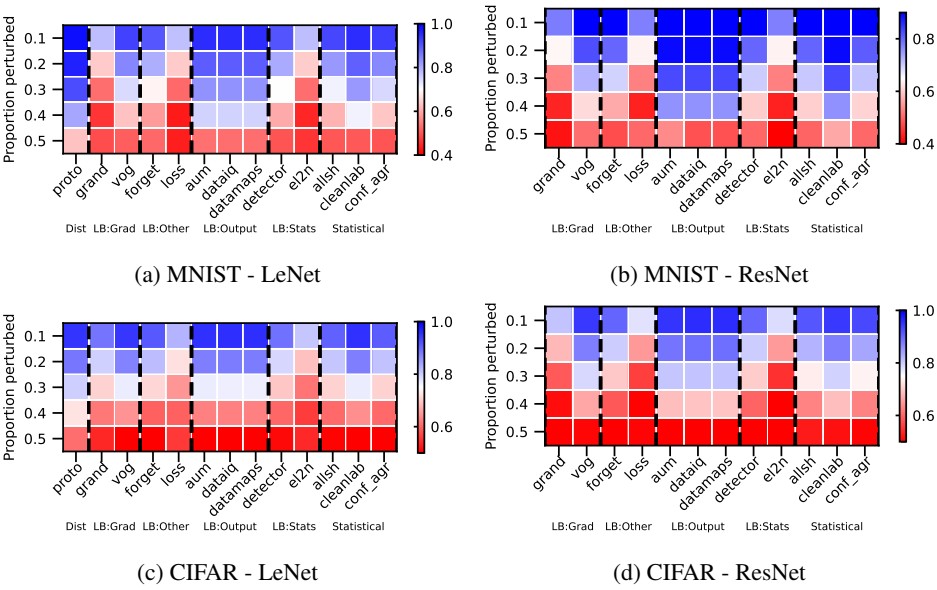

(a) MNIST - LeNet

(b) MNIST - ResNet

(c) CIFAR - LeNet

(d) CIFAR - ResNet

Figure 29: Instance mislabeling - AUPRC. We vary the proportion perturbed. i.e. proportion of hard examples. Blue is better, red worse.

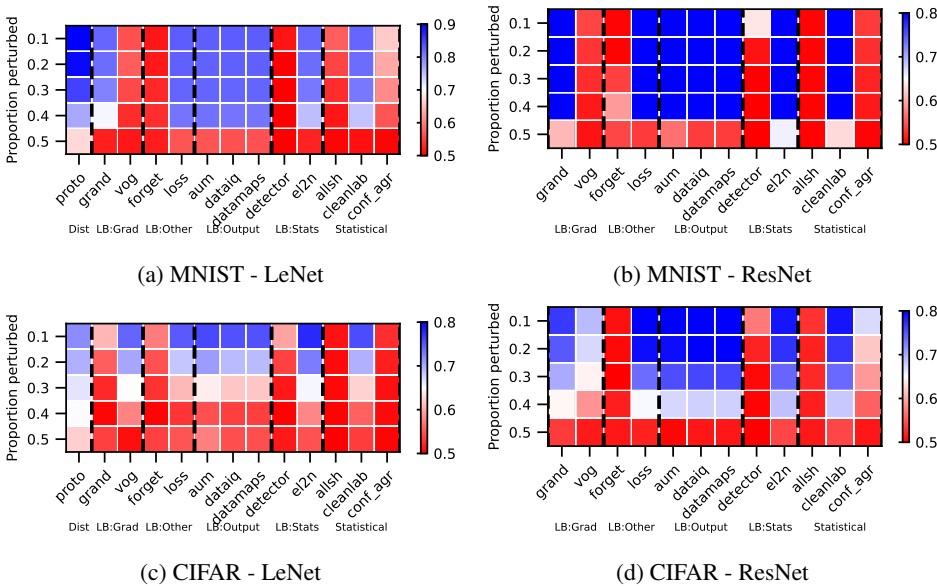

(a) MNIST - LeNet

(b) MNIST - ResNet

(c) CIFAR - LeNet

(d) CIFAR - ResNet

Figure 30: Instance mislabeling - AUROC. We vary the proportion perturbed. i.e. proportion of hard examples. Blue is better, red worse.

**Rankings.**

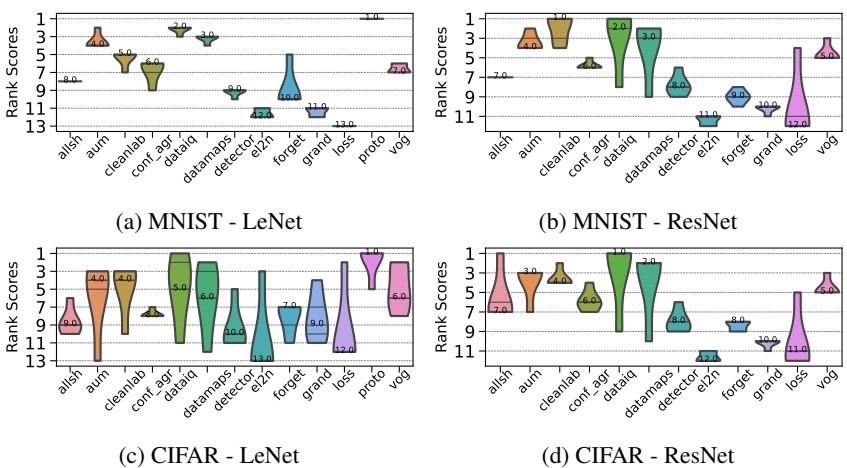

(a) MNIST - LeNet

(b) MNIST - ResNet

(c) CIFAR - LeNet

(d) CIFAR - ResNet

Figure 31: Instance mislabeling - AUPRC. Ranking of HCMs.

**CD diagrams.**

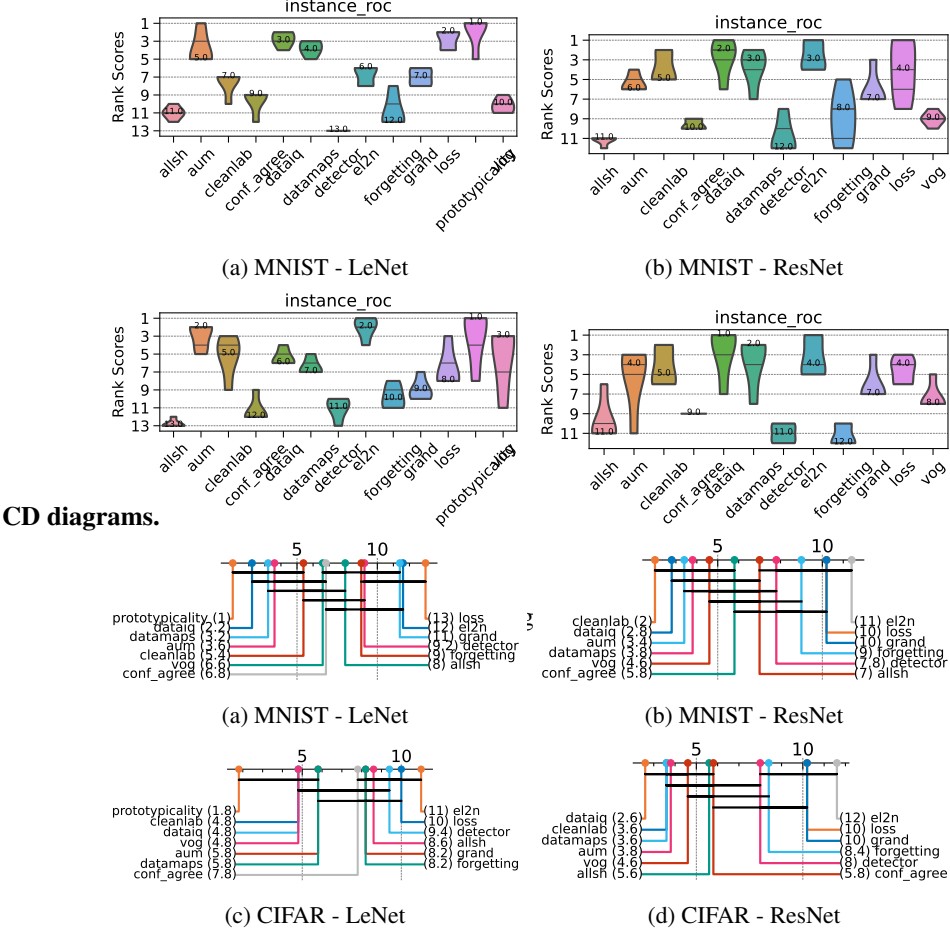

(a) MNIST - LeNet

(b) MNIST - ResNet

(c) CIFAR - LeNet

(d) CIFAR - ResNet

Figure 33: Instance mislabeling - AUPRC. Critical difference diagrams. Black lines connect methods that are not statistically significantly different.

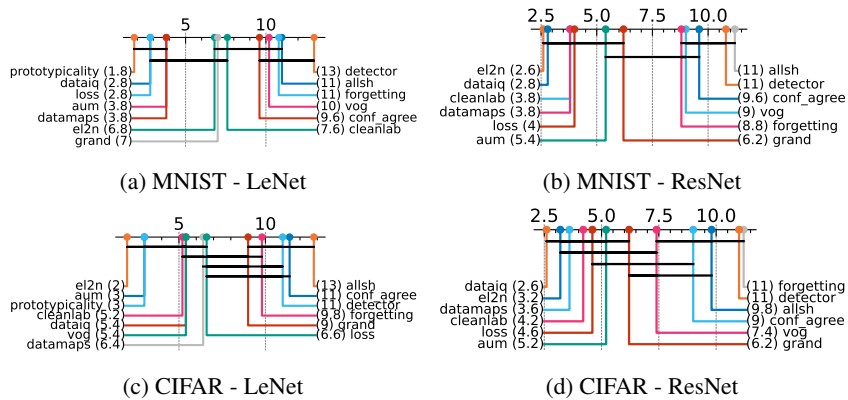

(a) MNIST - LeNet

(b) MNIST - ResNet

(c) CIFAR - LeNet

(d) CIFAR - ResNet

Figure 34: Instance mislabeling - AUROC. Critical difference diagrams. Black lines connect methods that are not statistically significantly different.

## Matrix compare

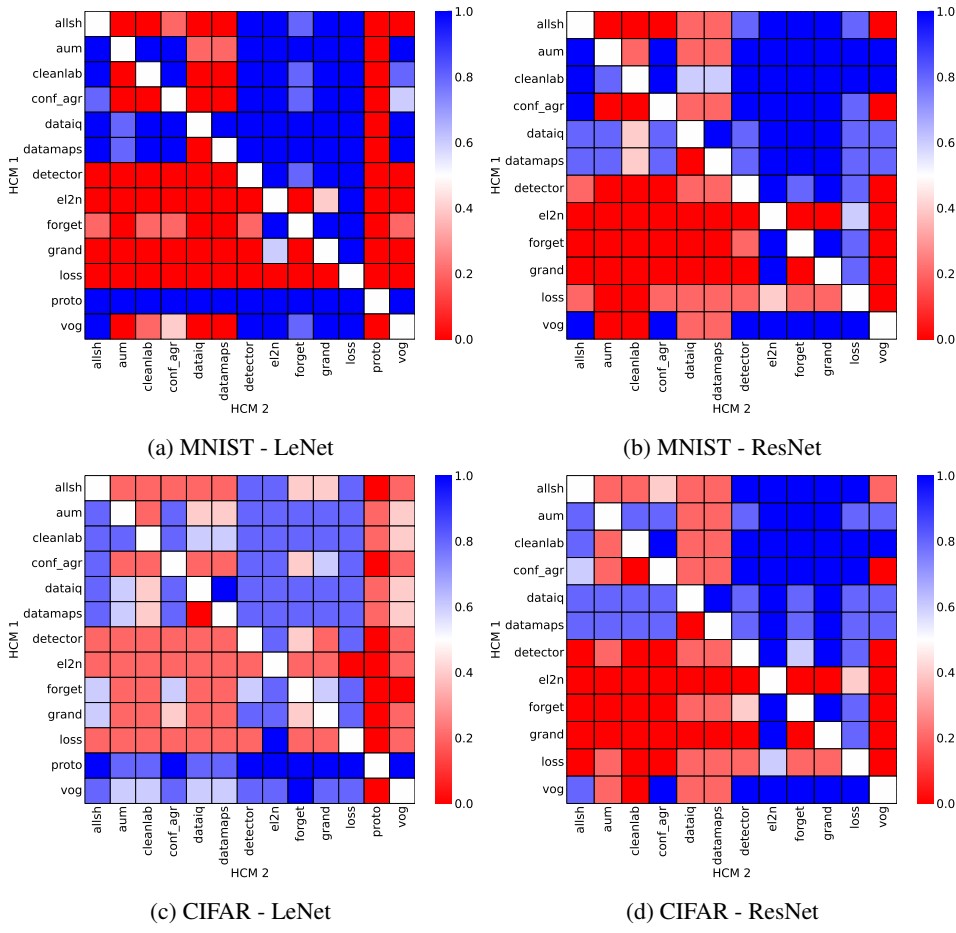

(a) MNIST - LeNet

(b) MNIST - ResNet

(c) CIFAR - LeNet

(d) CIFAR - ResNet

Figure 35: Instance mislabeling. Head-to-head comparison matrix assessing for runs when a certain HCM outperforms another. i.e. when y-axis (HCM 1) beats x-axis competitor (HCM 2).

### D.5.4   MISLABELING: UNIFORM

**Heatmaps.**

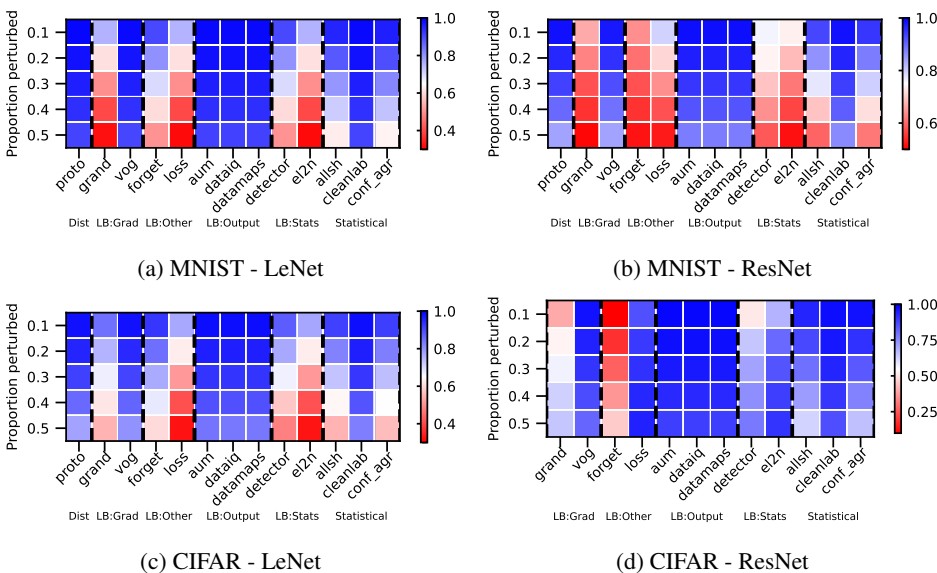

(a) MNIST - LeNet

(b) MNIST - ResNet

(c) CIFAR - LeNet

(d) CIFAR - ResNet

Figure 36: Uniform mislabeling - AUPRC. We vary the proportion perturbed. i.e. proportion of hard examples. Blue is better, red worse.

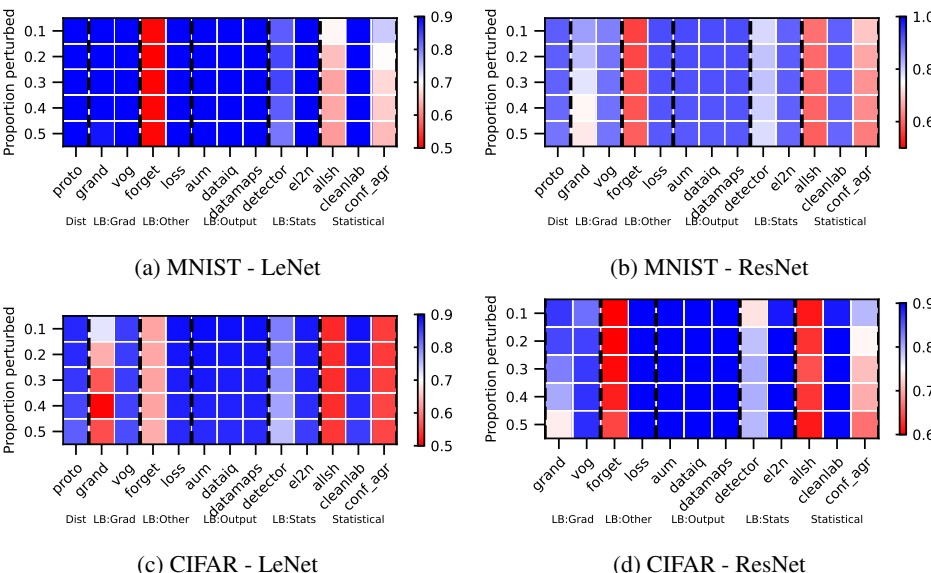

(a) MNIST - LeNet

(b) MNIST - ResNet

(c) CIFAR - LeNet

(d) CIFAR - ResNet

Figure 37: Uniform mislabeling - AUROC. We vary the proportion perturbed. i.e. proportion of hard examples. Blue is better, red worse.

**Rankings.**

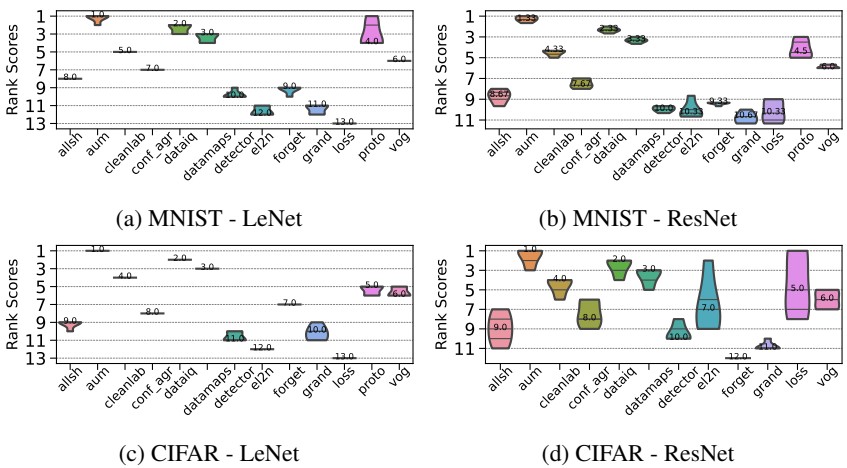

(a) MNIST - LeNet

(b) MNIST - ResNet

(c) CIFAR - LeNet

(d) CIFAR - ResNet

Figure 38: Uniform mislabeling - AUPRC. Ranking of HCMs.

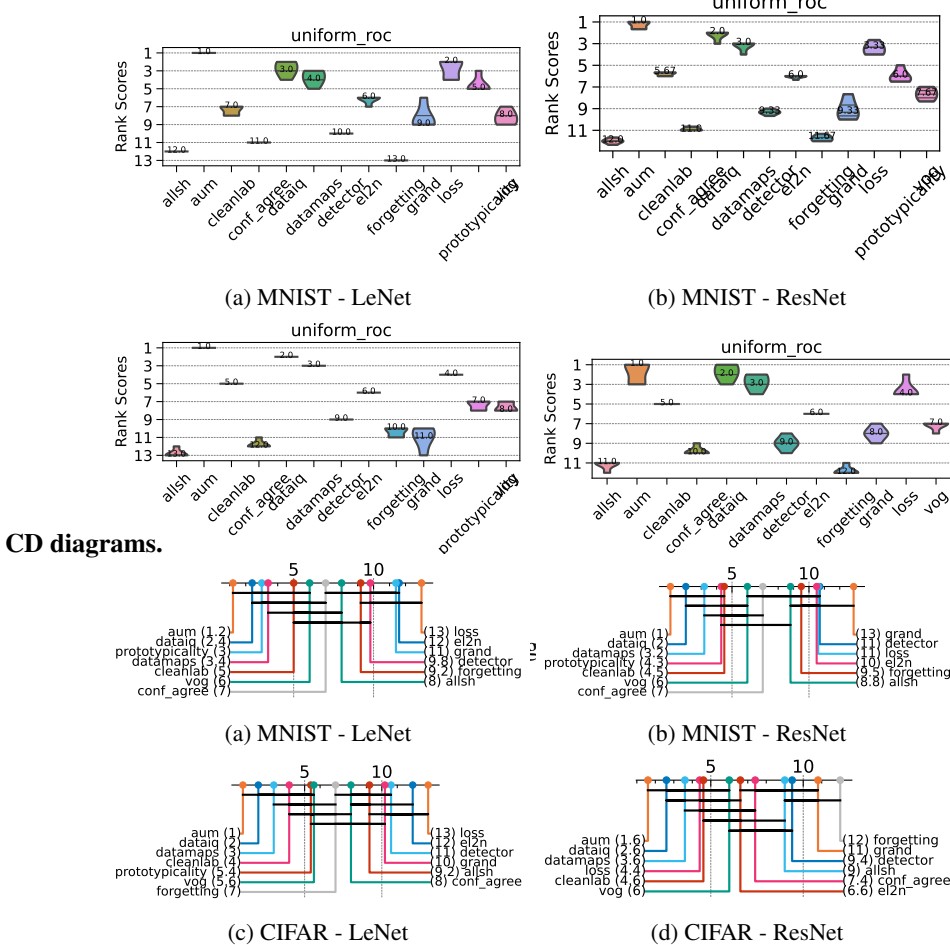

(a) MNIST - LeNet

(b) MNIST - ResNet

(c) CIFAR - LeNet

(d) CIFAR - ResNet

**CD diagrams.**

(a) MNIST - LeNet

(b) MNIST - ResNet

(c) CIFAR - LeNet

(d) CIFAR - ResNet

Figure 40: Uniform mislabeling - AUPRC. Critical difference diagrams. Black lines connect methods that are not statistically significantly different.

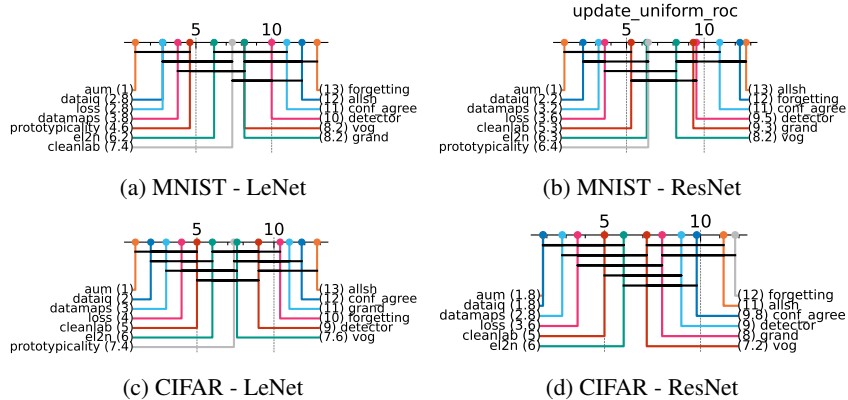

(a) MNIST - LeNet          (b) MNIST - ResNet

(c) CIFAR - LeNet          (d) CIFAR - ResNet

Figure 41: Uniform mislabeling - AUROC. Critical difference diagrams. Black lines connect methods that are not statistically significantly different.

**Matrix compare**

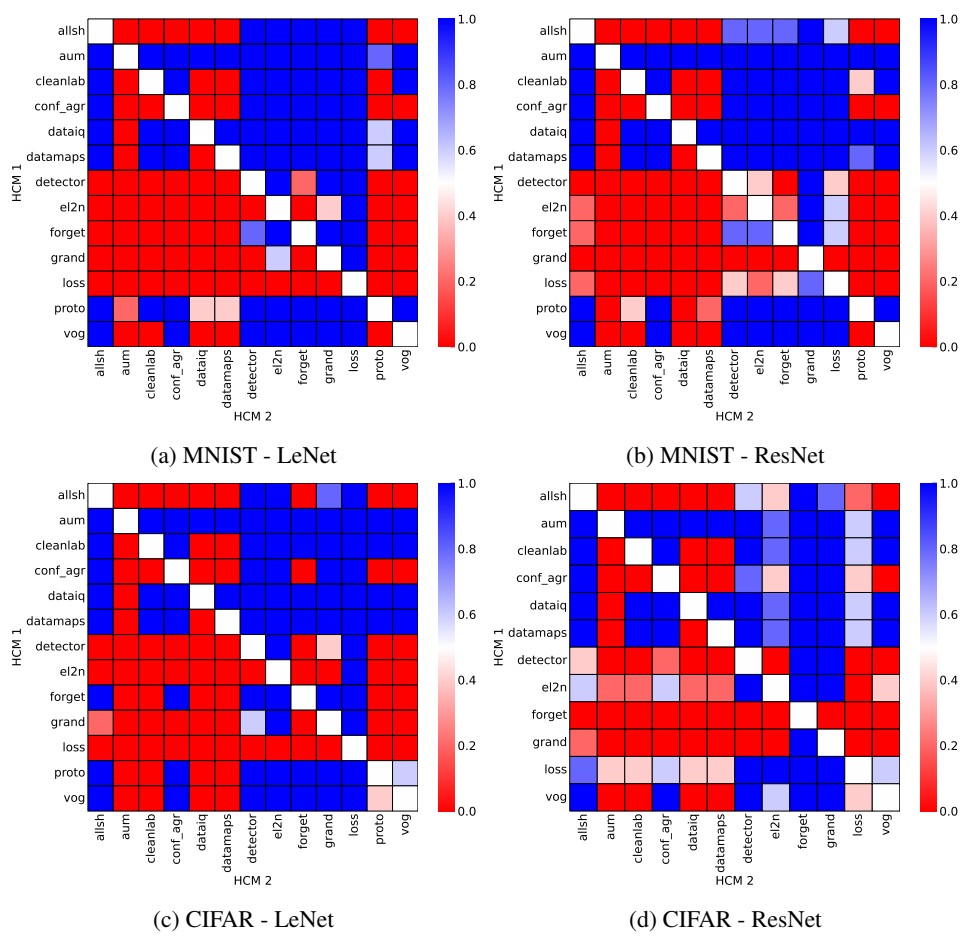

(a) MNIST - LeNet          (b) MNIST - ResNet

(c) CIFAR - LeNet          (d) CIFAR - ResNet

Figure 42: Uniform mislabeling. Head-to-head comparison matrix assessing for runs when a certain HCM outperforms another. i.e. when y-axis (HCM 1) beats x-axis competitor (HCM 2).

### D.5.5 NEAR OOD: COVARIATE

**Heatmaps.**

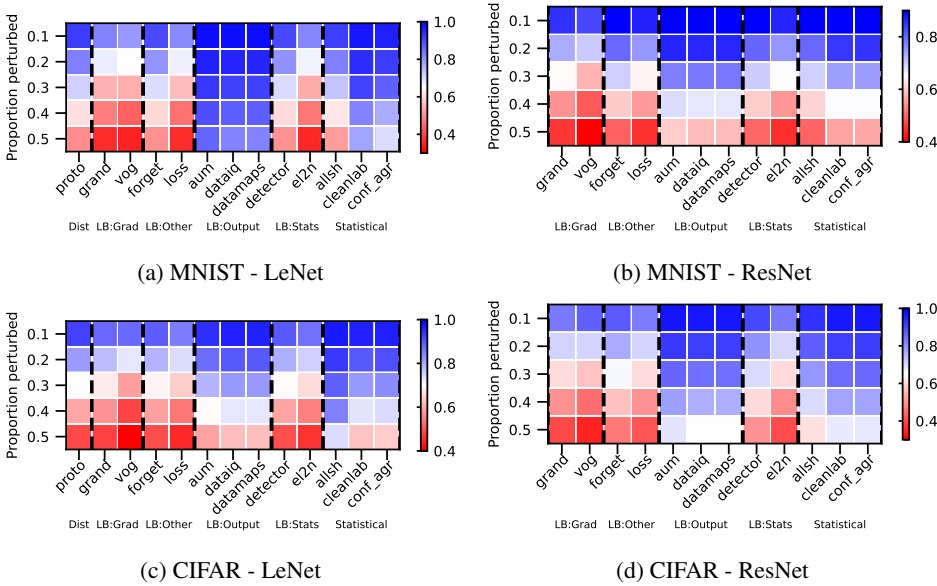

(a) MNIST - LeNet

(b) MNIST - ResNet

(c) CIFAR - LeNet

(d) CIFAR - ResNet

Figure 43: Near OOD: Covariate - AUPRC. We vary the proportion perturbed. i.e. proportion of hard examples. Blue is better, red worse.

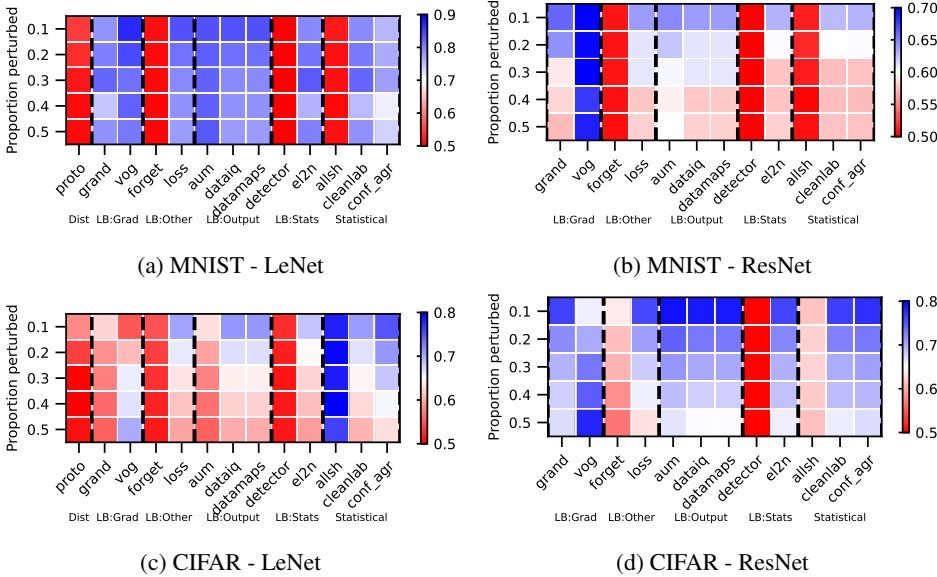

(a) MNIST - LeNet

(b) MNIST - ResNet

(c) CIFAR - LeNet

(d) CIFAR - ResNet

Figure 44: Near OOD: Covariate - AUROC. We vary the proportion perturbed. i.e. proportion of hard examples. Blue is better, red worse.

**Rankings.**

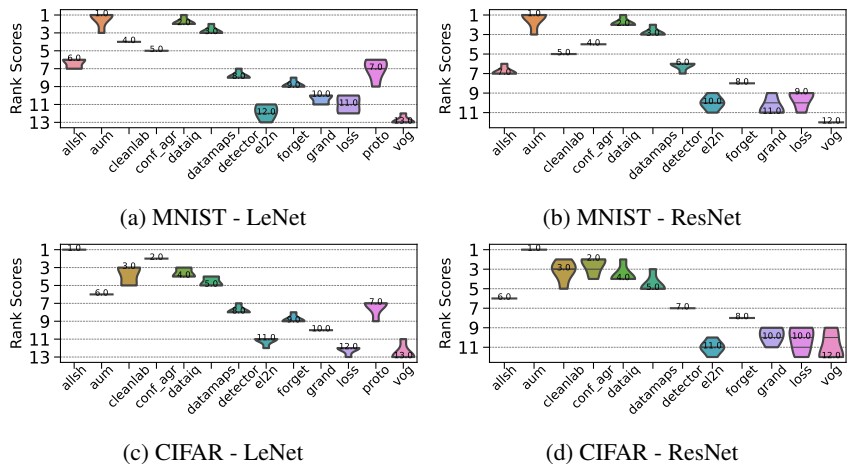

Figure 45: Near OOD: Covariate - AUPRC. Ranking of HCMs.

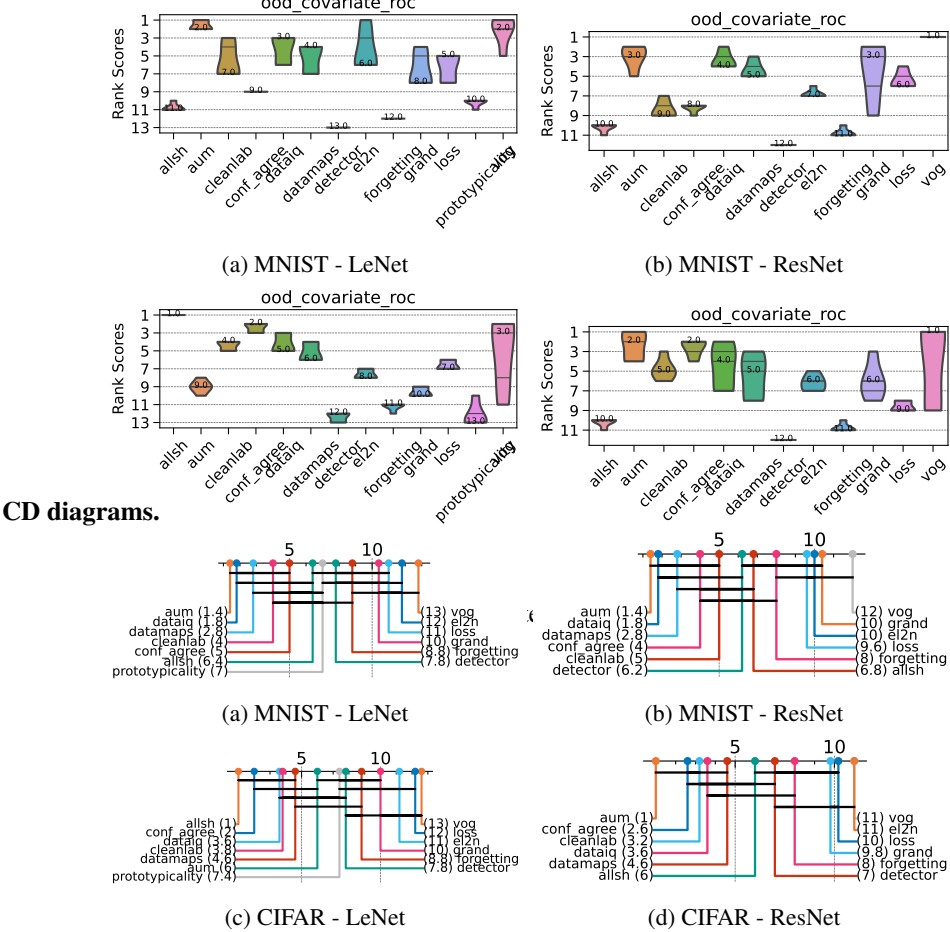

**CD diagrams.**

Figure 47: Near OOD: Covariate - AUPRC. Critical difference diagrams. Black lines connect methods that are not statistically significantly different.

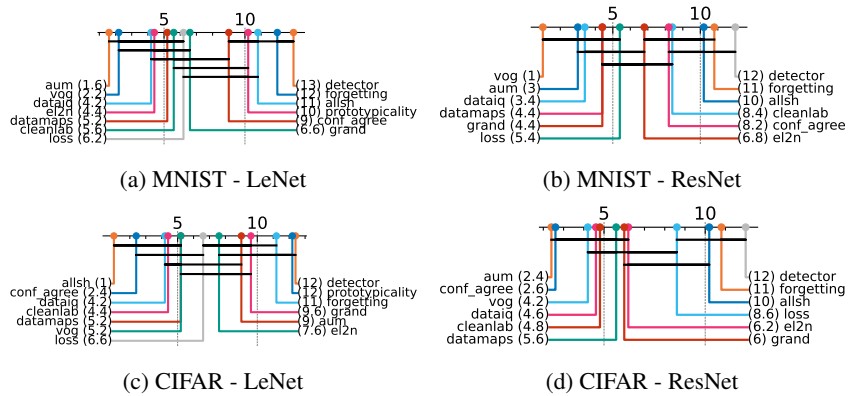

(a) MNIST - LeNet       (b) MNIST - ResNet

(c) CIFAR - LeNet       (d) CIFAR - ResNet

Figure 48: Near OOD: Covariate - AUROC. Critical difference diagrams. Black lines connect methods that are not statistically significantly different.

**Matrix compare**

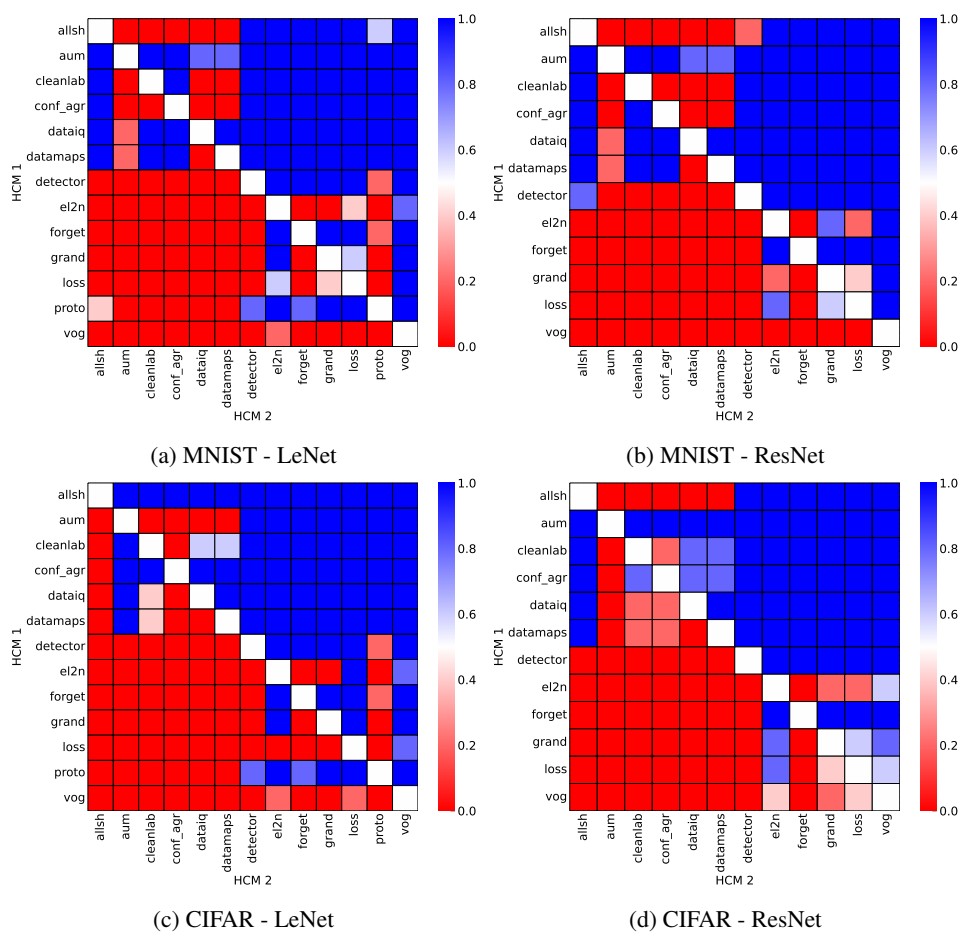

(a) MNIST - LeNet       (b) MNIST - ResNet

(c) CIFAR - LeNet       (d) CIFAR - ResNet

Figure 49: Near OOD: Covariate. Head-to-head comparison matrix assessing for runs when a certain HCM outperforms another. i.e. when y-axis (HCM 1) beats x-axis competitor (HCM 2).

### D.5.6 NEAR OOD: DOMAIN

**Heatmaps.**

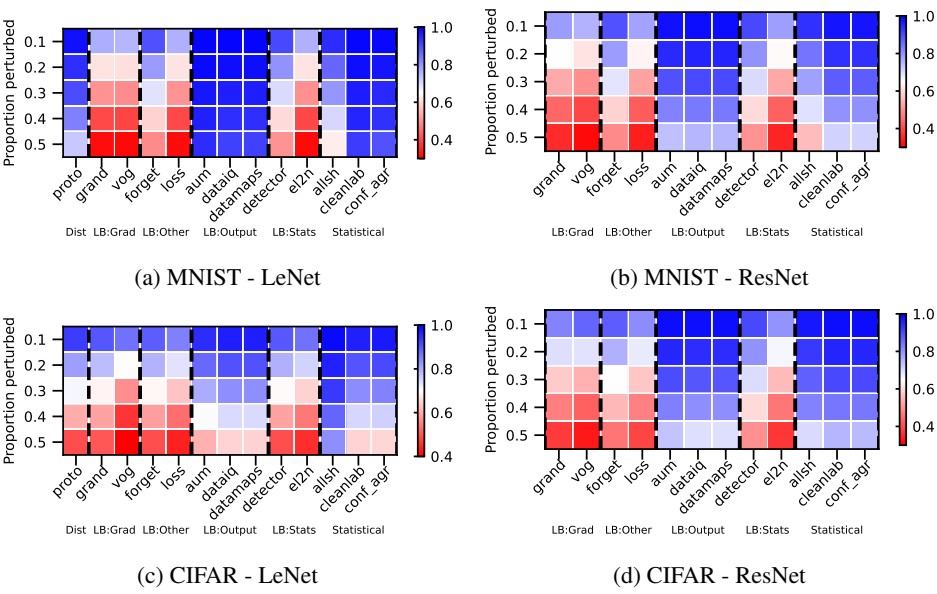

(a) MNIST - LeNet

(b) MNIST - ResNet

(c) CIFAR - LeNet

(d) CIFAR - ResNet

Figure 50: Near OOD: Domain - AUPRC. We vary the proportion perturbed. i.e. proportion of hard examples. Blue is better, red worse.

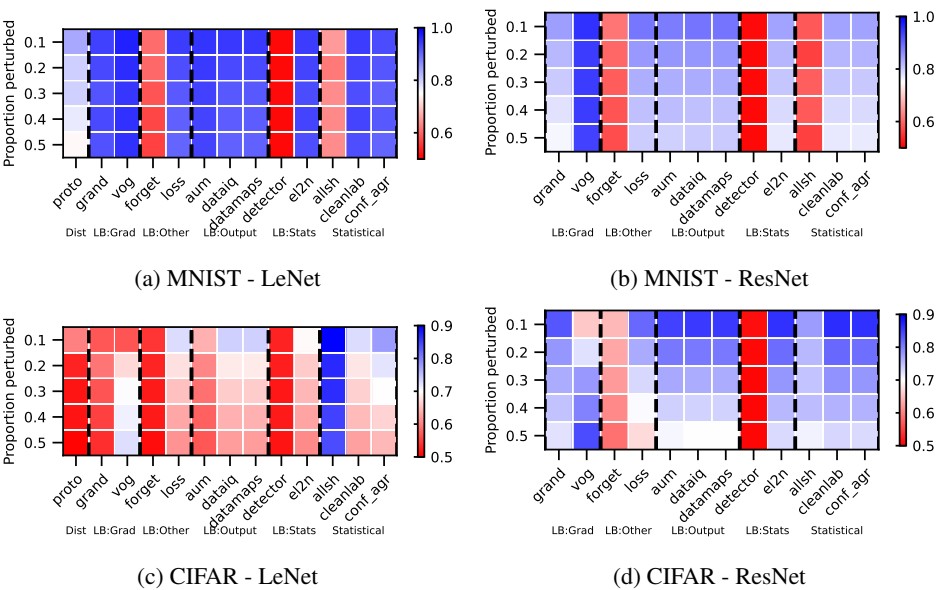

(a) MNIST - LeNet

(b) MNIST - ResNet

(c) CIFAR - LeNet

(d) CIFAR - ResNet

Figure 51: Near OOD: Domain - AUROC. We vary the proportion perturbed. i.e. proportion of hard examples. Blue is better, red worse.

**Rankings.**

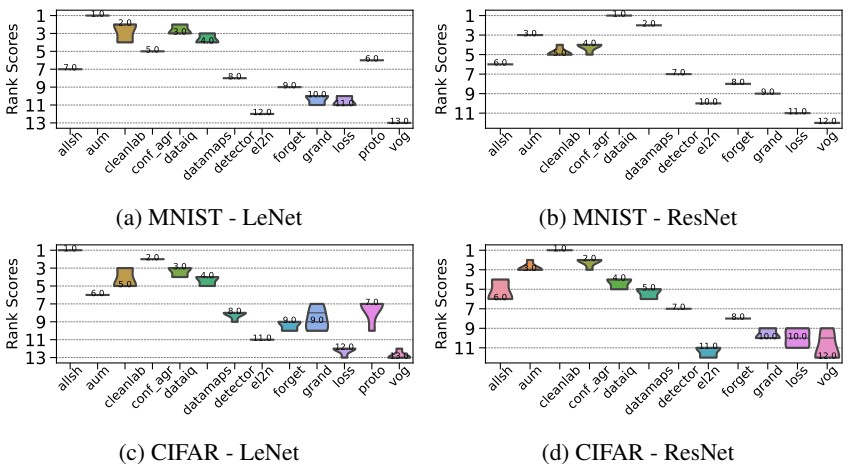

Figure 52: Near OOD: Domain - AUPRC. Ranking of HCMs.

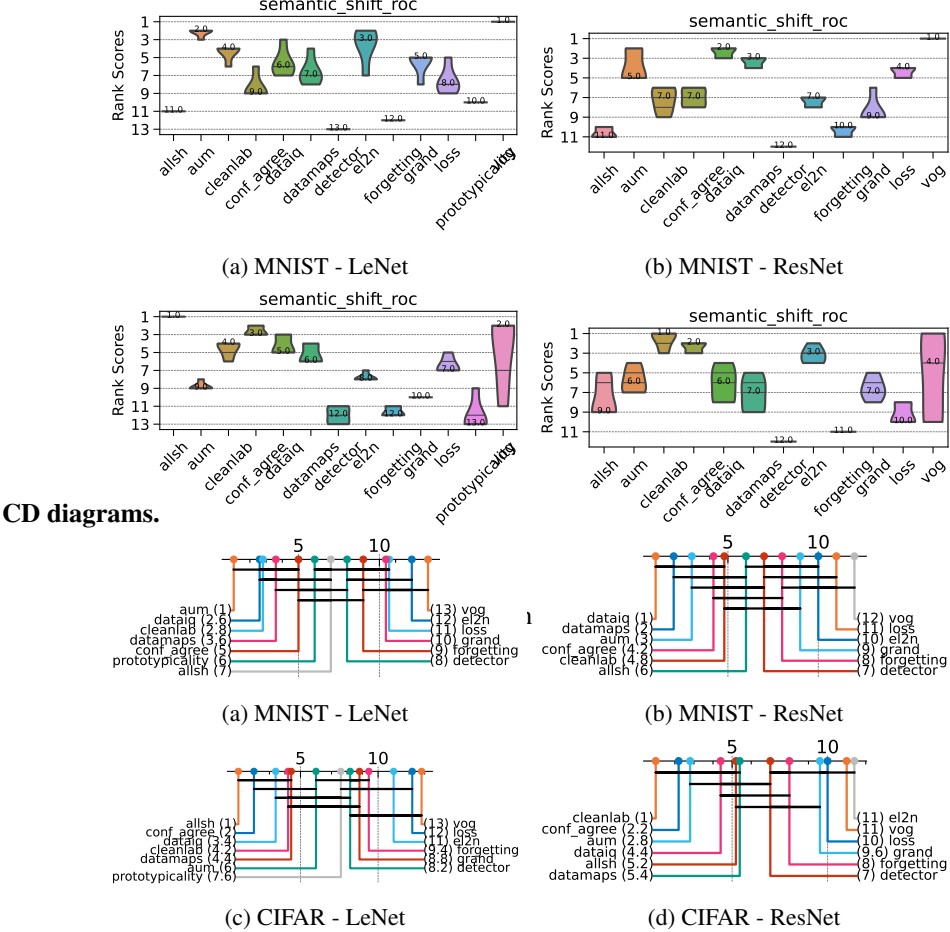

**CD diagrams.**

Figure 54: Near OOD: Domain - AUPRC. Critical difference diagrams. Black lines connect methods that are not statistically significantly different.

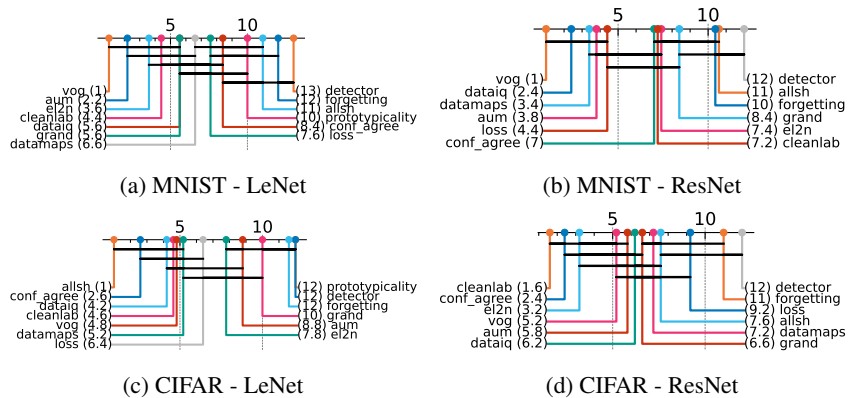

Figure 55: Near OOD: Domain - AUROC. Critical difference diagrams. Black lines connect methods that are not statistically significantly different.

**Matrix compare**

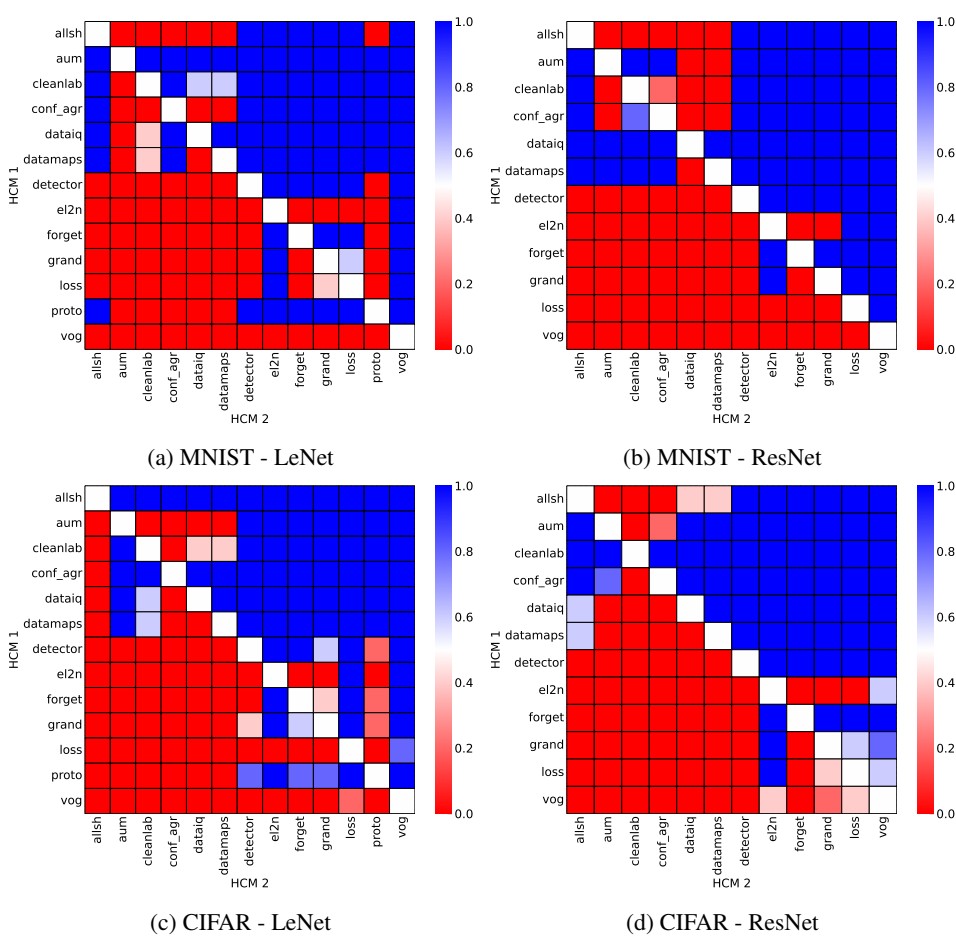

Figure 56: Near OOD: Domain. Head-to-head comparison matrix assessing for runs when a certain HCM outperforms another. i.e. when y-axis (HCM 1) beats x-axis competitor (HCM 2).

### D.5.7 FAR OOD

**Heatmaps.**

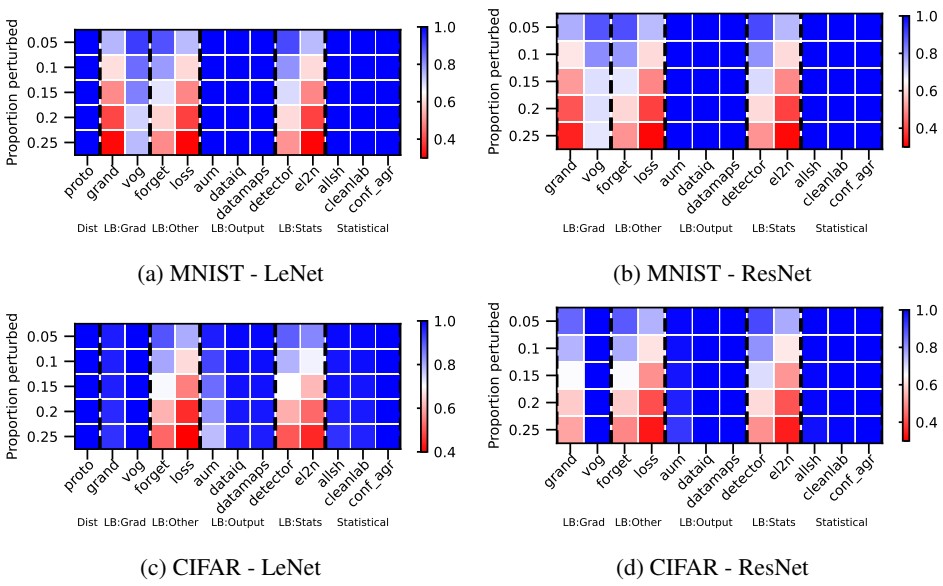

Figure 57: Far OoD - AUPRC. We vary the proportion perturbed. i.e. proportion of hard examples. Blue is better, red worse.

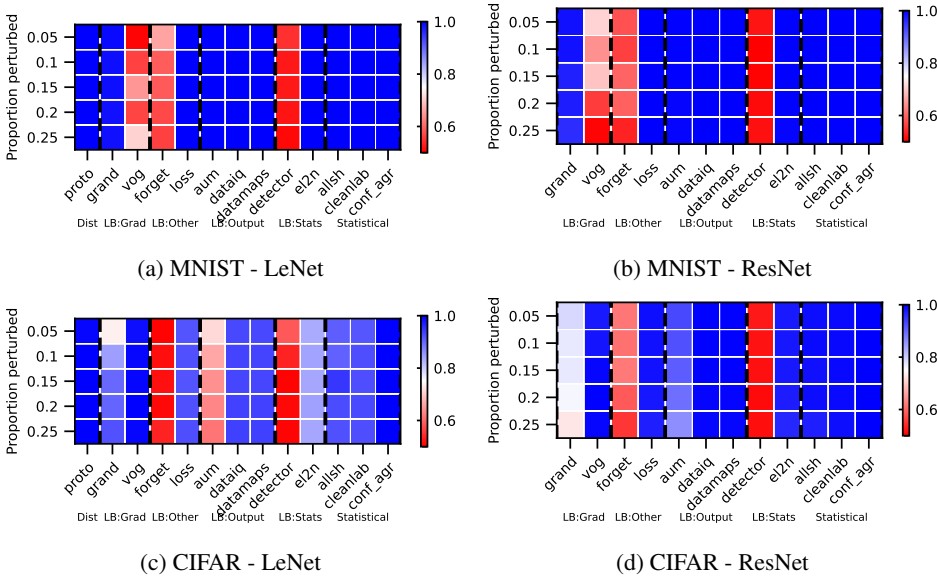

Figure 58: Far OoD - AUROC. We vary the proportion perturbed. i.e. proportion of hard examples. Blue is better, red worse.

**Rankings.**

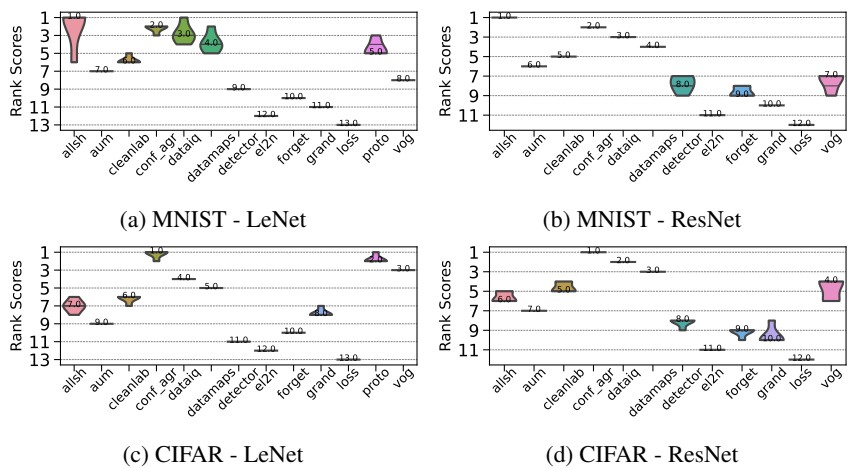

Figure 59: Far OoD - AUPRC. Ranking of HCMs.

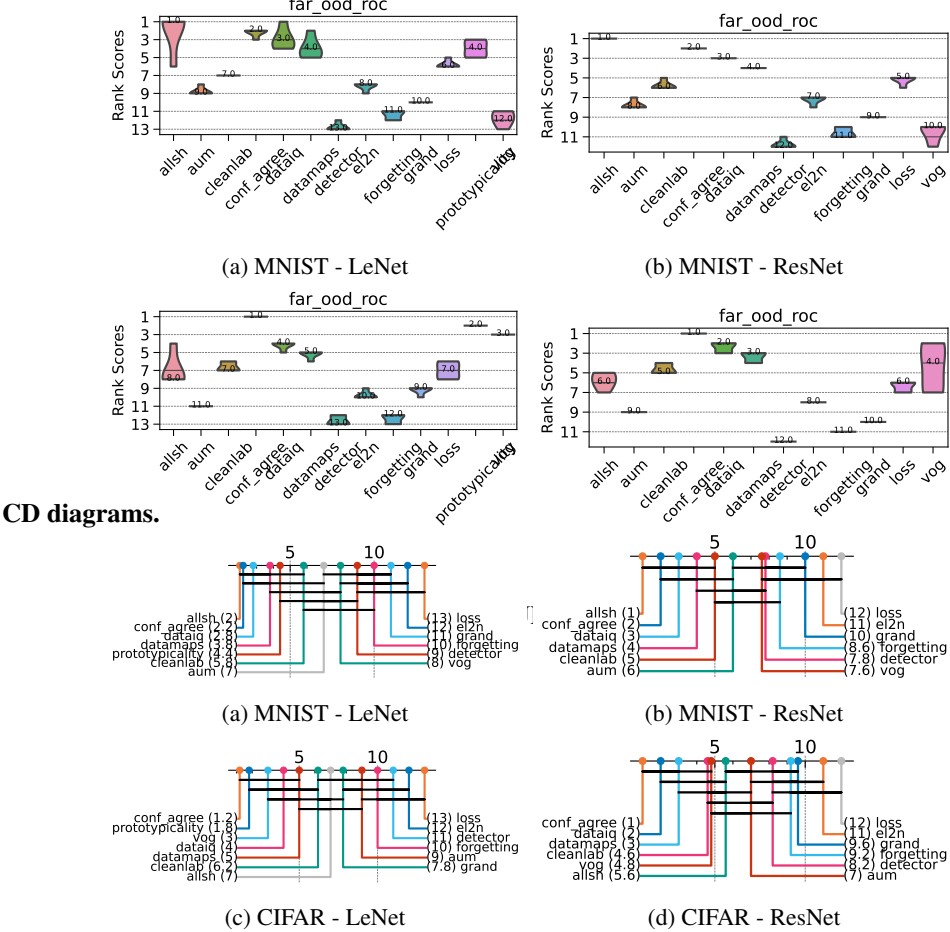

**CD diagrams.**

Figure 61: Far OoD - AUPRC. Critical difference diagrams. Black lines connect methods that are not statistically significantly different.

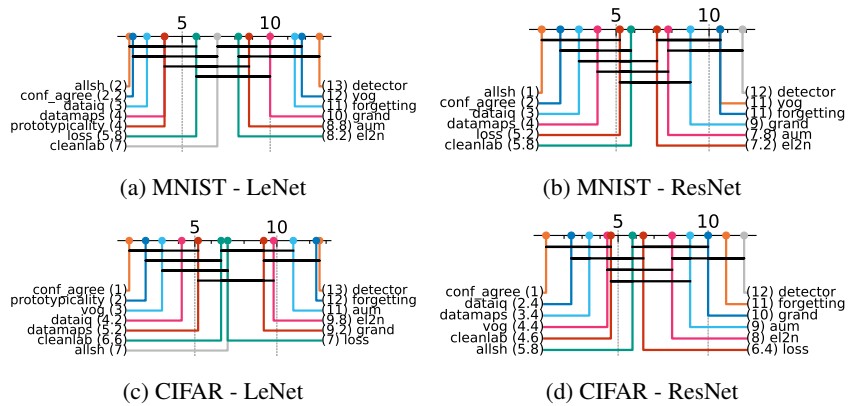

(a) MNIST - LeNet (b) MNIST - ResNet

(c) CIFAR - LeNet (d) CIFAR - ResNet

Figure 62: Far OoD - AUROC. Critical difference diagrams. Black lines connect methods that are not statistically significantly different.

**Matrix compare**

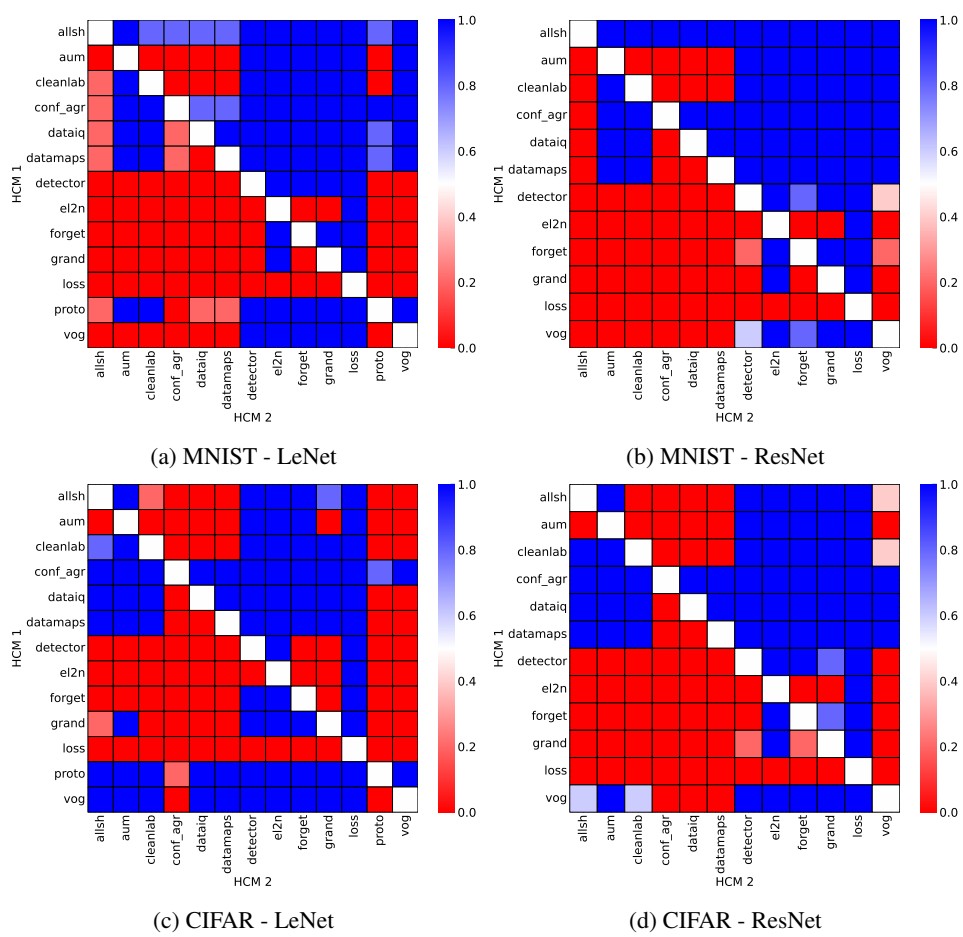

(a) MNIST - LeNet (b) MNIST - ResNet

(c) CIFAR - LeNet (d) CIFAR - ResNet

Figure 63: Far OoD. Head-to-head comparison matrix assessing for runs when a certain HCM outperforms another. i.e. when y-axis (HCM 1) beats x-axis competitor (HCM 2).

### D.5.8 ATYPICAL: CROP SHIFT

**Heatmaps.**

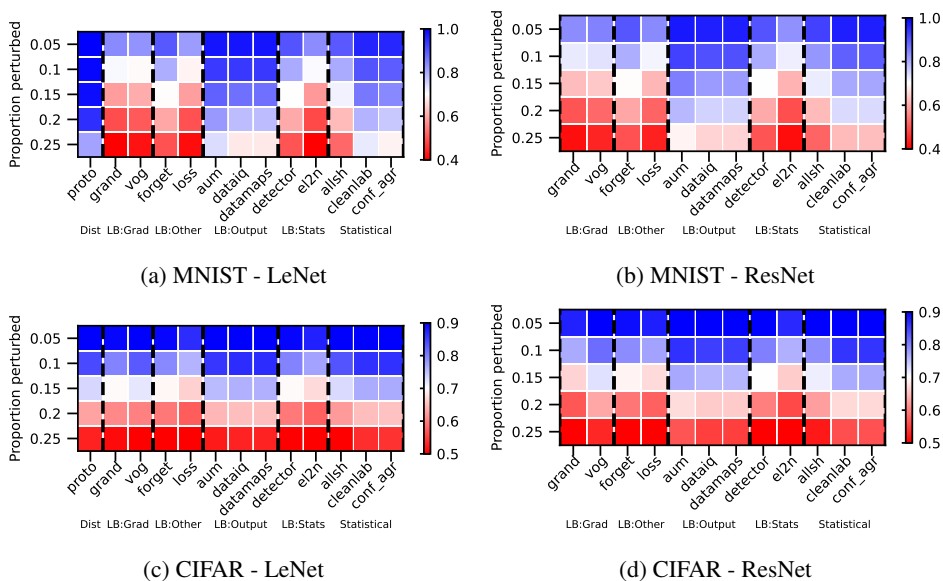

Figure 64: Atypical: Crop Shift - AUPRC. We vary the proportion perturbed. i.e. proportion of hard examples. Blue is better, red worse.

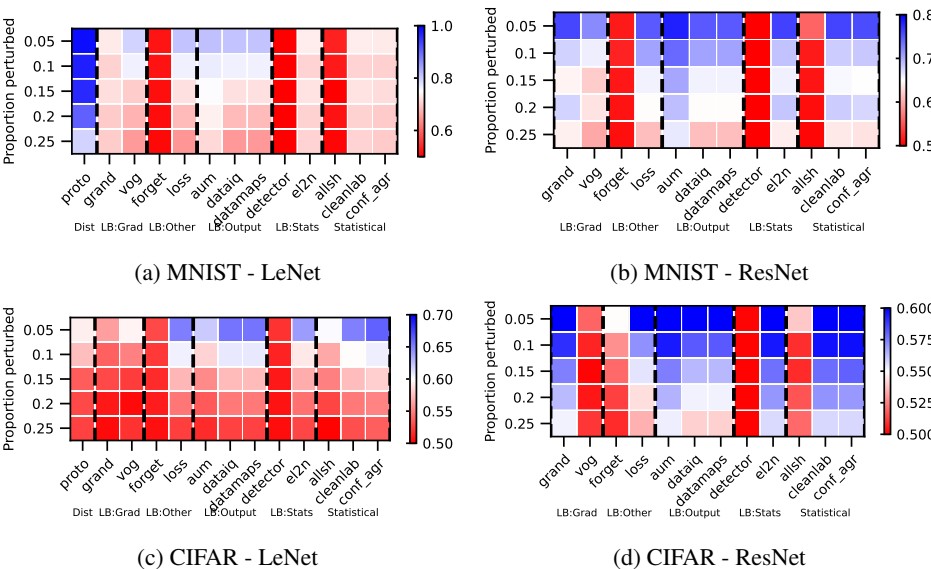

Figure 65: Atypical: Crop Shift - AUROC. We vary the proportion perturbed. i.e. proportion of hard examples. Blue is better, red worse.

**Rankings.**

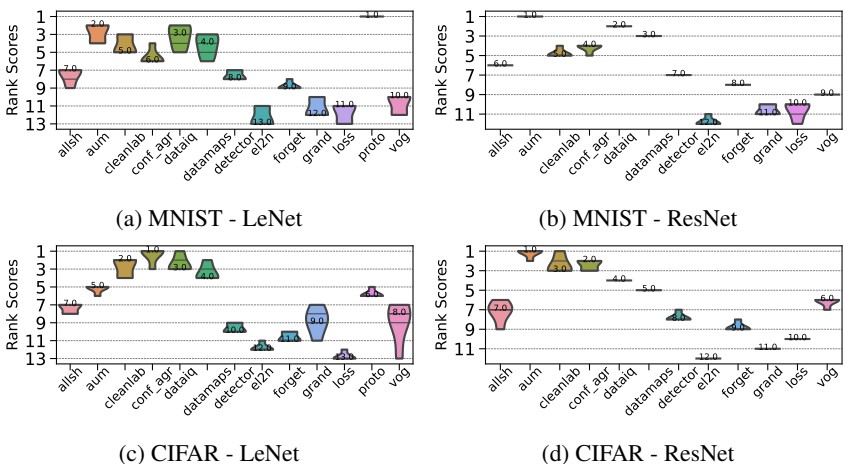

(a) MNIST - LeNet

(b) MNIST - ResNet

(c) CIFAR - LeNet

(d) CIFAR - ResNet

Figure 66: Atypical: Crop Shift - AUPRC. Ranking of HCMs.

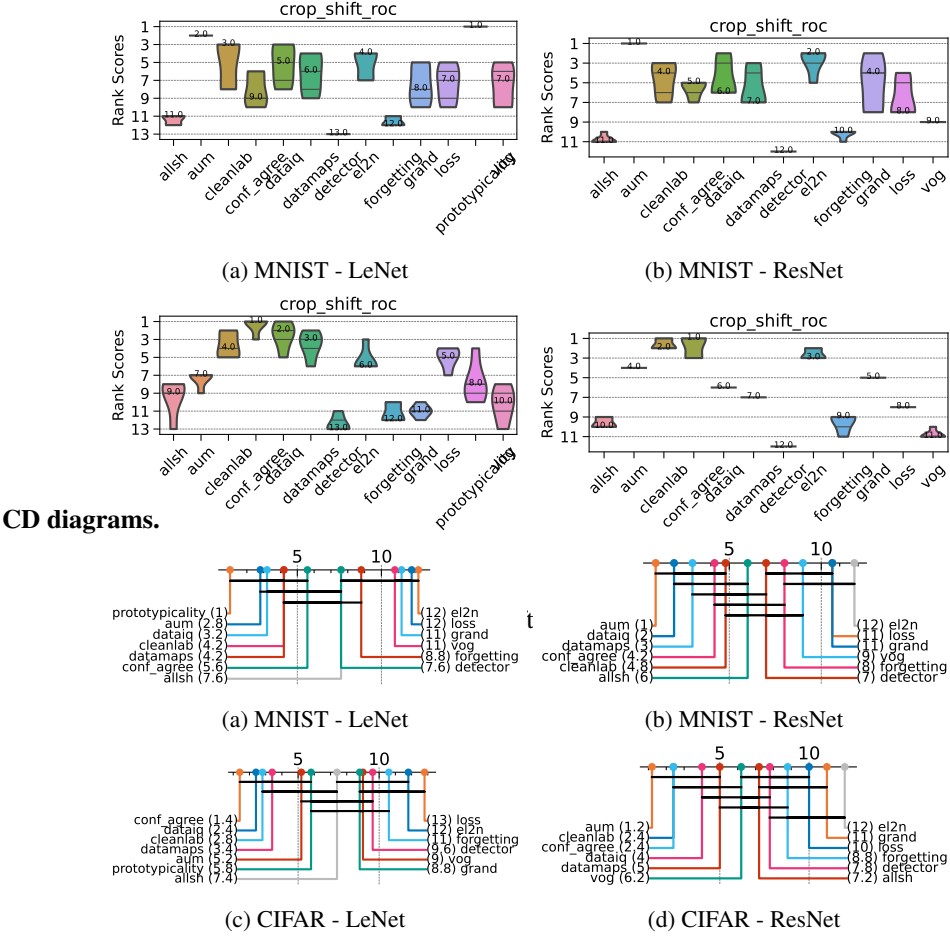

(a) MNIST - LeNet

(b) MNIST - ResNet

(c) CIFAR - LeNet

(d) CIFAR - ResNet

**CD diagrams.**

(a) MNIST - LeNet

(b) MNIST - ResNet

(c) CIFAR - LeNet

(d) CIFAR - ResNet

Figure 68: Atypical: Crop Shift - AUPRC. Critical difference diagrams. Black lines connect methods that are not statistically significantly different.

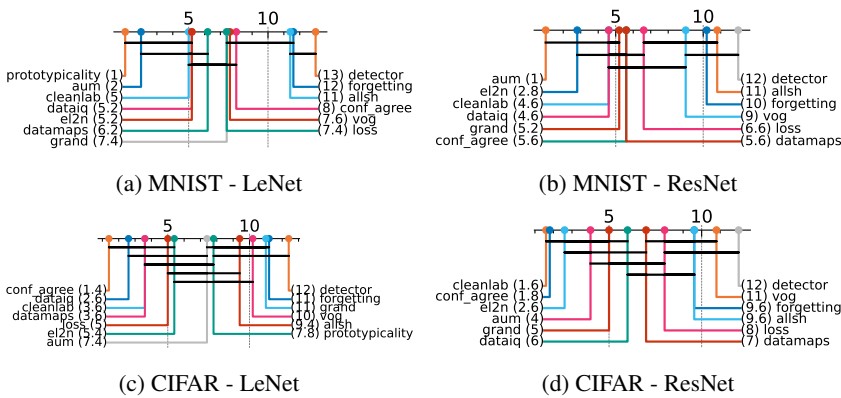

(a) MNIST - LeNet

(b) MNIST - ResNet

(c) CIFAR - LeNet

(d) CIFAR - ResNet

Figure 69: Atypical: Crop Shift - AUROC. Critical difference diagrams. Black lines connect methods that are not statistically significantly different.

## Matrix compare

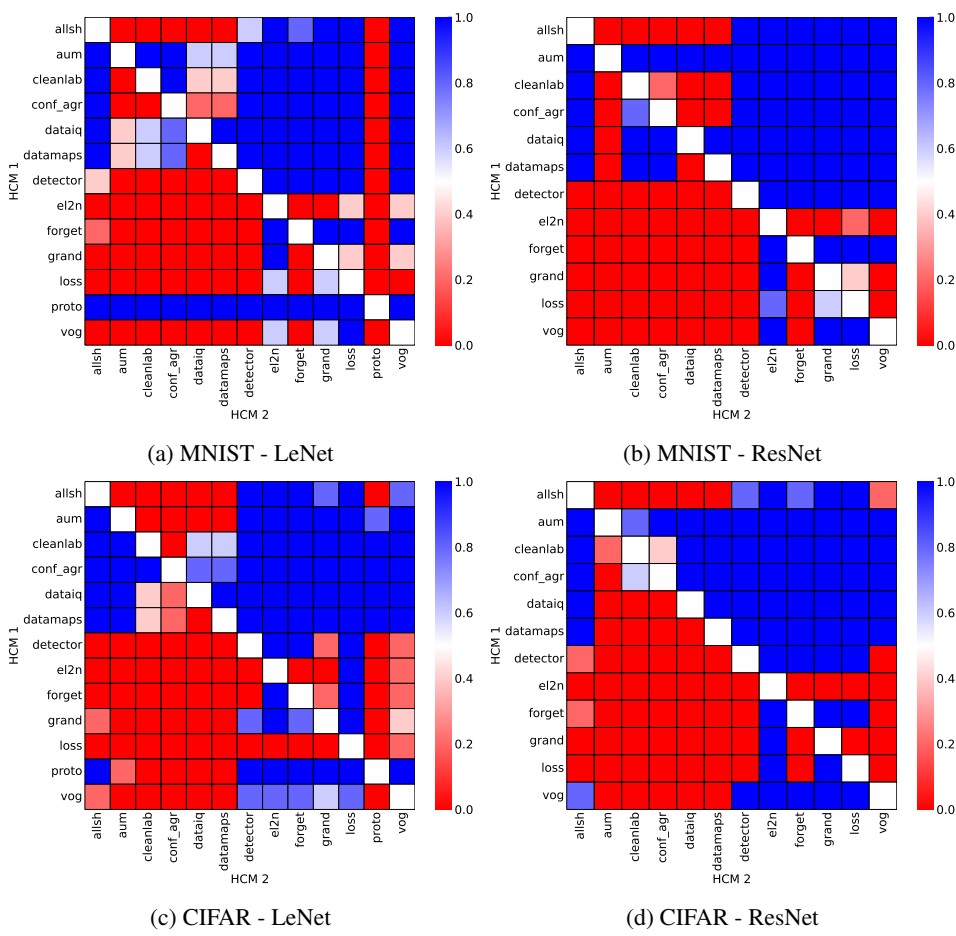

(a) MNIST - LeNet

(b) MNIST - ResNet

(c) CIFAR - LeNet

(d) CIFAR - ResNet

Figure 70: Atypical: Crop Shift. Head-to-head comparison matrix assessing for runs when a certain HCM outperforms another. i.e. when y-axis (HCM 1) beats x-axis competitor (HCM 2).

### D.5.9 ATYPICAL: ZOOM SHIFT

**Heatmaps.**

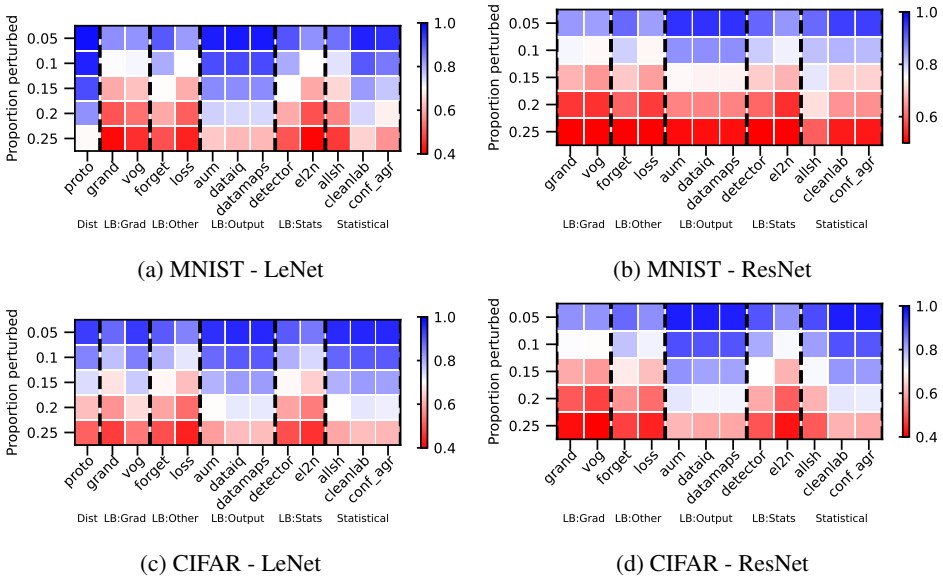

Figure 71: Atypical: Zoom Shift - AUPRC. We vary the proportion perturbed. i.e. proportion of hard examples. Blue is better, red worse.

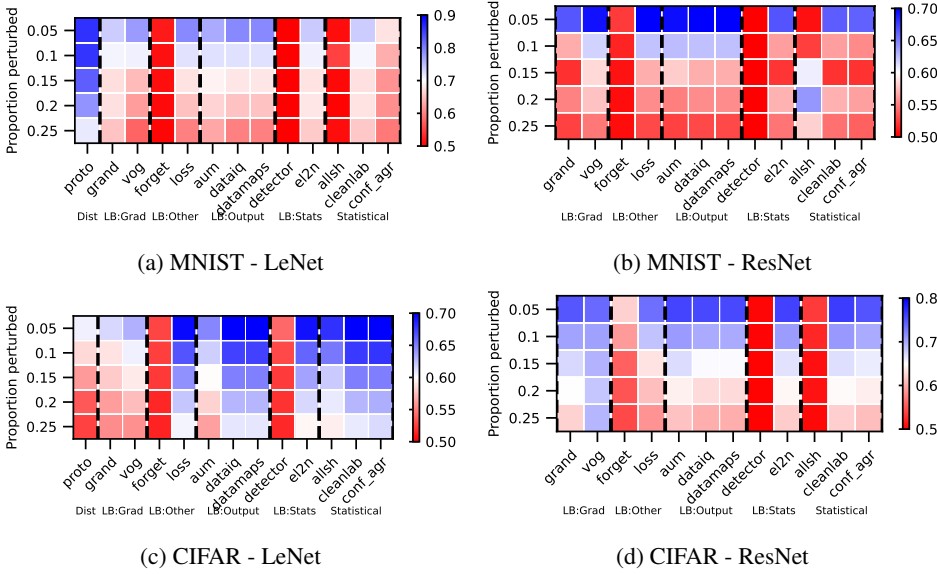

Figure 72: Atypical: Zoom Shift - AUROC. We vary the proportion perturbed. i.e. proportion of hard examples. Blue is better, red worse.

**Rankings.**

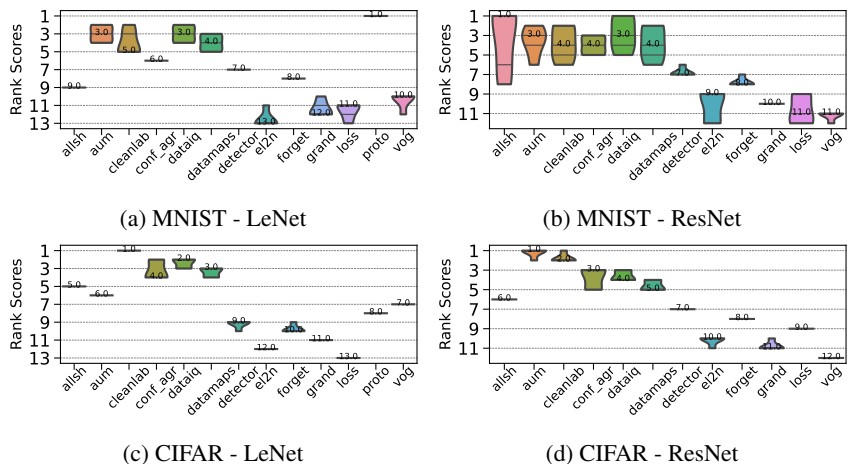

(a) MNIST - LeNet

(b) MNIST - ResNet

(c) CIFAR - LeNet

(d) CIFAR - ResNet

Figure 73: Atypical: Zoom Shift - AUPRC. Ranking of HCMs.

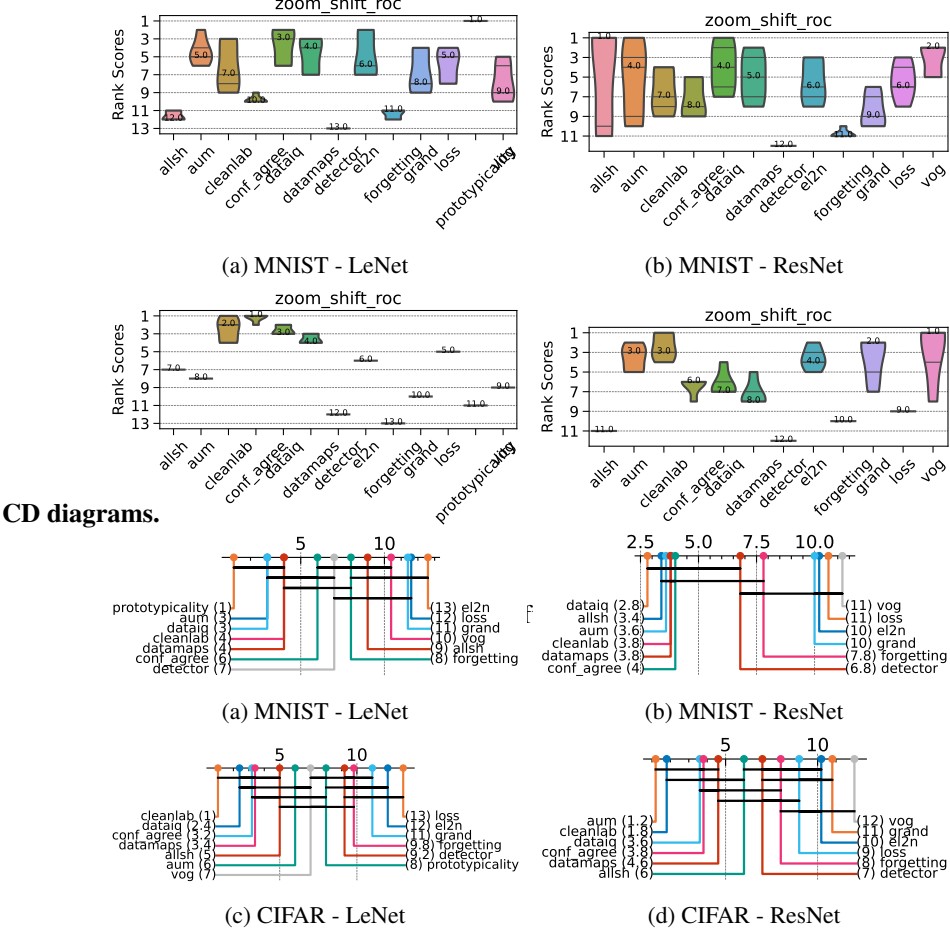

(a) MNIST - LeNet

(b) MNIST - ResNet

(c) CIFAR - LeNet

(d) CIFAR - ResNet

**CD diagrams.**

(a) MNIST - LeNet

(b) MNIST - ResNet

(c) CIFAR - LeNet

(d) CIFAR - ResNet

Figure 75: Atypical: Zoom Shift - AUPRC. Critical difference diagrams. Black lines connect methods that are not statistically significantly different.

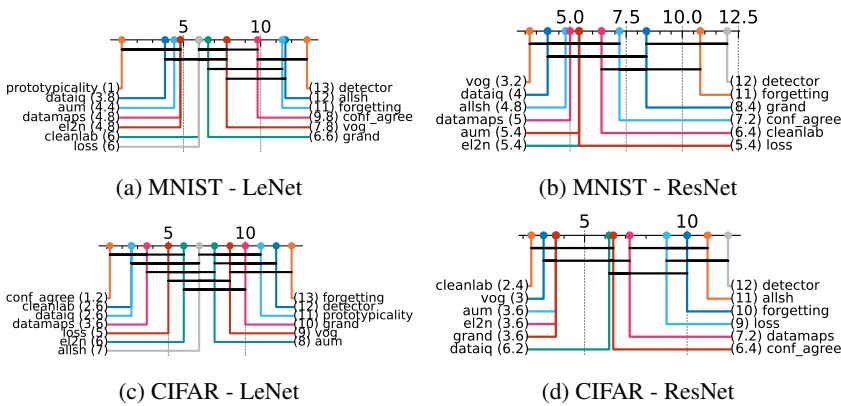

Figure 76: Atypical: Zoom Shift - AUROC. Critical difference diagrams. Black lines connect methods that are not statistically significantly different.

**Matrix compare**

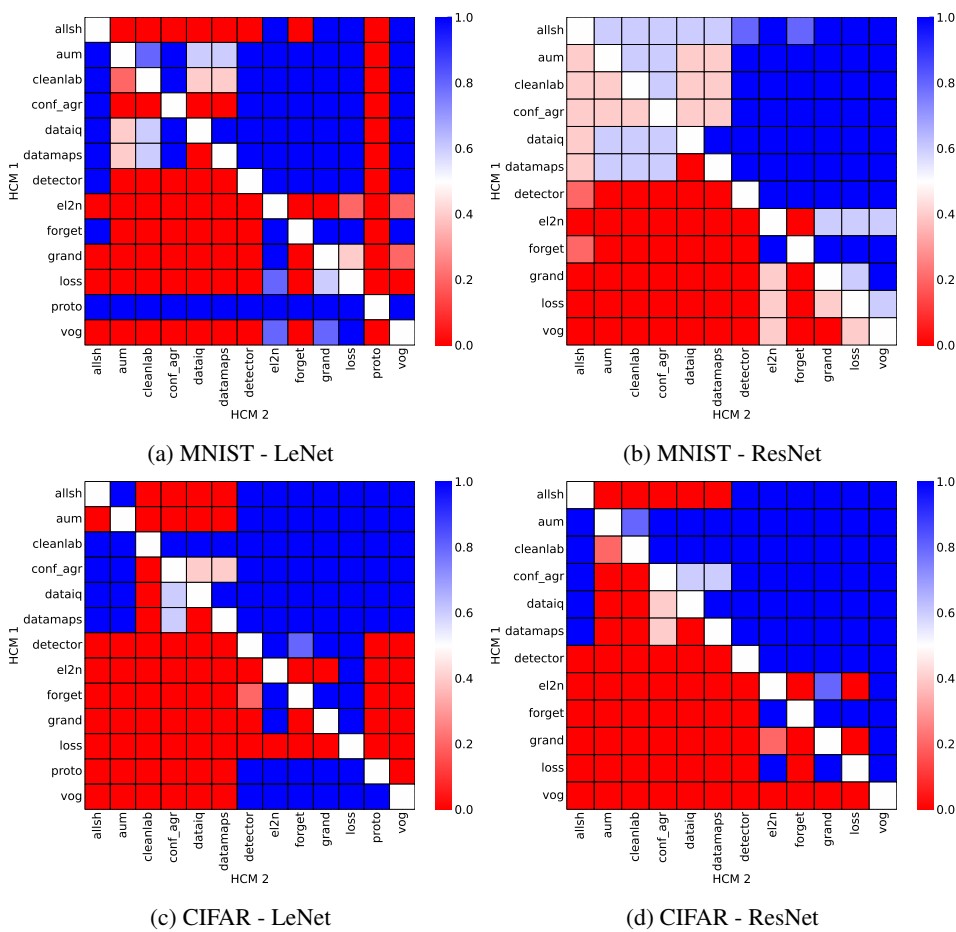

Figure 77: Atypical: Zoom Shift. Head-to-head comparison matrix assessing for runs when a certain HCM outperforms another. i.e. when y-axis (HCM 1) beats x-axis competitor (HCM 2).

## D.6   INDIVIDUAL RESULTS - TABULAR

### D.6.1   MISLABELING: UNIFORM

**Heatmaps.**

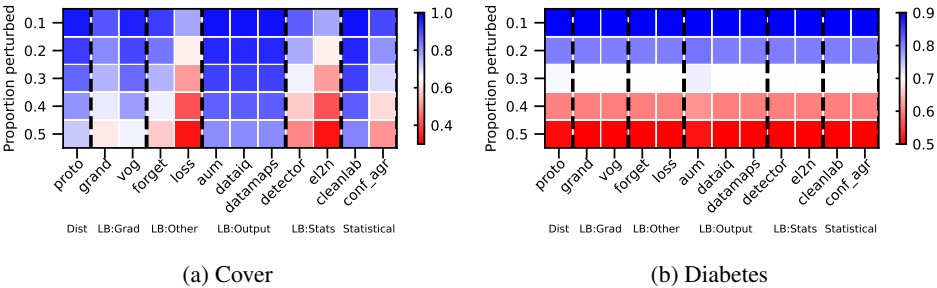

(a) Cover

(b) Diabetes

Figure 78: TABULAR: Uniform mislabeling - AUPRC. We vary the proportion perturbed. i.e. proportion of hard examples. Blue is better, red worse.

**Rankings.**

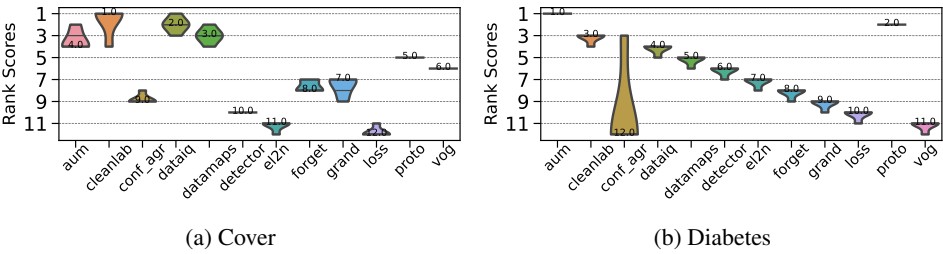

(a) Cover

(b) Diabetes

Figure 79: TABULAR: Uniform mislabeling - AUPRC. Ranking of HCMs.

**CD diagrams.**

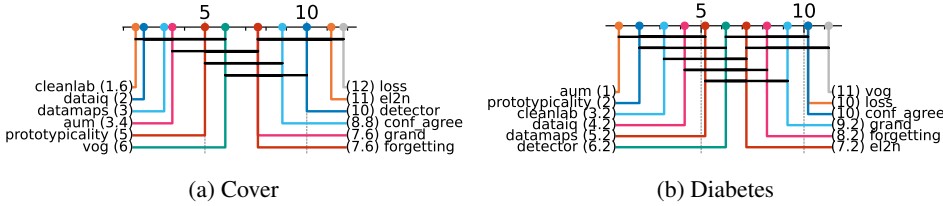

(a) Cover

(b) Diabetes

Figure 80: TABULAR: Uniform mislabeling - AUPRC. Critical difference diagrams. Black lines connect methods that are not statistically significantly different.

### D.6.2 MISLABELING: ASYMMETRIC

**Heatmaps.**

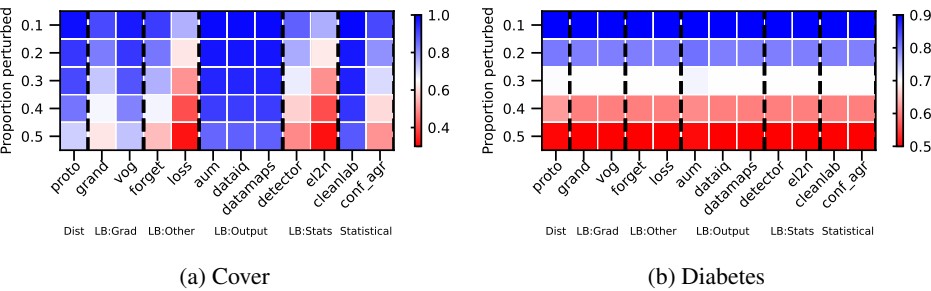

(a) Cover
(b) Diabetes

Figure 81: TABULAR: asymmetric mislabeling - AUPRC. We vary the proportion perturbed. i.e. proportion of hard examples. Blue is better, red worse.

**Rankings.**

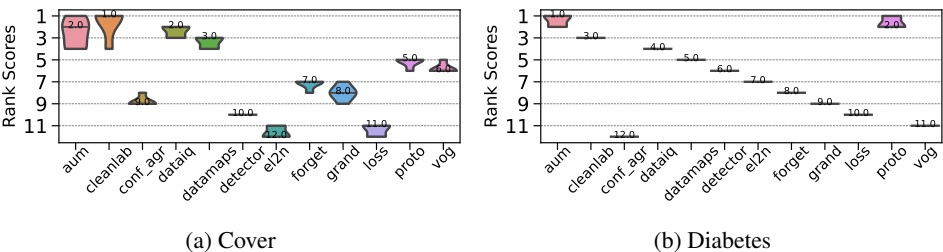

(a) Cover
(b) Diabetes

Figure 82: TABULAR: asymmetric mislabeling - AUPRC. Ranking of HCMs.

**CD diagrams.**

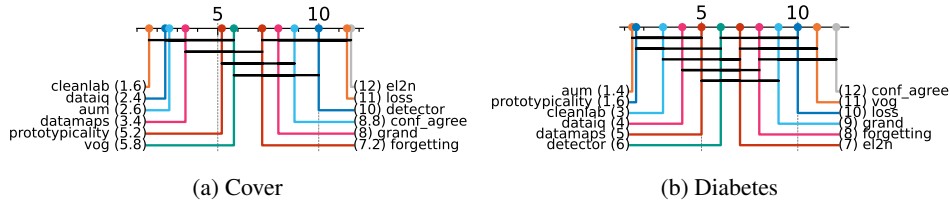

(a) Cover
(b) Diabetes

Figure 83: TABULAR: asymmetric mislabeling - AUPRC. Critical difference diagrams. Black lines connect methods that are not statistically significantly different.

### D.6.3 MISLABELING: INSTANCE

**Heatmaps.**

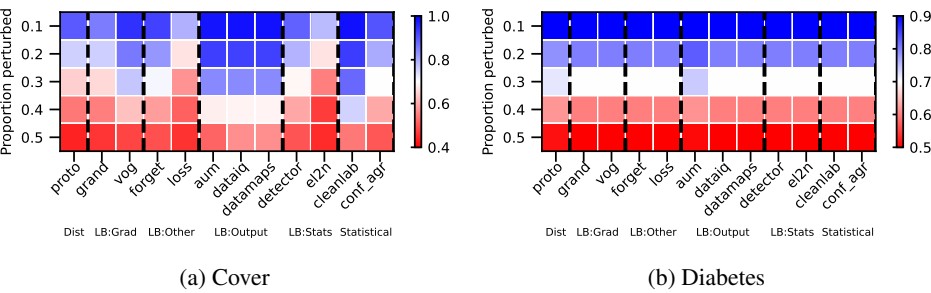

(a) Cover        (b) Diabetes

Figure 84: TABULAR: instance mislabeling - AUPRC. We vary the proportion perturbed. i.e. proportion of hard examples. Blue is better, red worse.

**Rankings.**

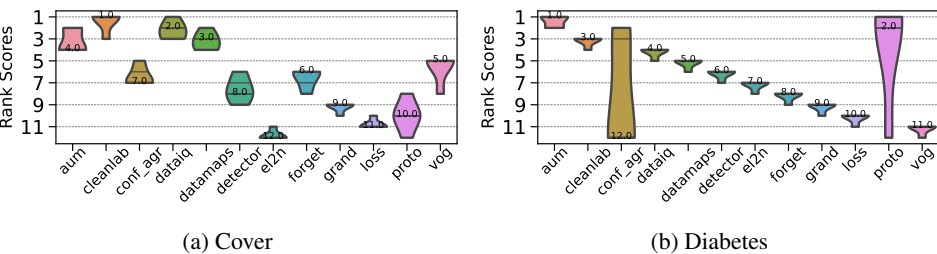

(a) Cover        (b) Diabetes

Figure 85: TABULAR: instance mislabeling - AUPRC. Ranking of HCMs.

**CD diagrams.**

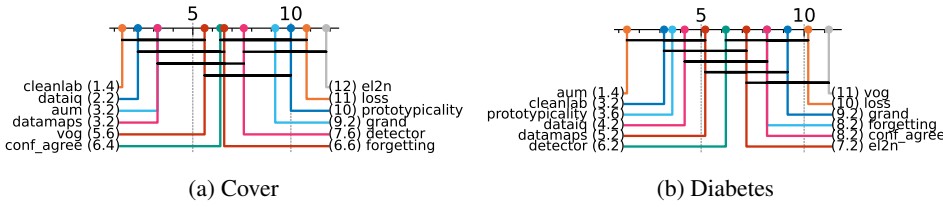

(a) Cover        (b) Diabetes

Figure 86: TABULAR: instance mislabeling - AUPRC. Critical difference diagrams. Black lines connect methods that are not statistically significantly different.

### D.6.4 Near OOD: Covariate

**Heatmaps.**

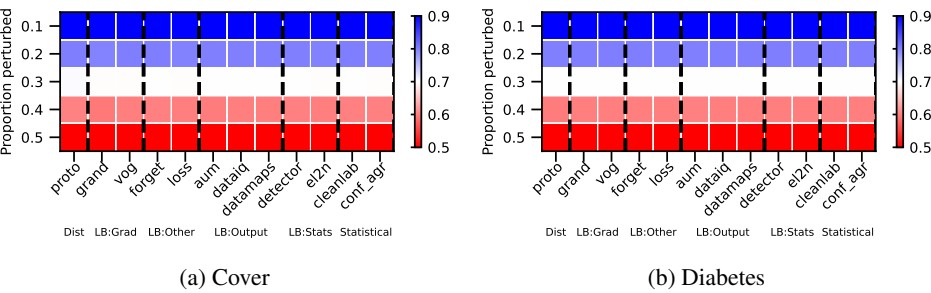

(a) Cover

(b) Diabetes

Figure 87: TABULAR: Near OOD - AUPRC. We vary the proportion perturbed. i.e. proportion of hard examples. Blue is better, red worse.

**Rankings.**

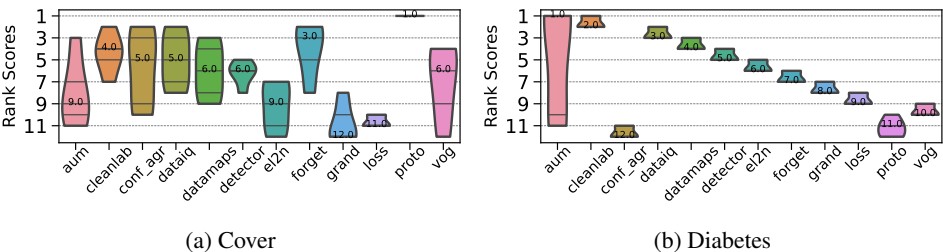

(a) Cover

(b) Diabetes

Figure 88: TABULAR: Near OOD- AUPRC. Ranking of HCMs.

**CD diagrams.**

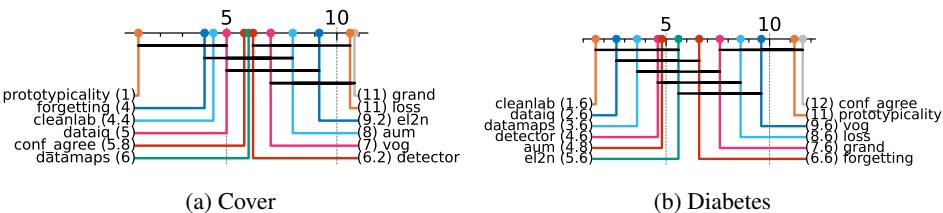

(a) Cover

(b) Diabetes

Figure 89: TABULAR: Near OOD - AUPRC. Critical difference diagrams. Black lines connect methods that are not statistically significantly different.

### D.6.5 FAR OOD

**Heatmaps.**

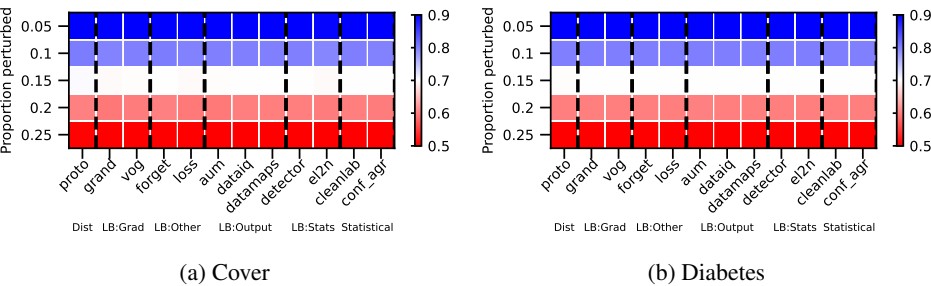

(a) Cover

(b) Diabetes

Figure 90: TABULAR: far ood - AUPRC. We vary the proportion perturbed. i.e. proportion of hard examples. Blue is better, red worse.

**Rankings.**

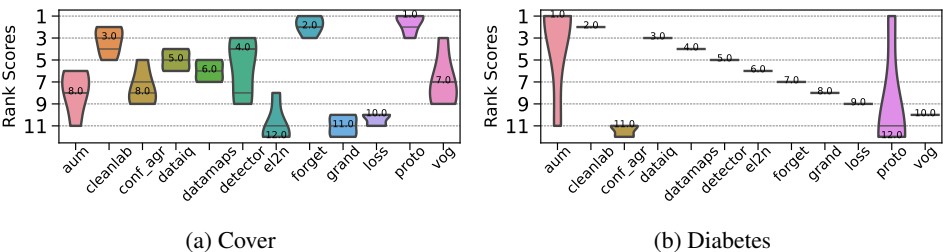

(a) Cover

(b) Diabetes

Figure 91: TABULAR: far ood - AUPRC. Ranking of HCMs.

**CD diagrams.**

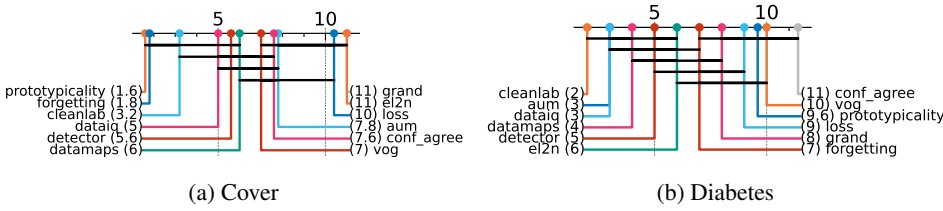

(a) Cover

(b) Diabetes

Figure 92: TABULAR: far ood - AUPRC. Critical difference diagrams. Black lines connect methods that are not statistically significantly different.

### D.6.6 ATYPICAL

**Heatmaps.**

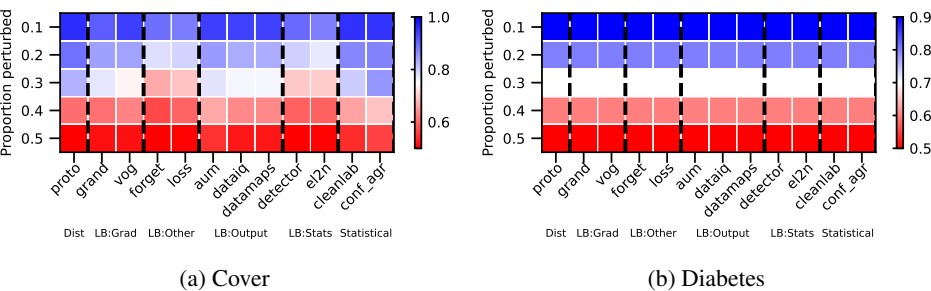

(a) Cover      (b) Diabetes

Figure 93: TABULAR: atypical - AUPRC. We vary the proportion perturbed. i.e. proportion of hard examples. Blue is better, red worse.

**Rankings.**

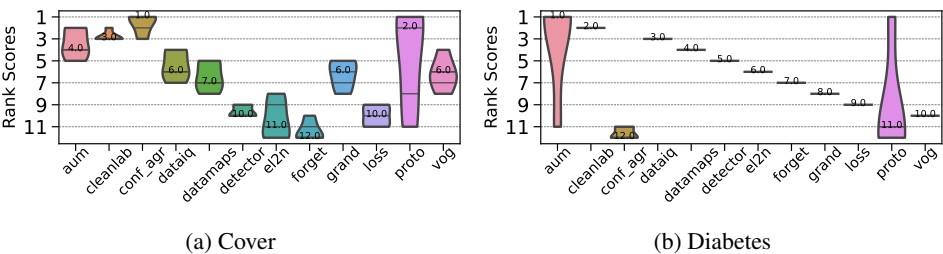

(a) Cover      (b) Diabetes

Figure 94: TABULAR: atypical - AUPRC. Ranking of HCMs.

**CD diagrams.**

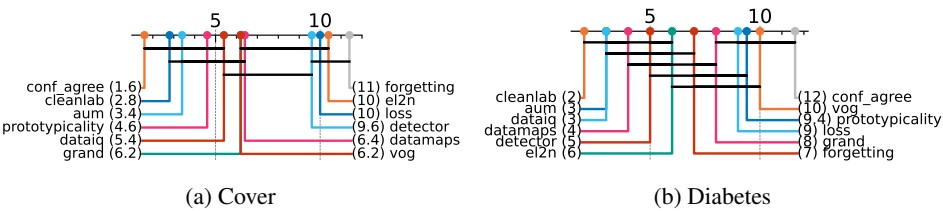

(a) Cover      (b) Diabetes

Figure 95: TABULAR: atypical - AUPRC. Critical difference diagrams. Black lines connect methods that are not statistically significantly different.

### D.7 ADDITIONAL RESULTS - MEDICAL X-RAY DATA

**Overview.** Our main experiments are conducted on the MNIST and CIFAR-10 datasets. The reason is that it is important to have "clean" datasets for the purposes of ensuring a good simulation environment. e.g. we do not want the original data to already have mislabeling. We now extend our analysis of HCMs to an additional medical imaging dataset from the NIH (Majkowska et al., 2020). These contain chest x-rays which have been audited by multiple expert radiologists to ensure they are "clean". We repeat our experiments from the main paper and show the results below.

We now analyze the results and show the findings about HCMs are retained for the X-Ray dataset.

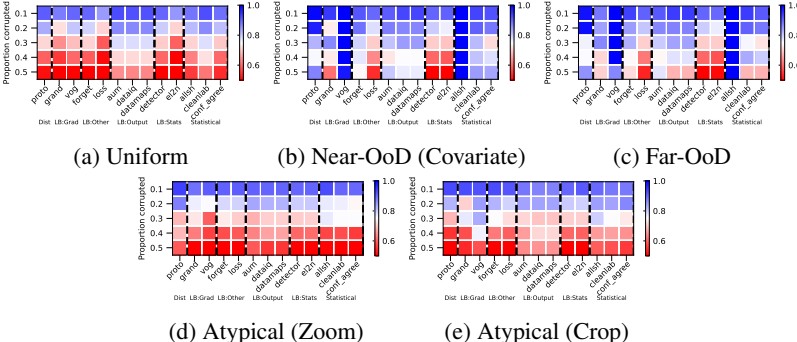

Figure 96: AUPRC on medical X-Ray for different HCMs for different hardness types aggregated across setups. We vary the proportion perturbed, i.e. the proportion of hard examples. Blue is better, red worse. We see variability of HCM capabilities across hardness types and proportions.

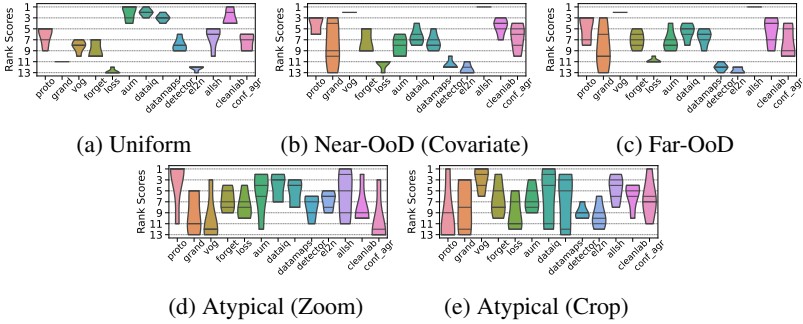

Figure 97: Performance rankings of HCMs vary depending on the hardness type.

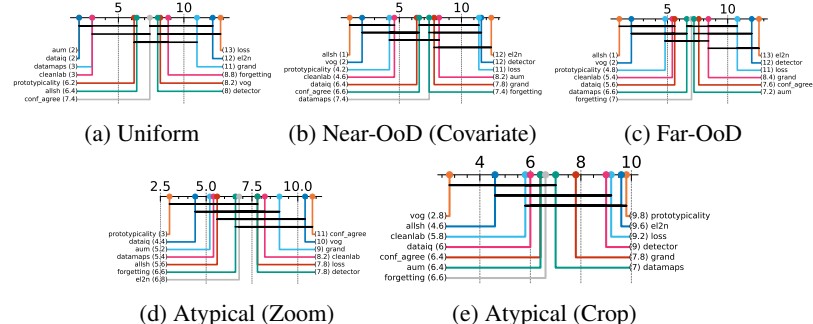

Figure 98: Critical difference diagrams highlight that similar categories/classes of HCMs *do not* have a statistically significant difference in their performance, indicated by the horizontal black lines linking HCMs which are not statistically different. The numbers in brackets denote mean rank.

We highlight the following main takeaways which are maintained from the main paper experiments, showing the findings indeed are similar across datasets:

- **Hardness types vary in difficulty.** Similarly to the main text, we find that different types of hardness are easier or harder to characterize. For instance, uniform mislabeling or Far-OoD are much easier than data-specific hardness like Atypical. Hence, we still see the performance differences.

- **Learning dynamics-based methods with respect to output confidence are effective general-purpose HCMs.** In selecting a general-purpose HCM, we find similar to the main text that HCMs that characterize samples using learning dynamics on the confidence — uncertainty-based methods, which use probabilities (DataMaps, Data-IQ) or logits (AUM), are the best performing in terms of AUPRC across the board.

- **HCMs typically used for computational efficiency are surprisingly uncompetitive.** Similarly to the main text, we find that HCMs that leverage gradient changes (e.g. GraNd), typically used for selection to improve computational efficiency, fare well at low $p$. However, at higher $p$, they become notably less competitive compared to simpler and computationally cheaper methods.

- **HCMs should only be used when hardness proportions are low.** In general, different HCMs have significantly reduced performance at higher proportions of hardness. This is expected as we get closer to 0.5 since it's harder to identify a clear difference between samples.

- **Individual HCMs within a broad "class" of methods are NOT statistically different.** We find from the critical difference diagrams that methods falling into the same class of characterization are not statistically significant from one another (based on the black connected lines), despite the minor performance differences between them. Hence, practitioners should select an HCM within the broad HCM class most suitable for the application.

- **Selecting an HCM based on the hardness is useful.** We find that confidence is a good general-purpose tool if one doesn't know the hardness. However, if one knows the hardness, one can better select the HCM.

Of course, there are subtle differences given the difference in datasets. As an example, the performance rankings exhibit higher variability. That said, this is a function of running fewer seeds compared to the main paper. The most important compared to raw ranking is the CD Diagram which provides an analysis of the statistical significance of the performance differences which the raw scores don't provide. In that case, despite the rankings being different, we see very many of the same HCMs from the main paper are similarly connected by black lines, indicating the performance differences are not statistically different.

## D.8    INSIGHTS: EFFECT OF SEVERITY OF PERTURBATION ON HARDNESS SCORES

**Overview.** We ask what is the effect of the severity of the perturbation on the hardness score. Specifically, are samples with greater perturbations considered "harder" than others by HCMs based on their scores.

We conducted an experiment for zoom shift, crop shift, and Near-OOD (covariate), where we have two severities of perturbation namely small and large. Specifically, for zoom (2x vs. 10x), crop shift (5 pixels vs. 20 pixels), and Near-OOD (std deviation of 0.5 vs. 2).

We report the results in Fig 99. The results show the percentage changes in hardness scores (in the direction of hardness, e.g. increase or decrease) for different HCMs for the different hardness types considered. In all cases, we see an increase in hardness score for increased severity, with some HCM scores showcasing greater sensitivity and changes than others. In particular, we see the gradient and loss-based approaches have the greatest changes as compared to the methods computing scores based on probabilities.

These results shed interesting insights into the sensitivities of different HCMS.

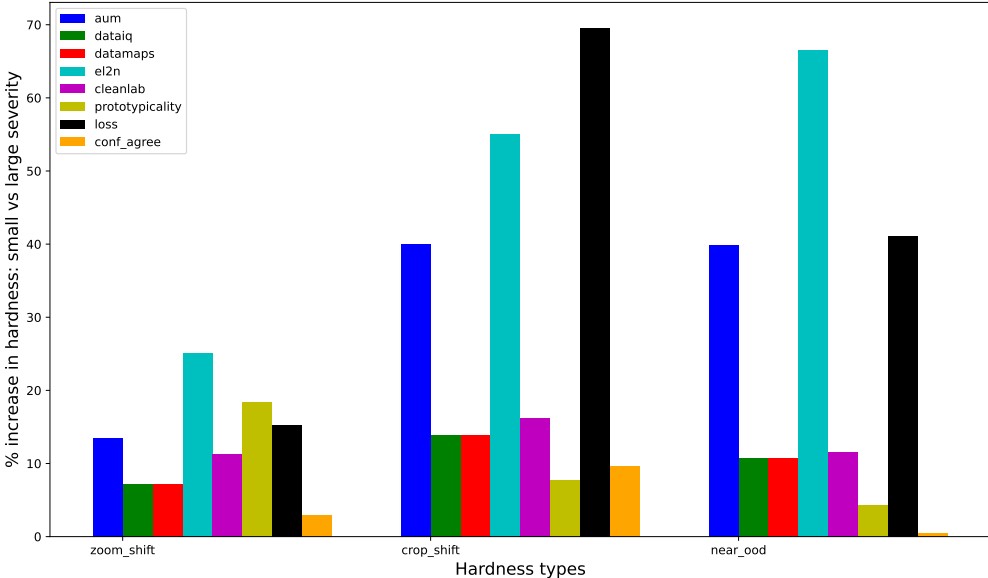

Figure 99: Effect on hardness scores with respect to perturbation severity from small to large.

