# OpenReview forum: "Dissecting Sample Hardness: A Fine-Grained Analysis of Hardness Characterization Methods for Data-Centric AI"
_ICLR.cc/2024/Conference — ICLR 2024 poster_

### Official Review · Reviewer_u8mR · 2023-10-21

**Soundness:** 3 good
**Presentation:** 3 good
**Contribution:** 4 excellent
**Rating:** 8
**Confidence:** 2

**Summary:**

The author claims that existing works on Hardness Characterization Methods (HCMs) fail to address comprehensive and quantitative evaluation of HCMs. Also, the author points out the absence of an integrated and practical software tool that can simulate and evaluate various types of HCMs. In this context, the author proposes a fine-grained taxonomy for categorizing multiple hardness types and introduces the Hardness Characterization Analysis Toolkit (H-CAT), which supports over 14K setups with 13 different HCMs across 8 hardness types. The author reports the performance of each HCM in various settings using H-CAT and provides invaluable takeaways and practical tips that were not seen in previous works.

**Strengths:**

1) This paper diligently considered all the recently proposed HCM algorithms within the past 3-4 years and developed H-CAT, reporting experimental results using it.
2) Through graphical illustration (Figure 1) and table analysis (Table 1), the need for a systemic evaluation framework for HCMs was effectively underscored.
3) Figures 3, 4, 5, and 6 analyzed the behavior of popular HCMs under various hardness settings from different perspectives. Additionally, the experiments conducted using the toolkit developed by the authors hold greater significance.
4) The author provided detailed instructions for use of H-CAT and additional experimental results through a voluminous appendix.

**Weaknesses:**

Hardness characterization is an important concept in various fields such as computer vision works like object detection and segmentation, as well as natural language processing and reinforcement learning. However, the proposed HCM evaluation framework in this paper can only handle image classification.

**Questions:**

1) In the Introduction, it is stated that "These 'hard' samples or data points can significantly hamper the performance of ML models." However, from an active learning perspective, can't we consider hard samples to be more informative and better samples with higher annotation efficiency?
2) On page 3, in the section "Data characterization," it is mentioned that difficult or easy samples are classified into separate easy or hard subsets. However, instead of categorizing sample difficulty into two binary sets, wouldn't it be better to dynamically control the learning process by assigning weights or using other methods based on continuous difficulty values? Additionally, excluding OOD/outlier and atypical samples from model training could potentially enhance the model's generalization capability. So, couldn't excluding them altogether from training be a suboptimal choice?
3) The author mentioned that existing works have a disadvantage of conducting indirect evaluation in downstream tasks. In that case, how does the proposed framework perform direct evaluation?
4) In Figure 6, Spearman rank correlation scores for HCMs are reported. As far as I know, Spearman rank correlation calculates the correlation between two variables. How was the correlation computed from multiple runs in this case?

---

> ### Author Response · Authors · 2023-11-16
> **Response to Reviewer u8mR [Part 1/2]**
>
> Dear ``Reviewer u8mR``
>
> Thank you for your thoughtful comments and suggestions! We give answers to each of the following in turn, along with corresponding updates to the revised manuscript. We have grouped answers for ease of readability as follows:
>
> (A) Hardness characterization in other settings __[Part 2/2]__
>
> (B) Comparison to hard samples in active learning __[Part 2/2]__
>
> (C) Usage of hardness scores __[Part 2/2]__
>
> (D) Direct evaluation clarification __[Part 2/2]__
>
> (E) Spearman rank correlation clarification __[Part 2/2]__

---

> ### Author Response · Authors · 2023-11-16
> **Response to Reviewer u8mR [Part 2/2]**
>
> ### (A) Hardness characterization in other settings
>
> We clarify that we primarily studied HCMs in the context of image classification, as most HCMs (10/13) have been developed in this context. While our current evaluation predominantly employs image datasets, we also incorporate tabular datasets ("Covertype" and "Diabetes130US") to demonstrate the generalizability of H-CAT across different data modalities. This inclusion of diverse datasets is a step towards extending the applicability of HCMs beyond image classification. We hope that the extensibility of our framework will encourage future work to cover even more domains as suggested by the reviewer.
>
> ---
>
> ### (B) Comparison to hard samples in active learning
>
> Thank you for pointing out the important distinction between the concepts of "hard" samples in active learning and in data-centric AI/hardness characterization. We wish to clarify the different contexts in which samples are termed as “hard” between the two areas of the literature.
>
> - **Active Learning**: "hard" samples refer to those instances where the model struggles to make accurate predictions, yet the samples are _correct_. These samples are valuable because they are informative. However, we note these samples are NOT inherently flawed or erroneous.
> - **Data-Centric AI and Hardness Characterization**: In contrast to active learning, in data-centric AI and hardness characterization, "hard" samples typically denote samples that have inherent issues or are erroneous, such as being mislabeled. Hence, the focus is on identifying and addressing these problematic samples to improve data quality. Consequently, we wish to detect and rectify the actual issues with the samples themselves, rather than to use them for informative purposes.
>
> Given this distinction, our work aligns with the second context, focusing on identifying and characterizing samples that are inherently problematic or erroneous. We thank the reviewer for bringing up this point.
>
> ---
>
> ### (C) Usage of hardness scores
>
>
> We agree with the reviewer that an important direction is how the types of samples could be used to dynamically control the learning process — for instance, a curriculum learning approach as suggested by the Reviewer in contrast to, for instance, pruning/sculpting them. These methods represent an interesting direction of how to leverage the hardness scores to enhance the performance of a downstream model.
>
> However, it is important to emphasize that these approaches, while important, are orthogonal to the primary focus of our work. Our research in H-CAT is centered on understanding, evaluating, and benchmarking the capability of HCMs to detect different types of hardness in samples. The core objective is to provide a comprehensive and systematic framework to evaluate the effectiveness of various HCMs in identifying hard samples across a range of hardness types.
>
> While we acknowledge the utility of the approaches suggested by the Reviewer in the context of model training, our research does not delve into the application of detected hard samples in training strategies. Instead, it focuses on the foundational aspect of accurately identifying these hard samples, which is a prerequisite for any subsequent application or strategy, including curriculum learning or sample pruning.
>
> We thank the reviewer for this suggestion and have included it in **Sec 6**, wherein future work can build on H-CAT and explore this dimension of what is the best way to use the hardness scores for a downstream model.
>
> **UPDATE:** Sec.6 now outlines work to understand the usage of hardness scores to guide future model training.
>
> ---
>
> ### (D) Direct evaluation clarification
>
> We clarify that we directly evaluate the capability of HCMs to detect specific hardness types. This approach allows us to understand which types of hardness each HCM can effectively identify, rather than merely assessing model improvement on a curated dataset, which might obscure the actual capabilities of different HCMs on different hardness types. offering valuable insights for the development and application of these methods. We hope that directly assessing which hardness types different HCMs are able to detect and showing their limitations will inspire researchers to develop new methodologies to address these limitations.
>
> ---
>
> ### (E) Spearman rank correlation
>
> We clarify that when we compute the Spearman correlation of scores across runs (different seeds), this is done between the hardness scores obtained for all combinations. This mirrors the approach adopted by Maini et al., 2022 and Seedat et al., 2022.

---

> > ### Comment · Reviewer_u8mR · 2023-11-22
> > **Post-rebuttal**
> >
> > Dear authors,
> >
> > Thank you for your sincere response.
> >
> > My concerns are all resolved and I acknowledge the contribution of this work.
> >
> > Best wishes,

---

> > > ### Author Response · Authors · 2023-11-23
> > > **Thank you for the response!**
> > >
> > > Dear Reviewer u8mR
> > >
> > > Thank you! And thanks again for your time and positive feedback!
> > >
> > > Regards
> > >
> > > Paper 7682 Authors

---

### Official Review · Reviewer_kMBG · 2023-10-30

**Soundness:** 3 good
**Presentation:** 4 excellent
**Contribution:** 2 fair
**Rating:** 6
**Confidence:** 4

**Summary:**

The paper proposes a new taxonomy for quantifying hardness of samples and then creates a framework on top of it (H-CAT) which uses existing hardness evaluation techniques to predict hardness categorized into different hardness types from their taxonomy. The paper then goes on to conduct hardness evaluation experiments on 2 image and 2 tabular datasets and highlights several key takeaways and recommendations for practitioners using their toolkit.

**Strengths:**

1. The taxonomy proposed by the paper is simple, clearly explicated, and practically useful. Relating it in detail to previous methods and frameworks and what's missing in existing HCMs based on this taxonomy helps readers create a good mental model of the space.
2. The comprehensive benchmarking of more than 13 HCMs across multiple datasets, on robust evaluation criteria is useful for the community.
3. Some takeaways like the efficacy of learning dynamics-based HCMs, lack of efficacy of computational-efficiency motivated methods, importance of significance testing and the variation in stability of the HCMs across seeds, backbones and parameterizations, all provide relevant, empirical information to practitioners using these methods.

**Weaknesses:**

1. Sticking to just MNIST and CIFAR10 limits the usefulness of the takeways. The dataset sizes and complexity we currently operate with are very different from MNIST/CIFAR10 and I'm afraid these learnings might not generalize to larger, more complex datasets.
2. The Near OOD, far OOD, and the atypical perturbations used in the experiments are oversimplified in my opinion. In the real world, defining near/far OOD and atypical have a complicated notion of support of the sample distribution, but the use of Gaussian noise, texture changes for near OOD and exchanging MNIST with CIFAR (which are drastically different) for far OOD, and using simple invariant functions like shift and zoom for atypical does not do justice to these concepts.
3. [Minor] The bulk of paper is focussed on creating a taxonomy for hardness evaluation and then building a software toolkit to evaluate existing methods for its quantification. Some initial discussions during the taxonomy creation and some of the non-obvious takeaways are instructive for the researchers in this field, but most of this work is geared towards practitioners. This isn't necessarily a negative thing, but limits the research novelty/impact of the work.

**Questions:**

1. As the severity of perturbations increases, do the methods used in HCMs also show an increase in the hardness score? Does this vary for learning-integrated methods vs others?
2. What is the effect of using augmentations during training for these models? I'm guessing the hardness predictions will change for some methods but not others.

---

> ### Author Response · Authors · 2023-11-16
> **Response to Reviewer kMBG [Part 1/3]**
>
> Dear ``Reviewer kMBG``
>
> Thank you for your thoughtful comments and suggestions! We give answers to each of the following in turn, along with corresponding updates to the revised manuscript. We have grouped answers for ease of readability as follows:
>
> (A) Beyond MNIST & CIFAR10 __[Part 2/3]__
>
> (B) Simplicity of perturbations __[Part 2/3]__
>
> (C) Nature of takeaways __[Part 2/3]__
>
> (D) Impact of perturbation severity __[Part 3/3]__
>
> (E) Effect of augmentations __[Part 3/3]__

---

> ### Author Response · Authors · 2023-11-16
> **Response to Reviewer kMBG [Part 2/3]**
>
> ### (A) Beyond MNIST & CIFAR10
>
> We wish to clarify our choice of MNIST and CIFAR-10 was strategic as these datasets are well-established in the HCM literature and, as discussed in **Sec 5**, these datasets are “clean” without significant mislabeling. This is essential to create controlled experimental conditions for benchmarking where hardness is introduced. Furthermore, to address concerns about generalizability, we also included evaluations on more complex tabular datasets in Appendix D, such as “Covertype” and “Diabetes130US” from OpenML. This approach ensures our findings are relevant across different data modalities and sizes.
>
> That said, we understand the Reviewer's potential concern and have conducted an additional experiment using a medical X-ray dataset from the NIH [R1], which features high-quality images curated by multiple medical experts. This additional experiment reinforces the applicability of our findings to more complex and real-world datasets. The results shown in the updated Appendix D.7 align with our initial findings, indicating that our observations regarding various hardness types are not limited to MNIST and CIFAR-10 datasets but are also relevant to more specialized datasets such as medical images. This further validates the generalizability and practical utility of our research across diverse data domains.
>
> **UPDATE:** Added a new Appendix D.7 with additional image experiment on a medical X-ray dataset
>
> ---
>
> ### (B) Simplicity of perturbations
>
> We acknowledge the Reviewer's concerns about the perceived simplicity of the OOD and atypical perturbations. We highlight two aspects: (1) our methodology mathematically satisfies the definitions of these perturbations, and (2) **Figure 7 (now Figure 3 in the revised paper)** illustrates that the perturbations we used are representative of hardness types that could realistically occur in actual datasets. Moreover, while our work is indeed a first step, it represents a significant advancement from the existing state of HCM research which doesn’t consider these issues (as shown in **Table 1**). We hope that our framework serves as a foundation for more complex and nuanced studies in the future.
>
> Finally, a key finding is that certain **HCMs even fail on these simple perturbations**, which uncovers a fundamental limitation of the HCM field that needs to be addressed.
>
> ---
>
> ### (C) Nature of takeaways [minor]
>
> The paper indeed has a practical aspect of the benchmarking framework. We wish to clarify that this enables the HCM field to rigorously evaluate HCMs for the first time, which is essential for the field to progress. We believe this will help the HCM field to move from the ad hoc and qualitative state (as shown in Table 1) toward a more comprehensive and rigorous evaluation across multiple hardness types.
>
> Specifically, our work responds to the recent calls for more rigorous benchmarking and understanding of existing ML approaches, as highlighted by Guyon (2022), Lipton & Steinhardt (2019), and Snoek et al. (2018). We then believe this benchmark offers critical insights into the HCMs which are distilled into (1) benchmarking takeaways: to guide researchers about future development of HCMs by highlighting their strengths and weaknesses and (2) practical takeaways: to guide selection and usage of HCMs.
>
> Overall, this work provides insights into what types of hardness different HCMs are able to detect and shows the limitations of HCMs in detecting certain hardness types, which we hope will inspire researchers to develop new methodologies to address these limitations.

---

> ### Author Response · Authors · 2023-11-16
> **Response to Reviewer kMBG [Part 3/3]**
>
> ### (D) Impact of perturbation severity
>
>
> We thank the reviewer for this interesting suggestion! We have conducted an experiment for zoom shift, crop shift, and Near-OOD (covariate), where we have two severities of perturbation namely small and large. For zoom (2x vs 10x), crop shift (5 pixels vs 20 pixels), and Near-OOD (std deviation of 0.5 vs 2).
>
> We report the results in Fig 99, in Appendix D.8. The results show the percentage changes in hardness scores (in the direction of hardness, e.g. increase or decrease) for different HCMs for the different hardness types considered. n all cases, we see an increase in hardness score$^1$ for increased severity, with some HCM scores showcasing greater sensitivity and changes than others. In particular, we see the gradient and loss-based approaches have the greatest changes as compared to the methods computing scores based on probabilities. We thank the reviewer for suggesting this interesting experiment which has helped to shed further insights into the sensitivities of different HCMs.
>
> For ease of access the result can also be found at this Imgur link: **https://imgur.com/tTv6lO7**
>
> **UPDATE**: Included a new Appendix D.8 with the experiment.
>
> ----
>
> ### (F) Effect of augmentations
>
> Regarding the role of augmentations, we wish to clarify the placement of HCMs in the ML pipeline. HCMs are typically used to audit and curate the training set before the downstream model training begins. On the other hand, data augmentations are commonly employed during the model training phase to enhance robustness and generalizability. Given this difference, the HCMs we benchmark from the literature _do not_ use augmentations at all. Thus, since our study is about assessing existing HCMs, not proposing a new HCM, we believe it is outside the scope of our current study. However, we believe it is an interesting idea for future HCM algorithms to explore — potentially as a way to address the limitations uncovered by our benchmark.
>
> ----
>
> ### References
>
> [R1] Majkowska, Anna, et al. "Chest radiograph interpretation with deep learning models: assessment with radiologist-adjudicated reference standards and population-adjusted evaluation." Radiology 294.2 (2020): 421-431.
>
> $^1$ For some HCMs increased hardness is an increased score, whereas for others is a decrease (see Appendix A). We report the direction of hardness.

---

> ### Author Response · Authors · 2023-11-21
> **Dear Reviewer kMBG**
>
> Dear Reviewer kMBG
>
> Thank you again for your time and expertise during the review process! Your suggestions have really helped us improve the paper.
>
> We just wanted to check in if there's anything else we could do before the discussion period ends.
>
> Regards,
>
> Paper 7682 Authors

---

### Official Review · Reviewer_zsTV · 2023-11-02

**Soundness:** 4 excellent
**Presentation:** 2 fair
**Contribution:** 2 fair
**Rating:** 8
**Confidence:** 4

**Summary:**

This paper proposes a new framework called H-CAT for evaluating HCM methods that measure how difficult it is for a classification model to learn from certain examples. The paper provides three possible reasons for an example to be 'hard':
- The example is mislabeled
- It is OOD or outlier
- The example is valid but is atypical

Now the methods that are used to estimate the any of hardness types above are called HCMs:
- Learning dynamics-based
- Gradient-based
- Distance-based

The paper presents a codebase which allows to evaluate the hardness types as well as the HCMs. The main takeaways are:
- Many HCMs disagree with each other.
- Some types of hardness are characterize to measure than others.
- Learning dynamics-based methods are generally more effective.
- The best HCM to use depends on the specific problem.

**Strengths:**

- The paper introduces a novel codebase for the systematic analysis of HCMs. The categorization done in section 2.1 is nice to read. It is great to define hardness more formally and identify different underlying reasons for the difficulty a classification model in learning from certain examples.

- The paper have sufficient experiments. The authors evaluated 13 HCMs, providing a robust validation of their findings.

- The takeaways are generally valuable. The authors offer practical advice on how to choose HCMs based on the type of data hardness, and they highlight the advantages of learning dynamics-based methods.

- Overall, the paper is well-written. However, reorganizing the content and moving some parts to the appendix and placing back some key figures and explanations to the main text could improve its readability.

**Weaknesses:**

- The paper could be structured better. Introducing and clearly defining "hardness types" and "HCMs" earlier would help readers understand the paper better. Some important explanations and figures that support the main concepts are in the appendix, which makes it difficult for readers to access them quickly and easily

- The paper would be more technically rigorous if it included detailed equations and unified notation for the various HCMs in the appendix. Providing a more detailed explanation of methods, AUM for example, in the main text would also make the paper more instructive.

- While the authors acknowledge certain limitations, two are particularly significant in my opinion:

1. Before evaluating HCMs, you need to know what type of hardness is present in the data. This may not always be possible in real-world scenarios.
2. The paper does not address the fact that hardness can change during the training process. A hard example at epoch 1 might not be hard at epoch 10.

**Questions:**

I would like to know authors opinion regarding the limitations mentioned above, particularly 1) not knowing the type of hardness and 2) evolving hardness.

Two recommendations:
- I would suggest to reorganize the content to ensure a logical flow of information, placing Figure 7, for example, and some details to the main body can help grasping key concepts.

- Add detailed equations in a unified notation for the HCMs explored in section A.2.*. This would make your paper a go-to paper whenever someone wants to learn about hardness literature.

--------------------
Post rebuttal note:
Thank you very much for your response and applying the changes I suggested. I respectfully disagree with Reviewer 5UHg- the fact that the community's focus is now on LLMs doesn't make this type of works worthless. I believe that this paper has significant potential to advance our understanding of hardness in machine learning and hence I vote towards acceptance.

---

> ### Author Response · Authors · 2023-11-16
> **Response to Reviewer zsTV [Part 1/2]**
>
> Dear ``Reviewer zsTV``
>
> Thank you for your thoughtful comments and suggestions! We give answers to each of the following in turn, along with corresponding updates to the revised manuscript. We have grouped answers for ease of readability as follows:
>
> (A) Paper updates: Appendix content to main paper __[Part 2/2]__
>
> (B) Unified notation for HCMs __[Part 2/2]__
>
> (C) Clarification — knowing hardness a priori __[Part 2/2]__
>
> (D) Clarification — Hardness changing over training __[Part 2/2]__

---

> ### Author Response · Authors · 2023-11-16
> **Response to Reviewer zsTV [Part 2/2]**
>
> ### (A) Paper updates: Appendix content to main paper
>
> We thank the reviewer for the suggested paper updates of moving contents from the appendix to the main paper which we believe have improved the paper.
>
> We have now moved Figure 7 (now Figure 3) into the main paper to readers' intuition of the differences between OOD and Atypical, as well as show that our perturbations are indeed realistic.
> Introduce hardness types in the introduction further with intuitive examples
> Building on our introduction of HCMs in the intro, we move the grouping of the broad classes of HCMs (e.g. learning-based) from the Appendix into our formulation in Section 2.
>
> **UPDATE**: moved Figure 7 (now Figure 3) into the main paper
>
> ---
>
> ### (B) Unified notation for HCMs
>
>
> We thank the reviewer for the suggestion of adding the equations and unified notation for the HCMs in the Appendix, to make it a go-to-paper on hardness literature. We describe this below and have **updated Appendix A** with the following changes.
>
> - Added the unified input-output formulation for any HCM. We have also formulated HCMs are defined by a scoring function $S$.
> - We mathematically define each scoring function under this unified formulation requiring a scoring function.
> - We have added a new Table 2 in the Appendix, formalizing the meaning of the scores per HCM.
>
> ---
>
> ### (C) Clarification — knowing hardness a priori
>
> We wish to clarify the potential misunderstanding about the hardness type being known and make the distinction between the benchmark framework and HCMs. Our benchmark framework tests specific hardness types. This is vital for establishing a controlled benchmarking environment. Without such a ground truth, it would be challenging, if not impossible, to accurately assess and compare the capabilities of different HCMs. However, we emphasize that the HCM itself does not have knowledge of the hardness type when being evaluated. We apologize if this was unclear.
>
> ---
>
> ### (D) Clarification — Hardness changing over training
>
> The inherent hardness of a sample remains consistent regardless of the training epoch. For instance, a sample that is hard due to mislabeling retains this characteristic throughout the training process. A mislabeled sample is indeed mislabeled at both epoch 1 and epoch 10. This constancy is crucial to our understanding and characterization of sample hardness. Our work on H-CAT then focuses on assessing the capability of the HCM scores to correctly identify these inherent hardnesses in the sample itself.

---

### Official Review · Reviewer_5UHg · 2023-11-02

**Soundness:** 2 fair
**Presentation:** 2 fair
**Contribution:** 1 poor
**Rating:** 1
**Confidence:** 5

**Summary:**

This paper did the following:
- A taxonomy of hardness -- mislabel, OOD, atypical
- Provide a library of modules for dataloading, hardness measurements across different types of hardness, unified interface and evaluator
- Propose that hardness is ill-defined, indeed
- Propose that hardness is usually being measured by downstream, which should not be
- Give a list of takeaways at the end
- Experimentd ran on cifar and mnist

**Strengths:**

The paper dedicated a significant portion to describing why hardness is not well-defined anywhere. It also provides a taxonomy of hardness that are split into mislabeling, OOD and atypical. It also provides a unified interface for evaluation.

**Weaknesses:**

The authors stated that hardness should not depend on downstream improvement. I find that to be very controversial. One model's hardness may be another model's easy. Maybe I have not understood this completely and would like clarification. Moreover, in the end you run classification on cifar and mnist to obtain a list of takeaways iiuc -- did the authors not just use downstream tasks (cifar and mnist classifications) to inform us of their takeaways? If I understood incorrectly, please provide clarification. Also the categorization of hardness into mislabel, ood and atypical is quite unsatisfying. Why mislabel? Also in this era of self-supervision, LLM autoregressive, etc., I find this paper a little not keeping up with time. What is hardness in this new day and age?

Experiments on cifar and mnist only --- authors claimed that they did not select imagenet because imagenet has less than 5% mislabel. Can you not artificially mislabel yourself then?

A large part of the paper reads like an engineering document. I appreciate that, but I think scientific insights are more important while the engineering information can be delegated to appendix.

**Questions:**

See weaknesses.

---

> ### Author Response · Authors · 2023-11-16
> **Response to Reviewer 5UHg [Part 1/3]**
>
> Dear ``Reviewer 5UHg``
>
> Thank you for your review. We give answers to each of the following in turn, along with corresponding updates to the revised manuscript. We have grouped answers for ease of readability as follows:
>
> (A) Relevance of the paper to current ML trends __[Part 2/3]__
>
> (B) Clarifying the role of downstream improvement __[Part 2/3]__
>
> (C) Clarifying takeaways are not based on downstream improvement __[Part 2/3]__
>
> (D) Categorization of hardness types __[Part 3/3]__
>
> (E) Experiments on CIFAR and MNIST __[Part 3/3]__
>
> (F) Inclusion of technical details in the main manuscript __[Part 3/3]__

---

> ### Author Response · Authors · 2023-11-16
> **Response to Reviewer 5UHg [Part 2/3]**
>
> ### (A) Relevance of the paper to current ML trends
>
> We respectfully disagree that a paper on hardness characterization is *“not keeping up with time”*. To answer the Reviewer’s question *“What is hardness in this new day and age?”*, as outlined in Section 1, data is a vital component of all machine learning methods and hence, methods for understanding data is critical. In our paper, we address data hardness, which is increasingly used to audit datasets. We examine hardness in both the input space $X$ and label space $Y$. Consequently, our work is relevant to many areas of ML, including multiple data modalities, learning paradigms, and model architectures. For instance, consider the following recent and high-profile LLM paper: Textbooks Are All You Need [R1], which shows the importance of identifying high-quality samples and filtering the dataset. This is the core premise of HCMs. In fact, hardness is even more relevant in current times as HCMs can help guide data filtering as datasets grow ever larger.
>
> Additionally, the rise of data-centric AI (Liang et al., 2022; Seedat et al., 2022b) as a key ML research field highlights the importance of HCMs as a mechanism to enable data quality. The relevance and growing interest in this topic is shown by tutorials on data-centric AI at the following major conferences: KDD, IJCAI, and NeurIPS in 2023. Our work directly contributes to this increasingly important field, addressing fundamental challenges in understanding and improving data quality.
>
> Moreover, the HCMs we benchmark are from 2019-2023 underscoring their relevance to current times (see Table 1). Finally, we believe H-CAT also addresses recent calls for more rigorous benchmarking (Guyon, 2022) and for better understanding of existing ML methods (Lipton & Steinhardt, 2019; Snoek et al., 2018).
>
> ---
> ### (B) Clarifying the role of downstream improvement
>
> We clarify that we did not state **“that hardness should not depend on downstream improvement”**, rather we assert HCMs should be evaluated beyond __only__ their ability to curate a dataset to improve downstream performance. While downstream performance is undoubtedly significant, only looking at downstream performance “overlooks the fundamental quantitative [hardness] identification task” (see abstract and Sec 2.2). Hence, the key contribution of our work is to quantitatively evaluate the capabilities of different HCMs to identify different types of hardness in datasets. This represents the first work to assess HCMs on this fundamental task in a comprehensive manner. This is important as in order to advance the field of hardness characterization we need to understand the fundamental mechanisms behind HCMs and their strengths and weaknesses in handling different types of hardness.
>
> ---
> ### (C) Clarifying takeaways are not based on downstream improvement
>
> We clarify that we do __not__ use downstream task performance to inform the takeaways. Rather we evaluate the fundamental hardness detection capability of HCMs on different hardness types, providing an objective and quantitative assessment of whether HCMs correctly detect/identify the hard samples they claim.

---

> ### Author Response · Authors · 2023-11-16
> **Response to Reviewer 5UHg [Part 3/3]**
>
> ### (D) Categorization of hardness types
>
> We clarify our categorization of hardness into mislabeling, OoD/outlier, and atypical is grounded in the existing body of ML literature which has highlighted these as prevalent and impactful types of data hardness in real-world data (see **Sec 2.1**). Mislabeling, in particular, is a pervasive issue in ML datasets (see Northcutt et al., 2021b) and poses a significant challenge to model performance in the real world. Please see the references used in our paper to motivate the categorization of hardness types in our taxonomy.
>
>  - Mislabeling: (Paul et al., 2021; Swayamdipta et al., 2020; Pleiss et al., 2020; Toneva et al., 2019; Maini et al., 2022; Jiang et al., 2021; Mindermann et al., 2022; Baldock et al., 2021, Northcutt et al., 2021a; Sukhbaatar et al., 2015, Jia et al., 2022; Han et al., 2020; Hendrycks et al., 2018; Song et al., 2022)
> - Near OoD (Anirudh & Thiagarajan, 2023; Mu & Gilmer, 2019; Hendrycks & Dietterich, 2018; Graham et al., 2022; Yang et al., 2023; Tian et al., 2021)
> - Far OoD (Mukhoti et al., 2022; Winkens et al., 2020; Graham et al., 2022; Yang et al., 2023)
> - Atypical: (Yuksekgonul et al., 2023, Feldman, 2020; Hooker et al., 2019; Hooker, 2021; Agarwal et al., 2022)
>
> We believe that our formal hardness taxonomy provides a structured framework for the ML community to engage with this critical but overlooked issue of the dimensions of hardness. Finally, while of course, other dimensions could be considered — our taxonomy is much broader than previously considered by HCMs and we hope the community will build upon it.
>
> ---
>
> ### (E) Experiments on CIFAR and MNIST
>
> We first clarify the misunderstanding, we did not say **“ImageNet has less than 5% mislabel”** but rather the opposite. On Page 7, we state **“other common image datasets like ImageNet…contain significant mislabeling (over 5%), hence we cannot perform controlled experiments”** (see Sec 5 and Appendix B). The issue is not that we cannot artificially mislabel it, but that conducting controlled experiments is challenging due to substantial ground truth mislabeling. Consequently, this motivated our usage of MNIST and CIFAR-10; as mentioned in Sec 5, these datasets are clean without significant mislabeling (<0.5%). This allows us to create controlled experimental conditions for benchmarking where hardness, such as mislabeling, is introduced.
>
> Additionally, we have conducted an additional assessment on a large-scale medical X-ray dataset from the NIH [R2], which has been audited and curated by multiple radiologists to ensure a high-quality and clean dataset for inclusion in our benchmark. We see that the major findings and takeaways are retained as shown in the updated Appendix D.7  for the X-ray dataset.
>
> **UPDATE:** Appendix D.7 with additional image experiment on a medical X-ray dataset
>
> ---
>
> ### (F) Inclusion of technical details in the main manuscript
>
> Without the engineering aspects and technical details, such as the development of H-CAT, one cannot perform the rigorous evaluation of HCMs. We believe our takeaways and findings have led to both insights and understanding into HCMs and their strengths and weaknesses, which have never been evaluated before our framework. These insights are not secondary to the engineering details; rather they are interdependent. The technical and engineering details does not diminish the scientific rigor or the insights which we obtain through our work. On the contrary, it enhances its relevance and applicability. Moreover, we believe our formulation of the hardness taxonomy under a common framework is a key scientific contribution of the paper.
>
> Additionally, the paper falls under the **ICLR call for papers of 'datasets and benchmarks'**. Hence, the benchmarking framework is a fundamental aspect of the paper. More specifically, the technical details of the benchmark are crucial to include in the paper to give the reader an understanding of how the insights in Section 5 are obtained.
>
> ---
> ### References
> [R1] Gunasekar, Suriya, et al. "Textbooks Are All You Need." arXiv preprint arXiv:2306.11644 (2023).
>
> [R2] Majkowska, Anna, et al. "Chest radiograph interpretation with deep learning models: assessment with radiologist-adjudicated reference standards and population-adjusted evaluation." Radiology 294.2 (2020): 421-431.

---

> ### Comment · Reviewer_5UHg · 2023-11-22
>
> This is what you wrote verbatim "Unfortunately, current HCMs have only been evaluated on specific types of hardness and often only qualitatively or with respect to downstream performance, overlooking the fundamental quantitative identification task."
>
> I understand your point. I was asking more of this: it seems all your key takeaways that you listed are based off cifar/mnist dataset and downstream task. Hence I find it hard to draw conclusion supporting "overlooking the fundamental quantitative identification task" because your key takeaways are based mnist and cifar ...
>
> While I do appreciate the whole framework, but I am asking is it really useful in the fundamental sense which your paper is kind of claiming iiuc?
>
> RE: "not keeping up with time" -- let me put it this way, from your paper, I don't know how to characterize hardness for LLM, LMM. What is OOD in LMM/LLM?
>
> On ImageNet: sorry, I misread. However, I am still concerned that the datasets used are very toy, very small. I would like to see some real world datasets.

---

> > ### Comment · Reviewer_5UHg · 2023-11-22
> >
> > Also, you did not answer my question (or at least I did not see it) on why categorize hardness into mislabel, ood and atypical. Do these three cover all the hardness in the world of data?
> >
> > You also only show for images. For claiming being a fundamental hardness framework, what about other modalities? I am not trying to split hair, but my point is that it comes down to these three categories on images, and you create samples of them on mnist and cifar and run experiments on them, and finally make those takeaways. That's what it is. All the modules are engineering models (and I am not disputing their usefulness), but everything comes down to what I described. What is so fundamental about it?

---

> > > ### Author Response · Authors · 2023-11-22
> > > **Response to Reviewer 5UHg comments [Part 1/2]**
> > >
> > > Dear Reviewer 5UHg,
> > >
> > > Thank you for your response. Please find answers to your questions below, many of which have been answered in our earlier response or in the paper itself, which we re-clarify below.
> > >
> > > _The response is across parts 1 and 2_
> > >
> > > ----
> > > ``Part 1/2``
> > >
> > > ----
> > > > "all your key takeaways that you listed are based off cifar/mnist dataset and downstream task."
> > >
> > >
> > > We clarify both misunderstandings:
> > >
> > > (1) Our takeaways are **not** just based on CIFAR/MNIST. As mentioned in **Section 5**, we also included tabular datasets from OpenML ( “Covertype” and “Diabetes130US” ). We also included a new real-world medical image dataset of chest X-rays (Majkowska et al., 2020) showing the translation of our takeaways — we referenced this new dataset in **point “(E): Experiments on CIFAR and MNIST” of our previous response**.
> > >
> > > (2) As mentioned in point **”(C) Clarifying takeaways are not based on downstream improvement”** of our previous response, “we do **not** use downstream task performance to inform the takeaways. Rather we evaluate the fundamental hardness detection capability of HCMs on different hardness types, providing an objective and quantitative assessment of whether HCMs correctly detect/identify the hard samples they claim.”
> > >
> > > ---
> > >
> > > > "While I do appreciate the whole framework, but I am asking is it really useful in the fundamental sense which your paper is kind of claiming iiuc?"
> > >
> > >
> > > We respectfully disagree and believe our paper makes useful contributions to the field of HCMs — along the following 4 dimensions.
> > >
> > > (1) Our work enables the HCM field to rigorously evaluate HCMs for the first time, which is essential for the field to progress. We believe this will help the HCM field to move from the ad hoc and qualitative state (as shown in Table 1) toward a more comprehensive and rigorous evaluation across multiple hardness types.
> > >
> > > (2) Our work responds to the recent calls for more rigorous benchmarking and understanding of existing ML approaches, as highlighted by Guyon (2022), Lipton & Steinhardt (2019), and Snoek et al. (2018).
> > >
> > > (3) Our benchmark offers critical insights into the HCMs which are distilled into (1) benchmarking takeaways: to guide researchers about the future development of HCMs by highlighting their strengths and weaknesses and (2) practical takeaways: to guide selection and usage of HCMs.
> > >
> > > (4) Overall, this work provides insights into what types of hardness different HCMs are able to detect and shows the limitations of HCMs in detecting certain hardness types, which we hope will inspire researchers to develop new methodologies to address these limitations.
> > >
> > > ----
> > >
> > > > "I don't know how to characterize hardness for LLM, LMM. "
> > >
> > >
> > > How to use HCMs for different applications is beyond the scope of the paper. We refer the Reviewer to the following resources where the Cleanlab HCM (which we have evaluated in our paper) has been used for two different LLM use cases. We hope the reviewer finds them helpful!
> > >
> > > (1) Curating a high-quality LLM fine-tuning dataset: https://cleanlab.ai/blog/fine-tune-LLM/
> > >
> > > (2) Reliable few-shot prompt selection: https://cleanlab.ai/blog/reliable-fewshot-prompts/
> > >
> > > We wish to reiterate that our paper is orthogonal to any one application or method, as we aim to benchmark different HCMs for the first time and to understand which hardness types they are capable of detecting. We primarily studied HCMs in the context of image classification, as almost all HCMs have been developed in this context. We also incorporate tabular datasets to demonstrate the generalizability of H-CAT across different data modalities.
> > >
> > > Overall, the primary goal of this work is to provide insights into what types of hardness different HCMs are able to detect and show the limitations of HCMs in detecting certain hardness types, which we hope will inspire researchers to develop new HCM methodologies to address these limitations.
> > >
> > > ----
> > >
> > > > "On ImageNet: sorry, I misread. However, I am still concerned that the datasets used are very toy, very small. I would like to see some real world datasets."
> > >
> > >
> > > We clarify that we do include real-world datasets. Please see **point (E) of our previous response** where we address this and include a real-world medical image dataset, which has been included in a new Appendix D.7.
> > > Please see the below snippet from our **previous response in (E)**:
> > >
> > >  _“Additionally, we have conducted an additional assessment on a large-scale medical X-ray dataset from the NIH [R2], which has been audited and curated by multiple radiologists to ensure a high-quality and clean dataset for inclusion in our benchmark. We see that the major findings and takeaways are retained as shown in the updated Appendix D.7 for the X-ray dataset.”_
> > >
> > >
> > >
> > > ``response continues``

---

> > > > ### Author Response · Authors · 2023-11-22
> > > > **Response to Reviewer 5UHg comments [Part 2/2]**
> > > >
> > > > ``response continued``
> > > >
> > > > ----
> > > > ``part 2/2``
> > > >
> > > > ---
> > > >
> > > > > you did not answer my question (or at least I did not see it) on why categorize hardness into mislabel, ood and atypical. Do these three cover all the hardness in the world of data?
> > > >
> > > > We would like to refer the Reviewer to point **(D) Categorization of hardness types** in our previous response where we **did answer** the question of hardness categorization. To expand further, in our formalism (Sec 2.1), we consider hardness manifesting in $X$, $Y$, or $X,Y$, and how these affect the joint probability distribution. Hence, the formalism itself has general applicability. While, of course, the implemented perturbations might not cover every possible scenario, it is still substantially broader than previously considered when evaluating HCMs and we hope the community will build upon it. **We include the snippet from our previous response below.**
> > > >
> > > > ``Snippet of previous response: (D) Categorization of hardness types``
> > > >
> > > > _We clarify our categorization of hardness into mislabeling, OoD/outlier, and atypical is grounded in the existing body of ML literature which has highlighted these as prevalent and impactful types of data hardness in real-world data (see Sec 2.1). Mislabeling, in particular, is a pervasive issue in ML datasets (see Northcutt et al., 2021b) and poses a significant challenge to model performance in the real world. Please see the references used in our paper to motivate the categorization of hardness types in our taxonomy._
> > > >
> > > > [_please see references in the previous response_]
> > > >
> > > > _We believe that our formal hardness taxonomy provides a structured framework for the ML community to engage with this critical but overlooked issue of the dimensions of hardness. Finally, while of course, other dimensions could be considered — our taxonomy is much broader than previously considered by HCMs and we hope the community will build upon it._
> > > >
> > > > ``snippet end``
> > > >
> > > > ----
> > > >
> > > > > You also only show for images. For claiming being a fundamental hardness framework, what about other modalities?
> > > >
> > > >
> > > > We clarify we do **not**  only show for images. As mentioned in **Section 5** of the paper we state: “Furthermore, to show generalizability across modalities, we also evaluate on **tabular datasets** “Covertype” and “Diabetes130US” benchmark datasets (Grinsztajn et al., 2022) from OpenML (Vanschoren et al., 2014)”.
> > > > That said it is important for us to focus on image datasets as this is the modality for which the almost all HCMs have been developed as we mention in both **Section 4.1** and **Section 5**.
> > > >
> > > > ----
> > > >
> > > > > I am not trying to split hair, but my point is that it comes down to these three categories on images, and you create samples of them on mnist and cifar and run experiments on them, and finally make those takeaways. That's what it is. All the modules are engineering models (and I am not disputing their usefulness), but everything comes down to what I described. What is so fundamental about it?
> > > >
> > > > What the reviewer describes about our paper is precisely a benchmarking paper, seeking to deepen our understanding of current HCMs — which has not been done previously. This required more than “just” engineering, for example, formalizing hardness in section 2.1 “Taxomony of hardness”. We detail these contributions below. Additionally, we wish to highlight that this year the ICLR call for papers includes a track specifically for **datasets and benchmarks** — which we noted as the primary submission area for this work. With this in mind, our work responds to the recent calls for more rigorous benchmarking and understanding of existing ML approaches, as highlighted by Guyon (2022), Lipton & Steinhardt (2019), and Snoek et al. (2018).
> > > >
> > > > In addressing these calls, our work provides four main contributions to the field: **(1) Hardness taxonomy:** we formalize a systematic taxonomy of sample-level hardness types; **(2) Benchmarking framework:** to evaluate the strengths of different HCMs across hardness types; **(3) Systematic and quantitative HCM evaluation:** prior to our study, no research had comprehensively evaluated HCMs across various hardness types nor directly evaluated HCMs' capabilities to correctly detect hard samples — we do for 13 different HCMs across 8 different hardness types; **(4) Insights:** our benchmark provides novel insights into the capabilities of different HCMs when dealing with different hardness types, exposing gaps and opportunities in current HCMs and offering practical tips for researchers and practitioners. We hope the H-CAT framework will both add rigor to the HCM field and address the critical need for a structured and empirical approach to understanding and improving HCM methodologies.

---

### Official Review · Reviewer_SFiV · 2023-11-03

**Soundness:** 2 fair
**Presentation:** 2 fair
**Contribution:** 3 good
**Rating:** 8
**Confidence:** 3

**Summary:**

The paper presents a benchmark for evaluating methods that make supervised learning models robust to hard data examples. The paper introduces a taxonomy of hard cases and defines each category in terms of where the corruption happens and how the data distributions differ. The authors developed a toolkit to automatically evaluate models and include 13 algorithms in their study. The paper summarizes the results and presents the main highlights, together with an extensive supplementary material with more details.

**Strengths:**

* The organization of the taxonomy is sound and formalizes observations from prior literature.
* The systematic experimental design is strong and the evaluations are exhaustive.
* The toolkit for evaluating and comparing performance across HCM seems well designed.

**Weaknesses:**

* The conclusions of the study seem limited. Despite the extensive experimental work, it is unclear how the hardness characterization field can make progress. While the paper claims to have practical tips, there is the underlying assumption that the hardness type is known and coming from only one type. This is not realistic in practice and undermines the value of the recommendations for practitioners.
* The paper insists that studying data hardness is critical for advancing data-centric AI, but it is unclear what data-centric AI is, and why HCMs are the key to advance it. A more fundamental question is, do we really need HCMs at all or we only need to create high quality datasets that eliminate corrupted examples?
* Following the need for HCMs, a baseline of training models without HCMs is missing. There are two scenarios that should be considered as baselines: 1) training a model without HCM with the same level of corruption introduced as the other models. 2) Training a model without HCM and with only the easy examples, i.e. removing the portion of corrupted samples. This would be informative to evaluate what cases in the taxonomy really need to be investigated further with HCMs and which ones can be addressed by simply cleaning the dataset.
* The taxonomy is manually defined based on the interpretation of previous literature, and the datasets used in the study are intentionally clean (MNIST and CIFAR) to simulate hard samples. Perhaps data hardness should be evaluated in a data-driven way, i.e., using a real world dataset and using data science, statistics, and machine learning to find out the nature of hard samples on the dataset. This reviewer finds the approach of defining the hardness of data artificial, and perhaps not reflecting the challenges of current problems and datasets.
* The paper points to the use of indirect measures, such as improvement in downstream performance, as an issue (Sec. 2.2). However, it seems like the metric that they use to evaluate methods and create the benchmark is downstream performance in the corresponding classification task. It was not clear what the proposed solution to this issue is in the benchmark.
* The writing style and paper formatting is too flashy and very distracting. Instead of having a natural flow, the paper has sentences in bold, boxes in colors, and other decorations that ask for too much attention, giving the impression that there is something more important to read than the argument the reader is focusing on. I found this style very hard to follow. In addition, the paper reads as if it was trying hard to sell a toolkit rather than presenting a study with solid conclusions.

**Questions:**

Unfortunately, I don't find this study very compelling as a full research article. As it is currently, the paper seems to be more appropriate for a Datasets & Benchmarks track at NeurIPS, for instance, rather than bringing in novel insights to advance hardness characterization methodology. Maybe I'm missing what the most important conclusion is, which is what I'd like the authors to clarify.

---

> ### Author Response · Authors · 2023-11-16
> **Response to Reviewer SFiV [Part 1/3]**
>
> Dear ``Reviewer SFiV``
>
> Thank you for your thoughtful comments and suggestions! We give answers to each of the following in turn, along with corresponding updates to the revised manuscript. We have grouped answers for ease of readability as follows:
>
> (A) Suitability for datasets & benchmarking __[Part 2/3]__
>
> (B) Conclusions of the paper & known hardness type __[Part 2/3]__
>
> (C) Clarification: What is data-centric AI & HCMs vs creating high quality datasets __[Part 3/3]__
>
> (D) Training models without HCMs __[Part 3/3]__
>
> (E) Evaluating data hardness __[Part 3/3]__
>
> (F) Indirect measures vs our use of direct measures __[Part 3/3]__
>
> (G) Writing updates __[Part 3/3]__

---

> > ### Comment · Reviewer_SFiV · 2023-11-21
> > **Thank you for the clarifications and revisions!**
> >
> > I want to thank the authors for addressing my questions and recommendations. The paper is indeed valuable for advancing HCMs methodology.
> >
> > In particular, thank you for bringing to my attention that "datasets and benchmarks" are part of the call for papers in ICLR. In addition, I appreciate that the authors have clarified that the performance metric is based on assessing the detection of hard samples, rather than downstream performance. Given that the metric is based on classification rates, it is easily confused with the main task on these datasets. I encourage the authors to give a distinctive name to the metric(s), to make it more clear and explicit for other readers and new comers to the HCM literature.
> >
> > Other aspects I appreciate is that the presentation style was updated, that the new experiments include a medical image dataset, and that the paper has been reorganized according to other feedback. The new version of the paper is great!

---

> > > ### Author Response · Authors · 2023-11-21
> > > **Thank you for the response & score increase**
> > >
> > > We are delighted we were able to address your concerns and are grateful for your increased score to an 8. We appreciate your constructive feedback and concrete guidance, which enabled us to improve the paper!
> > >
> > > As per the reviewers suggestion, we have made the detection metric more clear both in text and by referring to it with a distinctive name to avoid confusion. e.g. D-AUPRC to refer to detection AUPRC. We have updated the uploaded manuscript with these clarificiations.
> > >
> > > Thanks again for your time, suggestions and positive feedback!
> > >
> > > Paper 7682 Authors

---

> ### Author Response · Authors · 2023-11-16
> **Response to Reviewer SFiV [Part 2/3]**
>
> ### (A) Suitability for datasets & benchmarking
>
> We thank the reviewer for acknowledging the suitability of our paper for a benchmarking track. We wish to highlight that this year the ICLR call for papers includes a track specifically for **datasets and benchmarks** — which we noted as the primary submission area for this work. With this in mind, our work responds to the recent calls for more rigorous benchmarking and understanding of existing ML approaches, as highlighted by Guyon (2022), Lipton & Steinhardt (2019), and Snoek et al. (2018).
>
> Our work provides four main contributions to the field: **(1) Hardness taxonomy:** we formalize a systematic taxonomy of sample-level hardness types; **(2) Benchmarking framework:** to evaluate the strengths of different HCMs across hardness types; **(3) Systematic and quantitative HCM evaluation:** prior to our study, no research had comprehensively evaluated HCMs across various hardness types nor directly evaluated HCMs' capabilities to correctly detect hardness — we do for 13 different HCMs across 8 different hardness types;** (4) Insights:** our benchmark provides novel insights into the capabilities of different HCMs when dealing with different hardness types, exposing gaps and opportunities in current HCMs and offering practical tips for researchers and practitioners. We hope the H-CAT framework will both add rigor to the HCM field and address the critical need for a structured and empirical approach to understanding and improving HCM methodologies.
>
> We also highlight that similar analyses proposing a framework and benchmark have also previously been accepted by ICLR:
>  - A Fine-Grained Analysis on Distribution Shift (Wiles et al., 2022)
> - BREEDS: Benchmarks for Subpopulation Shift (Santurkar et al., 2021)
> - MEDFAIR: Benchmarking Fairness for Medical Imaging (Zong et al., 2023)
> - Long Range Arena: A Benchmark for Efficient Transformers (Tay et al., 2021)
> - A framework for benchmarking Class-out-of-distribution detection and its application to ImageNet (Galil et al., 2023)
>
> ---
> ### (B) Conclusions of the paper & known hardness type
>
> Our work enables the HCM field to rigorously evaluate for the first time, which is essential for the field to progress. We believe this will help the HCM field to move from the current ad hoc and qualitative state (as shown in **Table 1**) toward more comprehensive and rigorous evaluation across multiple hardness types.
>
> We emphasize that our benchmark also offers critical insights into current HCMs, which we have distilled into **(1) benchmarking takeaways**: to guide the future development of HCMs by highlighting their strengths and weaknesses; and **(2) practical takeaways**: to guide selection and usage of HCMs. Overall, this work highlights what types of hardness different HCMs are able to detect and shows the limitations of HCMs in detecting certain hardness types, which we hope will inspire new methodologies to address these limitations.
>
> We also wish to clarify the potential misunderstanding about the hardness type being known and make the distinction between the benchmark framework and HCMs. Our benchmark framework tests specific hardness types. This is vital for establishing a controlled benchmarking environment. Without such a ground truth, it would be challenging, if not impossible, to accurately assess and compare the capabilities of different HCMs. However, we emphasize that the HCM itself does not have knowledge of the hardness type when being evaluated. We apologize if this was unclear.
>
> Finally, regarding the concern of a single source of hardness. While in the main paper, we mainly focused on a single source of hardness at a time to disentangle effects, as mentioned in **Section 6**, we consider the case where multiple types of “hardness” manifest simultaneously (e.g. Near-OoD and mislabeling together) and provide an example of simultaneous hardness in **Appendix D**. We hope this clarifies the Reviewer's concern.

---

> ### Author Response · Authors · 2023-11-16
> **Response to Reviewer SFiV [Part 3/3]**
>
> ### (C) Clarification: What is data-centric AI & HCMs vs creating high-quality datasets
>
> We thank the Reviewer for the opportunity to clarify the role of HCMs within data-centric AI. Data-centric AI focuses on systematic methods to enhance data quality — with hardness characterization being fundamental by identifying such hard examples in an algorithmic manner. We fully agree with the Reviewer we need to create high-quality datasets that eliminate corrupted examples — we emphasize that HCMs are one potential algorithmic tool to achieve this. Algorithmic identification is especially important when manual curation is impractical, for example at scale. Hence, we can think of HCMs being the algorithmic mechanism to achieve the data quality goals of data-centric AI. To illustrate the practical value and need for HCMs in creating high-quality datasets, we highlight that  (i) THE _Cleanlab HCM_ was used to audit ML benchmark datasets for mislabeling (Northcutt et al, 2021b), (ii) the _Data Maps HCM_ was used to audit and create high-quality RNA data in biology [R1], and (iii) the _Data-IQ HCM_ was used to audit and curate high-quality sleep staging data [R2].
>
> ---
>
> ### (D) Training models without HCMs
>
> We wish to clarify the goal of this work is to evaluate the detection capabilities of various HCMs in identifying different types of data hardness, rather than assessing their curation impact on downstream model performance (e.g. suggested baseline without HCMs). The reason is previous research has already demonstrated the value of using HCMs to curate datasets to train downstream models. In contrast, our work complements the existing literature which already assesses the suggested baseline and extends previous work by providing a detailed analysis of how effectively different HCMs can identify various hardness types, which is a novel and necessary perspective for the field.
>
> ---
>
> ### (E) Evaluating data hardness.
>
> We agree with the Reviewer that, in the real world, data hardness should be evaluated in a data-driven way (e.g. with data science and ML methods): this is precisely what an HCM does when used to identify and audit hard samples in a dataset. However, benchmarking and evaluating HCMs and their detection capabilities requires the creation of a controlled benchmarking environment. We note real datasets do not have ground truth of hardness; hence, to accurately assess the detection capabilities of HCMs, we need to establish a ground truth in our benchmark which we do via clean datasets (CIFAR and MNIST). We clarify this selection is a strategic decision to establish a clear baseline and understand the specific capabilities of each HCM under known conditions. Moreover, these datasets are widely adopted in the HCM literature. We contend that the use of such datasets doesn't detract from the study's relevance; instead, it ensures that the evaluation of HCMs is precise.
>
> We also wish to clarify that indeed the hardness types defined in our framework are reflective of challenges and problems in real data. **Figure 7 (now moved to the main paper as Figure 3)** illustrates that the hardness types are representative of hardness types that could realistically occur in actual datasets.
>
> ---
>
> ### (F) Indirect measures vs our use of direct measures
>
> We thank the reviewer for the opportunity to clear up this misunderstanding. By 'indirect,' we refer to the HCM literature only assessing changes in downstream performance rather than the actual hardness detection task –- i.e. we don’t know if HCMs do the task they claim to do. In contrast, we address the gap in the literature as shown in Table 1 and directly evaluate the capability of HCMs to detect specific hardness types. This approach allows us to understand which types of hardness each HCM can effectively identify, rather than merely assessing model improvement on a curated dataset, which might obscure the actual capabilities of different HCMs on different hardness types, offering valuable insights for the development and application of these methods.
>
> ---
>
> ### (G) Writing updates
>
> We thank the reviewer for the suggestions to help improve the paper and its readability. We have made the following changes based on your suggestions.
>
> - Changed the citation color from blue to black to reduce the color on the page.
> - Reduced the in-text bolding, except to delineate sections.
> - Removed the color boxes in many places
> - We have also streamlined the description of the toolkit. This has allowed space for additional insights to be moved from the Appendix to the main paper
>
>
>
> ---
>
> ### References
> [R1] Nabi, Afshan, et al. "Discovering misannotated lncRNAs using deep learning training dynamics." Bioinformatics 39.1 (2023): btac821.
>
> [R2] Heremans, Elisabeth RM, et al. "U-PASS: an Uncertainty-guided deep learning Pipeline for Automated Sleep Staging." arXiv preprint arXiv:2306.04663 (2023).

---

### Author Response · Authors · 2023-11-16
**Response Overview**

We thank the Reviewers for their insightful and positive feedback, and their time during the review process!

We are encouraged the reviewers found our taxonomy "sound" (``R-SFiV``), "clearly articulated, and practically useful" (``R-kMBG``). They noted that "hardness is not well-defined anywhere" (``R-5UHg``) and hence is "great to define hardness more formally" (``R-zsTV``). By showing "what's missing in existing HCMs" (``R-kMBG``) this shows the "need for a systemic evaluation framework for HCMs" (``R-u8mR``). We are glad the reviewers deemed that the "systematic experimental design is strong" (``R-SFiV``) and the toolkit "well designed"(``R-SFiV``). This allowed for  "exhaustive"(``R-SFiV``) and "comprehensive benchmarking" (``R-kMBG``), providing "robust validation" (``R-zsTV``) and takeaways that are "valuable"(``R-zsTV``) and of  "greater significance" (``R-u8mR``) which is  "useful for the community" (``R-kMBG``).


We address specific questions and comments below. We have also updated and uploaded the manuscript with the following key changes highlighted in green.

- Additional experiments on a new medical x-ray dataset [Appendix D]
- Additional experiment to understand the effect of the perturbation severity on hardness scores [Appendix D]
- Moving content into the main paper from the appendix
- Formalism of each HCM [Appendix A]
- Writing updates for greater clarity and visual appeal

On the basis of our clarifications and updates, we hope we have addressed the Reviewers' concerns.

Thank you for your kind consideration!

Paper 7682 Authors

---

> ### Author Response · Authors · 2023-11-20
> **Dear Reviewers**
>
> Dear Reviewers
>
> We are sincerely grateful for your time and energy in the review process. We hope that our responses and manuscript updates have been helpful.
>
> Please let us know of any leftover concerns and if there was anything else we could do to address any further questions or comments!
>
> Thank you!
>
> Paper 7682 Authors

---

### Meta-Review · Area_Chair_FCVq · 2023-12-11

**Metareview:**

This paper was reviewed by five knowledgeable referees, whose main concerns included:
1. The presentation of the paper, which could be significantly improved by clearly stating the motivation for HCMs, emphasizing the contributions and restructuring certain sections (SFiV, zsTV)
2. Missing baselines which would not consider HCMs (SFiV)
3. The hardness characterization which appeared oversimplified (5UHg, kMBG, u8mR, SFiV)
4. The datasets considered, too small and toy-ish (CIFAR-10 and MNIST), limiting the usefulness of the takeaways (SFiV, 5UHg, kMBG)
5. The potentially limited applicability: assumptions to know the hardness types present in the data (zsTV), the unclear extension to other modalities (5UHg), and that hardness may change throughout training (zsTV)
6. The unclear impact of perturbations (kMBG)

The rebuttal addressed most of the reviewers' concerns. In particular, it clarified the motivation for HCMs, the contributions (while reminding the reviewers that datasets and benchmarks appear in the relevant topics), and by rewriting parts of the paper as suggested by the reviewers. The rebuttal noted that beyond MNIST/CIFAR a tabular dataset was considered as well as an image-based medical imaging dataset (added during discussion). The rebuttal also argued the hardness characterization and addressed the concerns w.r.t. potentially limited applicability and impact of perturbations. After rebuttal/discussion, 4/5 reviewers recommend acceptance. The fifth reviewer remains hesitant with the datasets chosen (small scale) and as a result questions the validity of the takeaways on larger scale datasets and different modalities. Yet, after discussion with the authors the reviewer communicates that they are willing the increase their final recommendation to 5. The authors share their concerns w.r.t. this reviewer. The AC acknowledges reading the concerns and taking them into consideration when making the decision. The AC agrees with the reviewers that considering larger scale datasets would strengthen the contribution, however, the AC also appreciates the potential of this benchmarking contribution to advance our understanding of hardness in machine learning. Therefore, the AC recommends to accept.

**Justification For Why Not Higher Score:**

The datasets considered are small scale, and the generalization of the takeaways to larger scale setups remains unclear.

**Justification For Why Not Lower Score:**

The benchmarking contribution holds the potential to advance our understanding of hardness characterization in machine learning, and show help rigorously evaluate these approaches quantitatively.

---

### Decision · Program_Chairs · 2024-01-16

Accept (poster)